# BI-CONTINUOUS AND COMPLETE $SE(2)$-INVARIANTS PARAMETRIZE ALL CLOUDS OF UNORDERED POINTS

## ABSTRACT

The most basic form of a rigid object is a cloud of unordered points, for example, a set of corners or other salient features. The rigid shape of a point cloud in the Euclidean plane is its $SE(2)$-equivalence class under rigid motion (a composition of translations and rotations). We introduce complete invariants (with no false negatives, no false positives) and a bi-Lipschitz continuous metric that satisfies all axioms, provides a 1-1 matching between points in clouds, and is computable in a quadratic time of the number $m$ of points. The realizability property implies that the space of all rigid clouds is efficiently parametrized by vectorial invariants like geographic coordinates. The new invariants justified that any of 130K+ molecules in the QM9 database is uniquely determined by the rigid shape of its atomic cloud.

## 1 MOTIVATIONS FOR NEW COMPLETE AND BI-CONTINUOUS INVARIANTS

Many real objects are *rigid* so that their shapes are preserved under *rigid motion* composed of translations and rotations in $\mathbb{R}^n$ Atz et al. (2021), which form the group $SE(n)$. The slightly weaker equivalence is by *isometries* (distance-preserving transformations), which form the group $E(n)$.

The basic input of a rigid shape is a cloud of $m$ unordered points in $\mathbb{R}^n$ Wang & Solomon (2019). The practical cases are dimensions $n \leq 3$ and larger numbers $m$ (hundreds) of unordered points without outliers Shi et al. (2021). Because of noise, repeated measurements of the same object can produce slightly different point clouds that cannot be exactly matched with the original one by rigid motion. If noise is ignored up to any threshold $\varepsilon > 0$, sufficiently many tiny perturbations make all clouds equivalent by the transitivity axiom: if $A \sim B$ and $B \sim C$, then $A \sim C$ Brink et al. (1997).

Hence all small deviations between rigid classes of point clouds should be distinguished, all these classes live in a continuous space of rigid clouds. This important space was continuously parametrized only for $m = 3$ points. Even the case of $m = 4$ unordered points was open, see Fig. 1.

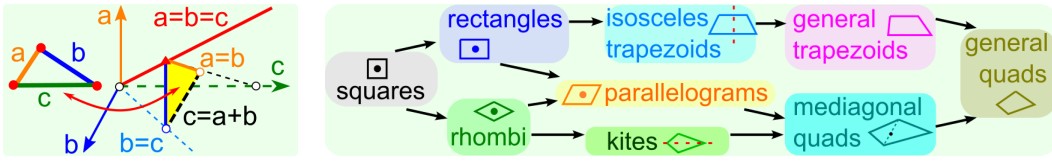

Figure 1: **Left**: the space of 3-point clouds $\{0 < a \leq b \leq c \leq a+b\}$ under isometry is parametrized by distances. **Right**: 4-point clouds were split only in discrete classes but live in a continuous space.

Machine learning mostly focused on discrete classifications (label prediction, clustering) or on improving various success measures for finite datasets, which can be considered discrete samples (of measure 0) in a continuous space of shapes. To make this approach generalizable to all real data outside finite datasets, we need to map continuous data spaces similar to a geographic map of Earth.

A continuous extension of machine learning needs new requirements because past accuracies were developed for discrete classifications or finite data. Though the key concepts of complete invariants and distance metrics were already studied Schmidt & Roth (2012); Li et al. (2021), Problem 1.1 introduces new conditions such as realizability and point matching that were not previously stated.

**Problem 1.1** (clouds and rigid motion can be replaced by any data and relations). *Find an invariant $I : \{clouds\ of\ unordered\ points\ in\ \mathbb{R}^n\} \to a\ space\ with\ a\ distance\ d\ satisfying\ the\ conditions\ below.*

*(a) Completeness: any clouds $A, B$ are related by a rigid motion of $\mathbb{R}^n$ if and only if $I(A) = I(B)$.*

*(b) Metric axioms: 1) $d(a, b) = 0 \Leftrightarrow a = b$; 2) $d(a, b) = d(b, a)$; 3) $d(a, b) + d(b, c) \geq d(a, c)$.*

*(c) Lipschitz continuity: there is a constant $\lambda$ such that if each point of a cloud $A \subset \mathbb{R}^n$ is perturbed up to Euclidean distance $\varepsilon$, then the invariant $I(A)$ changes by at most $\lambda\varepsilon$ in the metric $d$.*

*(d) Realizability: the image space $\{I(A) \mid all\ clouds\ A \subset \mathbb{R}^n\ of\ m\ unordered\ points\}$ is parametrized so that one can reconstruct $A$ up to rigid motion from any realizable value of $I$.*

*(e) Point matching: there is a constant $\mu$ such that a distance $d = d(I(A), I(B))$ guarantees a rigid motion matching all $m$ unordered points of clouds $A, B$ up to Euclidean distance $\mu d$.*

*(f) Computability: for a fixed dimension $n$, the invariant $I$, metric $d$, reconstruction in (d), and 1-1 point matching in (e) are all computable in polynomial time of the number $m$ of points.* ∎

The completeness (or injectivity) in (1.1a) means that an invariant $I$ finalizes the discriminative approach and provably distinguishes all clouds $A \not\cong B$ (not only from a finite dataset) that cannot be matched by rigid motion, so $I$ is a descriptor with *no false negatives* and *no false positives*. The universal approximation aims for the completeness of infinite-size invariants Maron et al. (2019); Keriven & Peyré (2019); Yarotsky (2022), so polynomial time in (1.1f) makes all conditions harder.

A complete invariant can give a discontinuous metric, say $d(A, B) = 1$ for all non-equivalent clouds without quantifying the similarity of near-duplicates. The Lipschitz continuity in (1.1c) is stronger than the classical $\varepsilon - \delta$ continuity because the Lipschitz constant $\lambda$ is universal for all inputs and perturbations. Due to the first axiom in (1.1b), any metric $d$ detects rigidly equivalent clouds by checking if $d(A, B) = 0$. Without the first axiom, many more distances including the zero $d \equiv 0$ satisfy the other axioms and are called *pseudo-metrics* Brécheteau (2019). If the third axiom in (1.1b) fails with any error $\varepsilon > 0$, results of clustering may not be trustworthy Rass et al. (2024).

The realizability in (1.1d) implies that the invariant $I$ is an invertible 1-1 map from the complicated *Cloud Rigid Space* $\mathrm{CRS}(\mathbb{R}^n; m)$ of classes of clouds under rigid motion to the explicitly parametrized space $I(\mathrm{CRS}(\mathbb{R}^n; m))$ of realizable values. Then with 100% certainty, we can sample any realizable value in $I(\mathrm{CRS}(\mathbb{R}^n; m))$ and reconstruct its cloud $A \subset \mathbb{R}^n$ up to rigid motion.

Point matching in (1.1e) can be interpreted as the Lipschitz continuity of the inverse map $I^{-1}$ so that any close values $I(A), I(B)$ guarantee the closeness of $A, B$ under rigid motion. Conditions (1.1c,e) mean that the metric $d$ is bi-Lipschitz: $\varepsilon/\mu \leq d(I(A), I(B)) \leq \lambda\varepsilon$, where $\varepsilon$ is the minimum perturbation needed to match all points of $A, B$. One can define metrics satisfying (1.1a,b,c) by minimizing deviations of unordered points over infinitely many rotations. Polynomial time in (1.1f) for all ingredients makes Problem 1.1 notoriously hard, previously solved only for $m = 3$ points.

Conditions (1.1a,b,c,f) and (1.1d,e,f) formalize the *discriminative* and *generative* goals, respectively. A full solution to Problem 1.1 will imply that the rigid classes of clouds can be efficiently visualized in the *moduli* space $I(\mathrm{CRS}(\mathbb{R}^n; m))$ replacing any latent space of non-invariants or incomplete (or discontinuous or non-realizable) invariants. Geographically, $I(\mathrm{CRS}(\mathbb{R}^n; m))$ can be compared with Earth's map, where any location can be reconstructed with all properties (altitude, precipitation, images, ...) from the latitude and longitude coordinates in known (realizable) ranges.

**Contributions**: the new invariant Nested Distributed Projection solves Problem 1.1 for all clouds of $m$ unordered points in dimension $n = 2$. Any cloud $A \subset \mathbb{R}^n$ can be reconstructed from a small part of the invariant (a vector in $\mathbb{R}^{n(m-(n+1)/2)}$) whose realizability in (1.1d) is guaranteed by explicitly written inequalities. Hence coordinates of this vector can be chosen in known ranges like latitude and longitude on Earth maps. The appendices cover all dimensions $n > 2$ and visualize geographic-style maps of cloud spaces for $m = 4$ points in $\mathbb{R}^2$. The implementation is in supplementary materials.

## 2 PAST WORK ON CLOUD CLASSIFICATIONS RELATED TO PROBLEM 1.1

This section reviews past approaches to Problem 1.1, which was open for $m > 3$ points even in $\mathbb{R}^2$.

**Ordered points**. Kendall's shape theory Kendall et al. (2009) studies $m$ ordered points $p_1, \ldots, p_m \in \mathbb{R}^n$ under isometries from the Euclidean group $\mathrm{E}(n)$. In this case, a complete invariant is the distance matrix Schoenberg (1935); Kruskal & Wish (1978) or the Gram matrix of scalar products $p_i \cdot p_j$, see (Weyl, 1946, chapter 2.9), Villar et al. (2021). A brute-force extension to $m$ unordered points requires $m!$ matrices due to $m!$ permutations, which is ruled out by (1.1f).

**Point cloud registration** for unordered points samples rotations Lin et al. (1986); Yang et al. (2020) and uses scale-invariant features Lowe (1999; 2004) to approximately match clouds. Trying to sort points along a fixed direction or in a clockwise order around their center of mass leads to discontinuities because distant points can have equal projections to a line or a circle. A basis (say, of principal directions) of a cloud Toews & Wells III (2013); Rister et al. (2017); Spezialetti et al. (2019); Zhu et al. (2022); Kurlin (2024) is similarly unstable under perturbations of points in cases of high symmetry, e.g. when eigenvalues degenerate, which often happens in real molecules for our main application. Converting a cloud by using extra parameters into a more complex object such as a continuous field $\mathbb{R}^3 \to \mathbb{R}$ Chauvin et al. (2022) or the persistent homology transform leads to the harder analog of Problem 1.1 for continuous surfaces instead of discrete clouds Turner et al. (2014).

**Geometric Deep Learning** Bronstein et al. (2021) studies neural networks that guarantee invariance or equivariance Thomas et al. (2018); Kondor & Trivedi (2018); Cohen et al. (2019); Fuchs et al. (2020); Deng et al. (2021). An *equivariant* descriptor $E$ satisfies the weaker condition $E(f(A)) = T_f(E(A))$ for any rigid motion $f$ of a cloud $A$, where $T_f$ may not be the identity as required for invariants Satorras et al. (2021); Chen et al. (2021); Aronsson (2022); Assaad et al. (2023); Xu et al. (2022); Su et al. (2022). Any linear combination of points such as the center of mass is equivariant but cannot distinguish clouds under translation. Equivariants were used for predicting forces acting on atoms to move them to a more optimal configuration. These time-dependent clouds $A_t$ can be studied directly by their invariant values $I(A_t)$ without intermediate forces. So neural networks optimize millions of parameters as in (Goyal et al., 2021, Table 4) to improve accuracies Dong et al. (2018); Akhtar & Mian (2018); Laidlaw & Feizi (2019); Guo et al. (2019); Colbrook et al. (2022) but need re-training any for new data. All such networks will have better generalizability if the inputs are invariants that satisfy the conditions of Problem 1.1 for all possible point clouds in $\mathbb{R}^n$.

**General metrics** between fixed clouds extend to their rigid classes by minimization over infinitely many rigid motions Huttenlocher et al. (1993); Chew & Kedem (1992); Chew et al. (1999). In $\mathbb{R}^2$, the time $O(m^5 \log m)$ Chew et al. (1997) for the Hausdorff distance Hausdorff (1919) will be improved in Theorem 5.3 to $O(m^{3.5} \log m)$ for a new metric, see approximations in Goodrich et al. (1999). The Gromov-Hausdorff and Gromov-Wasserstein metrics Mémoli (2011) are defined for metric-measure spaces also by minimizing over infinitely many correspondences between points, but cannot be approximated with a factor less than 3 in polynomial time unless P=NP, see Corollary 3.8 in Schmiedl (2017) and polynomial algorithms for partial cases in Majhi et al. (2024). Also, computing a metric between rigid classes of clouds is only a small part of Problem 1.1. Indeed, to efficiently navigate on Earth, in addition to distances between cities, we need a satellite-type view of the whole planet and hence a realizable bi-continuous invariant $I$, which can be considered an analog of the latitude and longitude coordinates on Earth.

**Can we 'sense' a shape?** Informally, Problem 1.1 asks the questions 'same or different clouds, and how much different?' The related problem 'Can we hear the shape of a drum?' Kac (1966) has the negative answer in terms of 2D polygons that are indistinguishable by spectral invariants Gordon et al. (1992a;b); Reuter et al. (2006); Cosmo et al. (2019); Marin et al. (2021). Problem 1.1 looks for stronger invariants that can completely 'sense' (not only 'hear') the rigid shape of any cloud.

**The simple cases** when Problem 1.1 was fully solved are only $n = 1$ or $m \leq 3$. In dimension $n = 1$, any rigid motion of $\mathbb{R}$ is a translation, so $\mathrm{CRS}(\mathbb{R}; m)$ of $m$ points $p_1, \ldots, p_m \in \mathbb{R}$ is the space $\mathbb{R}_+^{m-1}$ of sequential inter-point distances $d_i = p_{i+1} - p_i > 0$ for $i = 1, \ldots, m-1$. Including reflections, the *Cloud Isometry Space* $\mathrm{CIS}(\mathbb{R}; m)$ is the quotient of $\mathbb{R}_+^{m-1}$ under the cyclic equivalence $(d_1, \ldots, d_{m-1}) \sim (d_{m-1}, \ldots, d_1)$. For clouds of only $m = 2$ points in any dimension $n \geq 1$, $\mathrm{CRS}(\mathbb{R}^n; 2)$ is parametrized by a single inter-point distance $d > 0$. The final known case is $m = 3$ due to the SSS theorem saying that any triangles are congruent (isometric) if and only if they have the same side lengths. The space $\mathrm{CRS}(\mathbb{R}^n; 3)$ of 3-point clouds under isometry has the geographic-style parametrization $\{0 < a \leq b \leq c \leq a + b\}$ by inter-point distances $a, b, c$.

Problem 1.1 asks for a similarly explicit parametrization of $\mathrm{CRS}(\mathbb{R}^n; m)$ for all $m \geq 4$ and $n \geq 2$.

**Partial solutions** include the extensions Delle Rose et al. (2024); Hordan et al. (2024) of the Weisfeiler-Leman test Leman & Weisfeiler (1968), giving a binary answer Brass & Knauer (2000; 2004) by distinguishing all non-isometric clouds but without Lipschitz continuous metrics.

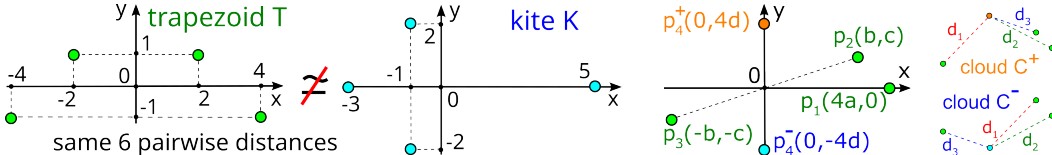

Figure 2: Non-isometric clouds of 4 points with the same 6 pairwise distances. **Left**: the trapezoid $T$ has points $(\pm 2, 1)$, $(\pm 4, -1)$. The kite $K$ has $(5, 0)$, $(-3, 0)$, $(-1, \pm 2)$. **Right**: the infinite family of non-isometric clouds $C^+ \not\simeq C^-$ sharing $p_1, p_2, p_3$ and depending on parameters $a, b, c, d > 0$.

Attempting to extend the SSS theorem, we can consider the Sorted Distance Vector (SDV) of all $\frac{m(m-1)}{2}$ inter-point distances between $m \geq 4$ unordered points. This SDV distinguishes all non-isometric clouds in general position in $\mathbb{R}^n$, see Boutin & Kemper (2004), but not infinitely many 4-point clouds even in $\mathbb{R}^2$, see Fig. 2. The SDV was strengthened Widdowson & Kurlin (2022) to the Pointwise Distance Distribution (PDD), which still cannot distinguish infinitely many non-isometric clouds in $\mathbb{R}^3$ (Pozdnyakov & Ceriotti, 2022, Fig. S4). All these counter-examples were distinguished by the Simplexwise Centered Distributions from Widdowson & Kurlin (2023), which satisfy (1.1a,b,c,f) but not (1.1d,e). Distance-based invariants do not allow easy realizability already for $m = 4$ points in $\mathbb{R}^2$ whose 6 inter-point distances should satisfy a non-trivial polynomial equation saying that the tetrahedron on 4 points has volume 0 in $\mathbb{R}^2$. Hence random distances between unordered points are realized by a point cloud in $\mathbb{R}^2$ with probability 0 Duxbury et al. (2016).

# 3 COMPLETE INVARIANTS OF UNORDERED CLOUDS UNDER RIGID MOTION

Any point $p = (x_1, \ldots, x_n) \in \mathbb{R}^n$ has *Euclidean* norm $|p| = \sqrt{\sum_{i=1}^{n} x_i^2}$. Any points $p$ and $q = (y_1, \ldots, y_n) \in \mathbb{R}^n$ are also interpreted as vectors, have the *Euclidean* distance $|p - q|$ and the *scalar* (dot) product of $p \cdot q = \sum_{i=1}^{n} x_i y_i$. Any vectors $p \perp q$ are *orthogonal* if and only if $p \cdot q = 0$.

While past representations used one basis (say, of principal directions of a given cloud $A \subset \mathbb{R}^n$), this section introduces a new representation based on variable projections that depend on $n - 1$ ordered points in $C$ consisting of $m$ unordered points. For simplicity, we consider dimension $n = 2$ when we have only $m$ choices for a single point $p \in A$. The appendices discusses the general case $n \geq 2$.

Fig. 3 summarizes the new invariant Nested Distributed Projection $I = \text{NDP}$ and Nested Bottleneck Metric $d = \text{NBM}$, which solve Problem 1.1 for $n = 2$, extended to $n > 2$ in the appendices.

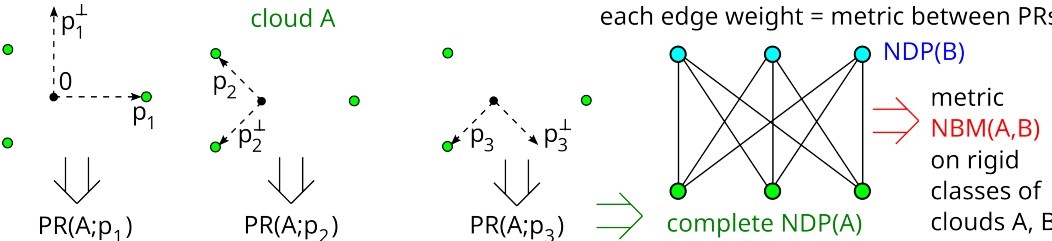

Figure 3: A Point-based Representation (PR) encodes a cloud $A$ in the basis of a point $p \in A$. All PRs are combined into the complete invariant $\text{NDP}(A)$. NDPs are compared by the Nested Bottleneck Metric (NBM) computed from a graph $\Gamma(A, B)$ with weights = distances between PRs.

For any cloud $A \subset \mathbb{R}^2$ of $m$ unordered points, the *center of mass* is $O(A) = \frac{1}{m} \sum\limits_{p \in A} p$. Shift $A$ so that $O(A)$ is the origin $0 \in \mathbb{R}^2$. For any $p = (x_1, x_2) \in A$, the vector $p^{\perp} = (-x_2, x_1)$ is orthogonal to $p$, so $p \cdot p^{\perp} = 0$, which holds even if $p = 0$. If $p$ is not at the origin (center of mass of $A$), we use the orthogonal basis $p, p^{\perp}$ to represent all other points of $A$. Definition 3.1 makes sense for $p = 0$.

**Definition 3.1** (point-based representation $\mathrm{PR}(A; p_1))$. *Let $A \subset \mathbb{R}^2$ be a cloud of points with the center of mass at the origin $0$. Fix a point $p = (x, y) \in A$ and set $p^{\perp} = (-y, x)$. For any $q \in A - \{p\}$, the $2 \times (m-1)$ matrix $M(A; p)$ has a column of the scalar products $q \cdot p, q \cdot p^{\perp}$. The point-based representation of $A$ with respect to $p$ is the pair $\mathrm{PR}(A; p) = \left[ |p|^2, M(A; p) \right]$.* ∎

We use $|p|^2$ and scalar products to make all components polynomial (smooth) in point coordinates. The matrix $M(A; p)$ has two rows (ordered according to $p, p^{\perp}$) and $m - 1$ unordered columns, so $M(A; p)$ can be considered a *fixed cloud* of $m - 1$ unordered points, not under rigid motion in $\mathbb{R}^2$.

**Example 3.2** (regular polygons in $\mathbb{R}^2$). *(a) For $m \geq 2$, let $A_m = \{R \exp \frac{2\pi i \sqrt{-1}}{m}\} \subset \mathbb{R}^2$, $i = 1, \ldots, m$, be the vertex set of a regular $m$-sided polygon. Then $A_m$ has the center of mass $O(A_m) = (0, 0)$ at the origin and is inscribed in the circle of the radius $R = R(A_m)$. In Definition 3.1, choose the point $p = (R, 0) \in A_m$, which doesn't affect $\mathrm{PR}(A_m; p)$ due to the rotational symmetry of $A_m$. Then the matrix $M(A_m; p)$ consists of $m - 1$ columns $\begin{pmatrix} R^2 \cos(2\pi i/m) \\ R^2 \sin(2\pi i/m) \end{pmatrix}$, $i = 1, \ldots, m - 1$. The point-based representation is the pair $\mathrm{PR}(A_m; p) = \left[ R^2, \left( \begin{pmatrix} R^2 \cos \frac{2\pi i}{m} \\ R^2 \sin \frac{2\pi i}{m} \end{pmatrix} \right)_{i=1}^{m-1} \right]$.*

*(b) Let the cloud $B_m \subset \mathbb{R}^2$ be $A_m$ after adding the extra point at the origin $0 \in \mathbb{R}^2$. For any point $p \in A_m$, the new point-based representation $\mathrm{PR}(B_m; p)$ is obtained from $\mathrm{PR}(A_m; p)$ above by adding the zero column to the matrix $M(A_m; p)$. For the new point at the origin $0$, we get $\mathrm{PR}(B_m; 0) = [0, M(B_m; 0)]$, where $M(B_m; 0)$ is the $2 \times m$ matrix consisting of zeros.* ∎

Table 1: Acronyms and references of all key concepts in the paper.

| | | | | | |
|---|---|---|---|---|---|
| PR | Point-based Representation | Def 3.1 | BD | Bottleneck Distance | Def 4.1 |
| NDP | Nested Distributed Projection | Def 3.4 | NBM | Nested Bottleneck Metric | Def 4.4 |
| NCP | Nested Compress. Projection | Def 3.4 | SRV | Sorted Radial Vector | Def 6.1 |
| BMD | Bottleneck Matching Distance | Def 4.3 | SDV | Sorted Distance Vector | Def 6.1 |
| PDD | Pointwise Distance Distribution | Def 6.1 | CRS | Cloud Rigid Space | Cor 5.5 |

**Theorem 3.3** (realizability of abstract PR). *Let $s > 0$ and $M$ be any $2 \times (m-1)$ matrix for $m \geq 2$. The pair $[s, M]$ is realizable as a point-based representation $\mathrm{PR}(A; p)$ for a cloud $A \subset \mathbb{R}^n$ of $m$ unordered points with $O(A) = 0$ and a point $p \in A$ if and only if $s + \sum\limits_{j=1}^{m-1} M_{1j} = 0 = \sum\limits_{j=1}^{m-1} M_{2j}$.* ∎

In Theorem 3.3, $s = |p|^2$ is the squared distance from a point $p \in A$ to $0 \in \mathbb{R}^2$. The equations mean that the sums of scalar products $(q \cdot p)$ and $(q \cdot p^{\perp})$ for all $q \in A$ equal to $0$, which is equivalent to $\sum q \in A = 0$ meaning that the center of mass $O(A)$ is $0$. Hence $s > 0$ and $m - 2$ columns of $M$ can be considered free parameters, which uniquely determine the remaining column of $M$.

Definition 3.4 combines point-based representations $\mathrm{PR}(A; p)$ for all points $p \in A$ into one invariant NDP (Nested Distributed Projection) that will be proved to satisfy all conditions of Problem 1.1.

The major advantage of NDP is its applicability to all real clouds $A \subset \mathbb{R}^2$ without any requirement of general position. Some points of a cloud $A$ may coincide, so $A$ can be a multiset of points.

**Definition 3.4** (invariants NDP, NCP). *Let $A \subset \mathbb{R}^2$ be any cloud of $m$ unordered points. The Nested Distributed Projection $\mathrm{NDP}(A)$ is the unordered set of $\mathrm{PR}(A; p)$ for all $p \in A$. If $k > 1$ representations $\mathrm{PR}(A; p)$ are equal then we collapse them to one representation with the weight $k/m$. The resulting set of unordered PRs with weights is the Nested Compressed Projection $\mathrm{NCP}(A)$.* ∎

For the vertex cloud $A_m$ from Example 3.2, the Nested Distributed Projection $\text{NDP}(A_m)$ consists of $m$ identical representations, so $\text{NCP}(A_m)$ is the single representation $\text{PR}(A_m; p)$ with weight 1. The invariant NDP is an expanded version of the NCP, where all PRs have equal weights $1/m$.

The full invariant $\text{NDP}(A)$ includes the faster *Radial Distance Invariant* $\text{RDI}(A)$ of only squared distances $|p|^2$ to the center of mass $O(A) = 0 \in \mathbb{R}^2$ from all points $p \in A$. If $A$ has a distinguished point $p$, e.g. a special atom in a molecule, the point-based representation $\text{PR}(A; p)$ is invariant.

**Theorem 3.5** (completeness of NDP). *The Nested Distributed Projection is complete in the sense that any clouds $A, B \subset \mathbb{R}^2$ of $m$ unordered points are related by rigid motion in $\mathbb{R}^2$ if and only if $\text{NDP}(A) = \text{NDP}(B)$ so that there is a bijection $\text{NDP}(A) \to \text{NDP}(B)$ matching all PRs.* ∎

Under a mirror reflection, for any point $p \in A$, one can assume after applying rigid motion that the basis $p, p^\perp$ maps to its mirror image $p, -p^\perp$. The mirror image $\bar{A}$ has $\text{NDP}(\bar{A})$ equal to $\overline{\text{NDP}}(A)$ that is obtained from $\text{NDP}(A)$ by reversing all signs in the last row of $M(A; p)$ for every $p \in A$.

The completeness of $\text{NDP}(A)$ under rigid motion in Theorem 3.5 implies the completeness of the pair $\text{NDP}(A), \overline{\text{NDP}}(A)$ under isometry including reflections. Further work can focus simplifying this pair to a smaller invariant while keeping the completeness. Since a bijection $\text{NDP}(A) \to \text{NDP}(B)$ between all (uncollapsed) PRs induces a bijection $\text{NCP}(A) \to \text{NCP}(B)$ respecting all weights of collapsed PRs, Theorem 3.5 implies the completeness of NCP under rigid motion in $\mathbb{R}^2$.

## 4 The nested bottleneck metric (NBM) on complete invariants

We will define the metric NBM on invariants NDP by using the bottleneck distance BD in Definition 4.1, a metric on point-based representations (PRs) in Definition 4.2, and a bottleneck matching distance in Definition 4.3. Extensions and proofs in high dimensions are in appendix D.

**Definition 4.1** (bottleneck distance BD). *For any $v = (v_1, \ldots, v_n) \in \mathbb{R}^n$, the* Minkowski *norm is $||v||_\infty = \max\limits_{i=1,\ldots,n} |v_i|$. For clouds $A, B \subset \mathbb{R}^n$ of $m$ unordered points, the* bottleneck distance $\text{BD}(A, B) = \inf\limits_{g:A \to B} \sup\limits_{p \in A} ||p - g(p)||_\infty$ *is minimized over all bijections $g : A \to B$.* ∎

Though the bottleneck distance is introduced as a minimum for $m!$ bijections $A \to B$ between fixed $m$-point clouds, Theorem 6.5 in Efrat et al. (2001) computes $\text{BD}(A, B)$ in time $O(m^{1.5} \log^2 m)$. The brute-force extension of $\text{BD}(A, B)$ under rigid motion requires a minimization for infinitely many rotations. However, $\text{NDP}(A)$ consists of only $m$ point-based representations $\text{PR}(A; p) = [|p|^2, M(A; p)]$, one for each $p \in A$. The fast algorithm for BD above can compare any $2 \times (m-1)$ matrices $M(A; p)$ and $M(B; q)$ as fixed clouds of unordered columns (points in $\mathbb{R}^2$).

In Definition 4.2, the notation $M/R$ means that all elements of the matrix $M(A; p)$ are divided by the *radius* $R(A) = \max\limits_{p \in A} |p|$ of a cloud $A$. Then PRM and further metrics have units of original points, e.g. in meters. One more division by $R(A)$ makes metrics invariant under uniform scaling.

**Definition 4.2** (Point-Based Representation Metric). *Let $\text{PR}(A; p), \text{PR}(B; q)$ be point-based representations of clouds $A, B \subset \mathbb{R}^2$ of $m$ unordered points for $p \in A$ and $q \in B$, respectively, see Definition 3.1. The* Point-based Representation Metric *between the PRs above is defined as*
$$\text{PRM} = \max\{\, |\,|p| - |q|\,|, \, |R(A) - R(B)|, \, w_M \,\}, \text{ where } w_M = \text{BD}\left(\frac{M(A; p)}{R(A)}, \frac{M(B; q)}{R(B)}\right). \quad ∎$$

We defined PRM as the maximum of 3 metrics to later get the simplest Lipschitz constant $\lambda = 2$ in (1.1d). Replacing the maximum with (say) a sum gives a metric with a higher $\lambda$ depending on $m$.

**Definition 4.3** (bottleneck matching distance BMD($\Gamma$)). *Let $\Gamma$ be a complete bipartite graph with $m$ white vertices and $m$ black vertices so that every white vertex is connected to every black vertex by an edge $e$ of a weight $w(e) \geq 0$. A* vertex matching *in $\Gamma$ is a set $E$ of $m$ disjoint edges of $\Gamma$. The* weight $W(E) = \max\limits_{e \in E} w(e)$ *is the largest weight in $E$. The* bottleneck matching distance *of the weighted bipartite graph $\Gamma$ is $\text{BMD}(\Gamma) = \min\limits_{E} W(E)$ is minimized over all vertex matchings.* ∎

Because $\Gamma$ is bipartite, any edge from a vertex matching $E$ joins a white vertex with a black vertex. Then $\text{BMD}(\Gamma)$ is minimized for all bijections $E$ between all white vertices and all black vertices of

$\Gamma$ similar to Definition 4.1. Definition 4.4 builds a graph $\Gamma(A, B)$ on all point-based representations of $A, B \subset \mathbb{R}^n$ and introduces the Nested Bottleneck Metric $\text{NBM}(A, B)$ as BMD of $\Gamma(A, B)$.

**Definition 4.4** (NBM : Nested Bottleneck Metric). *Let clouds $A, B \subset \mathbb{R}^2$ consist of $m$ unordered points. The complete bipartite graph $\Gamma(A, B)$ has $m$ white vertices (one for each $p \in A$) and $m$ black vertices (one for each $q \in B$). Any edge $e$ of $\Gamma(A, B)$ has endpoints associated with point-based representations $\text{PR}(A; p)$, $\text{PR}(B; q)$, and the weight $w(e) = \text{PRM}\big( \text{PR}(A; p), \text{PR}(B; q) \big)$. The Nested Bottleneck Metric is defined as $\text{NBM}(A, B) = \text{BMD}(\Gamma(A, B))$.* ∎

**Example 4.5** (4-point clouds $C^{\pm}$). *In $\mathbb{R}^2$, consider the 4-point clouds $C^{\pm} = \{p_1, p_2, p_3, p_4^{\pm}\}$, where $p_1 = (4a, 0)$, $p_2 = (b, c)$, $p_3 = -p_2 = (-b, -c)$, $p_4^+ = (0, 4d)$, and $p_4^- = (0, -4d)$ for parameters $a, b, c, d \geq 0$, see Fig. 2. We explicitly compute $\text{NDP}(C^{\pm})$ in the appendices to distinguish all clouds $C^+ \not\cong C^-$. Fig. 4 shows the new metric NBM by fixing one of 4 pairs of parameters, e.g. $b = c = 2$ in the top left picture, while other parameters vary between 0 and 4. The simultaneous swapping $a \leftrightarrow d$, $b \leftrightarrow c$ maps each cloud $C^{\pm}$ to its mirror image in the diagonal $x = y$ in $\mathbb{R}^2$, hence the metric between $C^{\pm}$ remains the same, which explains the symmetry of the plots in Fig. 4 (top). The metric NBM is positive, implying that that $C^+ \not\cong C^-$, except in the singular cases below.*

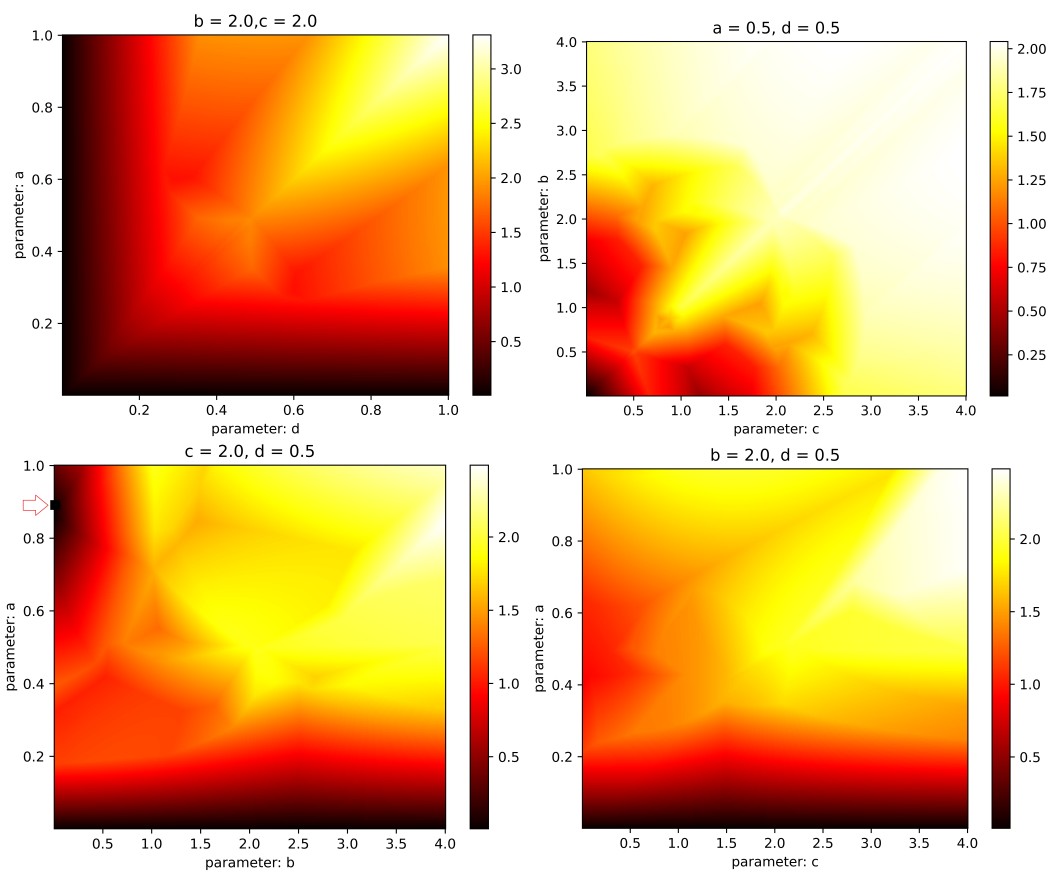

Figure 4: The Nested Bottleneck Metric NBM from Definition 4.4 for the 4-point clouds $C^{\pm} \subset \mathbb{R}^2$ that are not distinguished by their 6 pairwise distances in Fig. 2, see details in Example 4.5.

*If $a = 0$ or $d = 0$ or $b = c = 0$, the clouds are related by a 2-fold rotation around the origin 0. If $a = \frac{\sqrt{3}}{2} \approx 0.87$, $b = 0$, $c = 2$, $d = 0.5$, then $C^+$ consists of the vertices $(0, \pm 2), (2\sqrt{3}, 0)$ of an equilateral triangle, where $(0, 2)$ is the double point $p_2 = p_4^+$. For the same parameters, $C^-$ has the same points, but now $(0, -2)$ is the double point $p_3 = p_4^-$. Because these degenerate clouds are related by rotation, $\text{NBM} = 0$ in the black pixel at $a = \frac{\sqrt{3}}{2} \approx 0.87$, $b = 0$ in Fig. 4 (bottom left).* ∎

# 5 LIPSCHITZ-BI CONTINUITY AND POLYNOMIAL TIME ALGORITHMS

In this section, all algorithms for $m$ unodered points have polynomial times in $m$ in the RAM model.

**Theorem 5.1** (Lipschitz continuity of NBM). *Let $B \subset \mathbb{R}^2$ be obtained from a cloud $A \subset \mathbb{R}^2$ by perturbing every point of $A$ up to Euclidean distance $\varepsilon$. Then $\mathrm{NBM}(A, B) \leq 6\varepsilon$.* ∎

To illustrate Theorem 5.1, we generated uniformly random clouds $A$ in the unit square and cube. To get a perturbation $B$ of $A$, we shifted every point of $A$ by adding a uniformly random value in $[-\varepsilon, \varepsilon]$ to each coordinate, where $\varepsilon \in [0.01, 0.1]$ is a noise bound. Fig. 5 shows how the Nested Bottleneck Metric (NBM, averaged over several clouds) increases with respect to the noise bound.

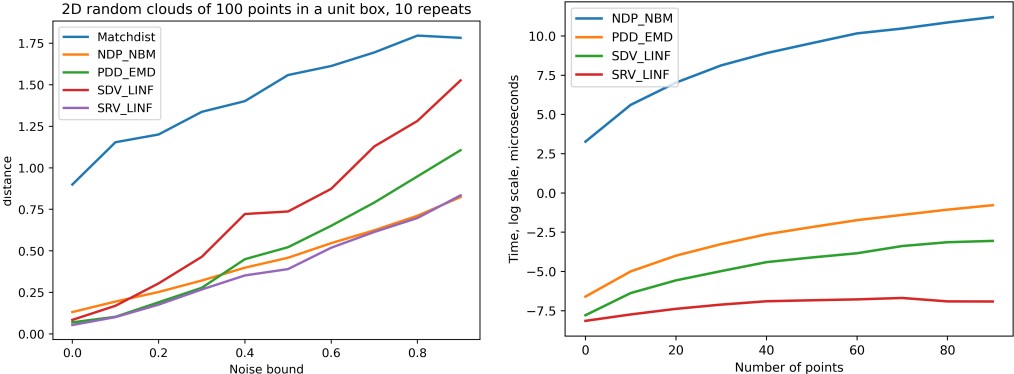

Figure 5: **Left**: $\mathrm{NBM}(A, B)$ between a random cloud $A$ and its $\varepsilon$-perturbation $B$ increases at most linearly in the noise bound $\varepsilon$ with a Lipschitz constant $\lambda_2 \approx 2$ as expected by Theorem 5.1. The experiments in section 6 estimated that $\lambda_2 \approx 2.76$. **Right**: the times (in microseconds, log scale).

**Theorem 5.2** (time of NDP). *For any cloud $A \subset \mathbb{R}^2$ of $m$ unordered points, the Nested Distributed Projection $\mathrm{NDP}(A)$ from Definition 3.4 is computable in time $O(m^2)$.* ∎

**Theorem 5.3** (time of NBM). *For any clouds $A, B \subset \mathbb{R}^2$ of $m$ unordered points, the Nested Bottleneck Metric $\mathrm{NBM}(A, B)$ is computable in time $O(m^{3.5} \log m)$.* ∎

Fig. 5 (right) illustrates the polynomial dependence of the NBM time in Theorem 5.3. Theorem 5.4 says that any $m$-point clouds $A, B \subset \mathbb{R}^2$ can be matched up to a perturbation proportional to the Nested Bottleneck Metric $d = \mathrm{NBM}$. If $d$ is small, all points of $A, B$ can be matched (up to $3\sqrt{2}d$) by rigid motion. In section 6, the experimental maximum of this approximate factor is $2.2 < 3\sqrt{2}$.

**Theorem 5.4** (point matching). *For any $m$-point clouds $A, B \subset \mathbb{R}^2$, one can find in time $O(m^{3.5} \log m)$ a rigid motion $f$ of $\mathbb{R}^2$ and a bijection $\beta : A \to B$ such that the match distance $\max_{q \in A} |f(q) - \beta(q)| \leq 3\sqrt{2}\mathrm{NBM}(A, B)$, see the comparison of distances in Fig. 5 (left).* ∎

By Theorem 5.1, perturbing every atom up to $\varepsilon$ (due to the ever-present thermal vibrations) changes NDP up to $6\varepsilon$ in the metric NBM. Conversely, by Theorem 5.4, if $\mathrm{NBM}(A, B) = \delta > 0$ is small, the atomic clouds $A, B$ can be approximately matched by rigid motion up to $3\sqrt{2}\delta$ atom-wise.

If clouds $A, B \subset \mathbb{R}^n$ consist of ordered points, one can easily *morph* (continuously transform) $A$ to $B$ by moving every $i$-th point of $A$ along a straight-line to the $i$-th point of $B$ for $i = 1, \ldots, m$. If $m$ points are unordered, there are $m!$ potential transformations, one for each permutation of $m$ points.

The brute-force association of every point $p \in A$ to its nearest neighbor $q \in B$ is justified only for fixed clouds because a rigid motion of $A$ can change a nearest neighbor of any point $p \in A$ in $B$.

Corollary 5.5 resolves this ambiguity challenge by a straight-line path connecting complete invariants in the *moduli* space $\mathrm{NDP}(\mathrm{CRS}(\mathbb{R}^2; m))$ of all realizable invariants, which effectively replaces the complicated *Cloud Rigid Space* $\mathrm{CRS}(\mathbb{R}^2; m)$ of $m$-point clouds under rigid motion in $\mathbb{R}^2$.

**Corollary 5.5** (continuous morphing). *Any clouds $A, B \subset \mathbb{R}^2$ of $m$ unordered points can be 'morphed' into each other in time $O(m^{3.5} \log m)$ by inverting a straight-line path between their complete invariants $\mathrm{NDP}(A), \mathrm{NDP}(B)$ in the space $\mathrm{NDP}(\mathrm{CRS}(\mathbb{R}^2; m))$ of realizable invariants.* ∎

## 6   A HIERARCHICAL EXPERIMENT ON 130K+ MOLECULES IN QM9

QM9 has 130K+ (130,808) molecules of up to 29 atoms with 3D coordinates obtained by quantum mechanical optimizations Ramakrishnan et al. (2014). Because many atoms are chemically identical, we compare molecules as clouds of unordered atomic centers without labels. The complete invariant NDP finalizes the hierarchy of the faster and progressively stronger invariants below.

**Definition 6.1** (invariants SRV, SDV, PDD). *Let $A \subset \mathbb{R}^n$ be a cloud of $m$ unordered points with the center of mass at $0 \in \mathbb{R}^n$. The* Sorted Radial Vector $\mathrm{SRV}(A)$ *has $m$ radial distances $|p|$ in decreasing order for all $p \in A$. The* Sorted Distance Vector $\mathrm{SDV}(A)$ *is the vector of $\frac{m(m-1)}{2}$ pairwise distances $|p-q|$ in decreasing order for distinct $p, q \in A$. For any integer $k \geq 1$ and $p \in A$, let $d_1(p) \leq \cdots \leq d_{m-1}(p)$ be Euclidean distances from $p$ to all other points $q \in A - \{p\}$ in increasing order. These distance lists become rows of the $m \times (m-1)$ matrix $D(S; k)$. Any $l > 1$ identical rows are collapsed into a single row with the* weight $l/m$. *The final matrix* $\mathrm{PDD}(S; k)$ *of unordered rows with weights is the* Pointwise Distance Distribution *Widdowson & Kurlin (2022).* ∎

For up to $m$ points, PDDs need sorting $m$ distance lists in time $O(m^2 \log m)$. Then PDDs are compared by the Earth Mover's Distance EMD Rubner et al. (2000) in time $O(m^3)$. Table 2 emphasizes that most clouds should be first distinguished by simpler and faster invariants SRV, SDV, PDD, so the complete NDP is used only in rare cases but is necessary to make important conclusions below.

Table 2: Hierarchy of invariants of $m$-point clouds $A \subset \mathbb{R}^2$: from the fastest to the complete.

| invariant time | SRV, $O(m \ln m)$ | SDV, $O(m^2)$ | PDD, $O(m^2 \ln m)$ | NDP, $O(m^2)$ |
|---|---|---|---|---|
| metric time | $L_\infty, O(m)$ | $L_\infty, O(m^2)$ | EMD, $O(m^3)$ | NBM, $O(m^{3.5} \ln m)$ |

**The ablation study** below shows the strength of complete NDP in comparison with the incomplete but faster SRV, SDV, PDD. All experiments were on AMD Ryzen 9 3950X 16-core RAM 8Gb.

We computed the pseudo-metric $L_\infty$ (max abs difference of corresponding coordinates) on SRVs of all 873,527,974 pairs of 3D atomic clouds having equal numbers of atoms in QM9, then 8,735,279 distances $L_\infty$ on SDVs of the 1% closest pairs, 87,352 EMDs on PDDs of the 1% closest pairs, and 10K distances NBM on NDPs for the final closest pairs. In this hierarchical computation, large values of $L_\infty$ (then EMD) guarantee that molecules are distant and cannot be closely matched by rigid motion. Tiny (and even zero) values of pseudo-metrics guarantee nothing because SDV and PDD can coincide for very different clouds, see Fig. 2 (right) and (Pozdnyakov et al., 2020, Fig. S4).

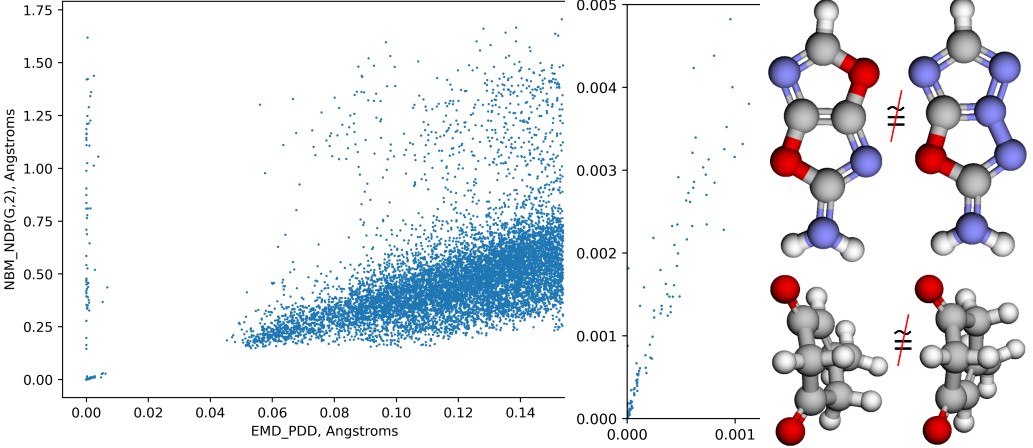

Figure 6: **Left**: each dot is a comparison of closest atomic clouds $A, B$ from QM9 by the pseudo-metric $x = \mathrm{EMD}(\mathrm{PDD}(A), \mathrm{PDD}(B))$ vs $y = \mathrm{NBM}(A, B)$ on complete invariants NDP using two base points. **Middle**: zoomed-in comparisons for small distances. **Top right**: the smallest NBM $\approx 0.15$Å for chemically different molecules is for 28141 and 130099. **Bottom right**: 70954 and 74130 are almost mirror images with EMD $\approx 0.0004$Å, well distinguished by NBM $\approx 1.619$Å.

Fig. 6 compares the new metric $y = \text{NBM}$ on complete NDPs with the pseudo-metric $x = \text{PDD}$. All pairs $A, B$ with $(x, y)$ close to the vertical axis in Fig. 6 (left) have $\text{EMD} \approx 0$ because they are almost mirror images (indistinguishable by PDD) well distinguished by higher values of NBM. Bonds in Fig. 6 (right) are shown by standard visualization, not used for invariants of clouds of points without any edges. Table 3 shows that all *chemically different* molecules (with non-equal distributions of elements) are distinguished by all invariants with the best separation by NDP.

Table 3: Closest chemically different molecules by distances in $\text{Å} = 10^{-10}$m, see Fig. 6 (right).

| invariant | metric | distance | molecule A | molecule B | composition A | composition B |
|---|---|---|---|---|---|---|
| $L_\infty$ | SRV | 0.02057 | 131923 | 5365 | H3 C4 N3 O2 | H4 C5 N2 O1 |
| $L_\infty$ | SDV | 0.05505 | 123533 | 24547 | H3 C4 N5 | H3 C5 N3 O1 |
| EMD | PDD | 0.05145 | 123533 | 24521 | H3 C4 N5 | H3 C5 N3 O1 |
| NBM | NDP | 0.14845 | 28141 | 130099 | H3 C4 N3 O2 | H3 C3 N5 O1 |

# 7 DISCUSSION: CONCLUSIONS, LIMITATIONS, AND SIGNIFICANCE

The experiments imply that mapping any molecule to (the rigid class of) its cloud of atomic centers is *injective* without losing any chemical information, so all chemical elements can be reconstructed from pure geometry. This result confirms our physical intuition that replacing atoms should perturb geometry at least slightly, which was impossible to establish without complete and Lipschitz continuous invariants. Hence all molecules of the same number $m$ of atoms live at different locations in the common *Cloud Rigid Space* $\text{CRS}(\mathbb{R}^3; m)$ of SE(3)-classes of clouds of $m$ unordered points.

Fig. 7 shows two simplest projections of the atomic clouds from QM9 considered as a finite sample from $\bigcup_{m=3}^{29} \text{CRS}(\mathbb{R}^3; m)$, see the familiar molecules such as H2O (water). Any small region on such a map can be zoomed in and displayed in other coordinates from the hierarchy in Table 2.

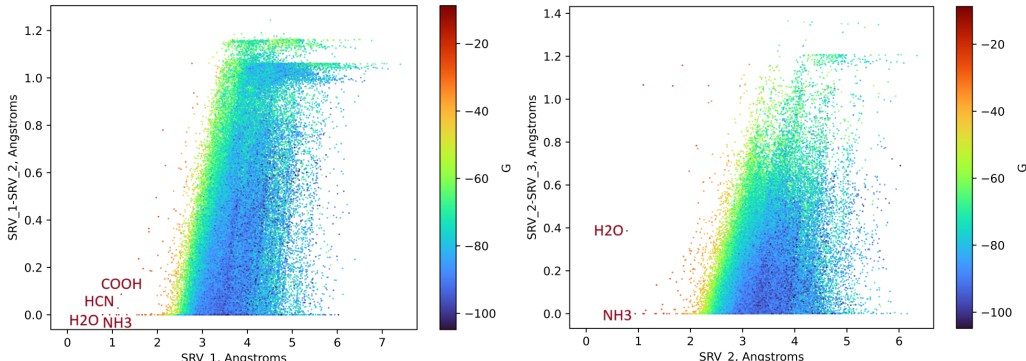

Figure 7: **Left**: every dot represents an atomic cloud with the invariant coordinates $x = \text{SRV}_1$, $y = \text{SRV}_1 - \text{SRV}_2$, all in Angstroms, where $1\text{Å} = 10^{-10}m \approx$ the smallest interatomic distance. **Right**: the subset of molecules with $\text{SRV}_1 = \text{SRV}_2$ (two equidistant atoms from the center of mass) is projected to $x = \text{SRV}_2$, $y = \text{SRV}_2 - \text{SRV}_3$. The color is by the free energy $G$ from QM9.

Problem 1.1 was stated for unordered clouds under rigid motion but was also solved for *isometry* and compositions of these equivalences with uniform scaling in $\mathbb{R}^2$, also for dimensions $n > 2$ in the appendices. For $m = 4$ points, plane quadrilaterals were previously classified in discrete classes in Fig. 1 (right), while appendix B shows the first continuous maps of the invariant space $\text{CRS}(\mathbb{R}^2; 4)$. Conditions 1.1(d,e,f) enable a generation of real clouds in $\text{CRS}(\mathbb{R}^n; m)$ from their invariants.

We compared atomic clouds of the same size in QM9 because atoms are real physical objects and cannot be considered outliers or noise. In other applications, for clouds with different numbers of points, we can replace the bottleneck distance BD in Definition 4.2 with any metric between fixed clouds of different sizes, e.g. the Hausdorff distance, to get a metric on PRs. Then we can compare NDPs of any clouds as weighted distributions by EMD. The limitation is the proof of Theorem 5.4 in dimension $n = 2$, though the experiments on QM9 indicate the Lipschitz continuity of $\text{NDP}^{-1}$ in $\mathbb{R}^3$. All other conditions in Problem 1.1 are proved in the appendices for any $n \geq 2$.

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

**Introduction to appendices**. The key contribution is a theoretically justified solution to Problem 1.1. The experiments on the QM9 database of 130K+ molecules are considered complimentary.

Example 4.5 and its extension in Example B.2 prove that infinitely many pairs of non-isometric clouds $C^+ \not\cong C^-$ (depending on 4 free parameters and having the same 6 pairwise distances) are distinguished by the new invariants. This result is impossible to justify by any finite experiment. Example 4.5 demonstrated the non-zero distances between the complete invariants of $C^{\pm}$ in Fig. 4.

The completeness and bi-Lipschitz continuity of the proposed invariants enabled the new experiments on 130K+ real molecules in section 6, which were not previously possible because all past invariants did not satisfy all conditions of Problem 1.1, especially the realizability condition that provides geographic-style maps on cloud spaces.

The key contribution is a solution to Problem 1.1justified by Theorem C.9 and Lemmas 3.3, 5.1, 5.2, 5.3, which are extended to any Euclidean space $\mathbb{R}^n$ in the appendices. Theorem 3.3 enables a visualization of cloud spaces, which were unknown even for $m = 4$ unordered points in $\mathbb{R}^2$.

• The *Cloud Isometry Space* $\mathrm{CIS}(\mathbb{R}^n; m)$ of clouds of $m$ unordered points under isometry in $\mathbb{R}^n$.

• The *Cloud Rigid Space* $\mathrm{CRS}(\mathbb{R}^n; m)$ of clouds of $m$ unordered points under rigid motion in $\mathbb{R}^n$.

• The *Cloud Similarity Space* $\mathrm{CSS}(\mathbb{R}^n; m)$ of clouds of $m$ unordered points under *geometric similarity*, which is a composition of isometry and uniform scaling in $\mathbb{R}^n$.

• The *Cloud Dilation Space* $\mathrm{DCS}(\mathbb{R}^n; m)$ of clouds of $m$ unordered points under orientation-preserving geometric similarity (rigid motion and uniform scaling) in $\mathbb{R}^n$.

Here is a summary of the supplementary materials.

• Appendix A extends section 6 with more details of new invariants and metrics computed on the QM9 database and compared with past pseudo-metrics.

• Appendix B discusses parametrization of $\mathrm{CSS}(\mathbb{R}^2; m)$ and includes Examples 4.5 and B.2 computing the new invariants NDP in detail for infinitely many 4-point clouds from Example 4.5.

• Appendices C, D, E prove all theoretical results from sections 3, 4, 5, respectively.

• The zip folder with supplementary materials includes the code for computing all invariants and metrics as well as tables with all coordinates of colorful maps of QM9 and distances.

## A    EXTRA DETAILS OF EXPERIMENTS IN SECTION 6

The maps of QM9 in Fig. 8 are based on eigenvalues and too dense without clear separation. Even if we zoom in, these incomplete invariants will not separate molecules because 3D clouds have at most 3 eigenvalues. The complete invariants NDP contain much more geometric information. Fig. 9 and 10 show that distances on stornger invariants have larger values and hence better separate molecules, though all these distances have the same Lipschitz constant 2.

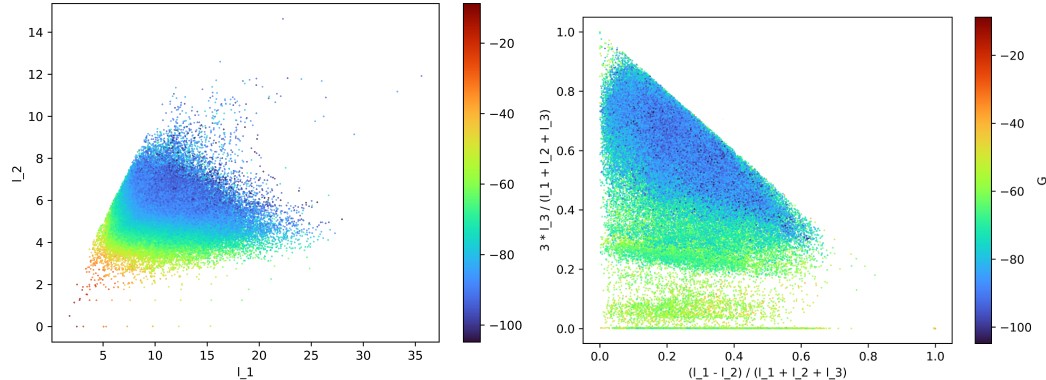

Figure 8: **Left**: each dot represents one QM9 molecule whose atomic cloud has two largest roots $l_1 \geq l_2$ of eigenvalues (moments of inertia Nemec (2022) or elongations in principal directions) in Angstroms ($1\mathring{A} = 10^{-10}m \approx$ smallest interatomic distance). The color represents the free energy $G$ characterizing molecular stability. **Right**: each dot represents one QM9 molecule whose atomic cloud has coordinates $x, y$ expressed via the roots $l_1 \geq l_2 \geq l_3 \geq 0$ of three eigenvalues.

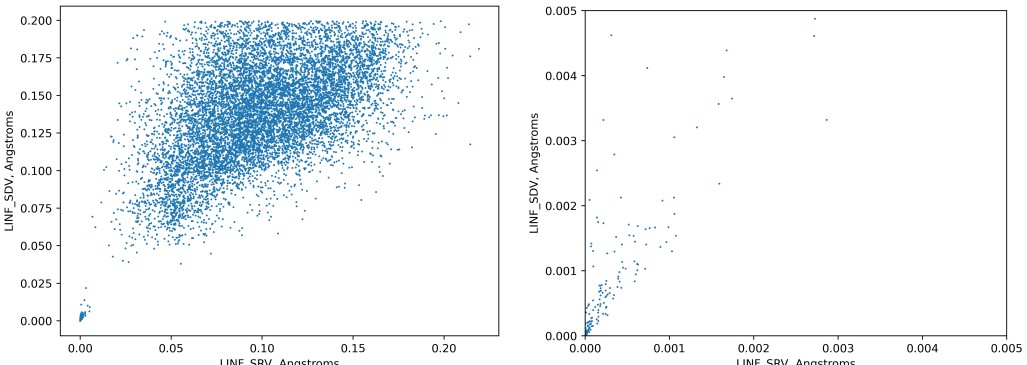

Figure 9: **Left**: each dot is a comparison of closest atomic clouds $A, B$ from QM9 by the distances $L_\infty$ on SRV vs $L_\infty$ on SDV. **Right**: zoomed-in comparisons for very small distances.

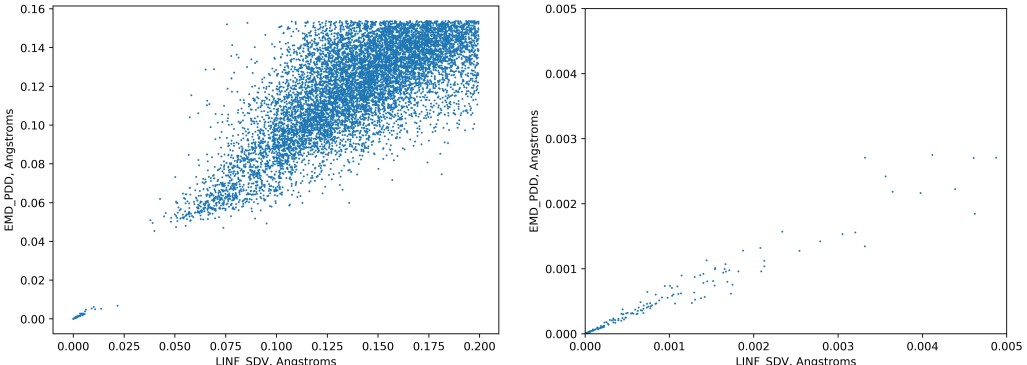

Figure 10: **Left**: each dot is a comparison of closest atomic clouds $A, B$ from QM9 by the distances $L_\infty$ on SDV vs EMD on PDD. **Right**: zoomed-in comparisons for very small distances.

# B    MAPS OF CLOUD SPACES AND EXPLICIT COMPUTATIONS OF INVARIANTS

This section explains how cloud spaces can be visualized by considering the previously known and new types of 4-point clouds (quads) in $\mathbb{R}^2$. This geographic-style approach extends to any number $m$ of points in $\mathbb{R}^n$.

For any cloud $A \subset \mathbb{R}^n$, the center $O(A) = 0 \in \mathbb{R}^n$ is the origin. For $n = 2$, let $p\{1\}$ consist of a single point $p_1 \in A$ with $|p_1| = R(A) = R$. We can fix $p_1 = (R, 0)$ in $\mathbb{R}^2$. Then all points $p_2, \ldots, p_m$ are in the disk $D = \{x^2 + y^2 \leq R^2\}$. Since $\sum\limits_{i=2}^{m} p_i = -p_1 = (-R, 0)$, $p_m$ is determined from $p_2, \ldots, p_{m-1} \in D$ that satisfy only one equation

$$R^2 \geq |p_m|^2 = |(R, 0)^T + \sum_{i=2}^{m-1} p_i|^2 = (R + x)^2 + y^2,$$

where $(x, y)$ are the coordinates of $s = \sum\limits_{i=2}^{m-1} p_i$. The domain of $s$ is the intersection $J = D \cap \{(R + x)^2 + y^2 \leq R^2\}$.

For $m = 3$, we have $s = (x, y) = p_2$. The symmetry $p_2 \leftrightarrow p_3$ allows us to choose any $p_2$ in the left half (yellow) $D_3$ of the intersection $J$ in Fig. 11 (left). Then the Rigid Cloud Space $\mathrm{CRS}(\mathbb{R}^n; 3)$ is parametrized by any radius $R > 0$ and $p_2 \in D_3$. All equilateral triangles have $p_2 = (-\frac{1}{2}R, \pm\frac{\sqrt{3}}{2}R)$. All isosceles triangles have $p_2$ in the boundary $\partial D_3$ whose points should be identified under $(x, y) \mapsto (x, -y)$. All $p_2 = (x, 0)$ with $-R \leq x \leq -\frac{1}{2}R$ represent degenerate triangles with the vertices $(R, 0)$, $(x, 0)$, $(-R - x, 0)$ in the same line.

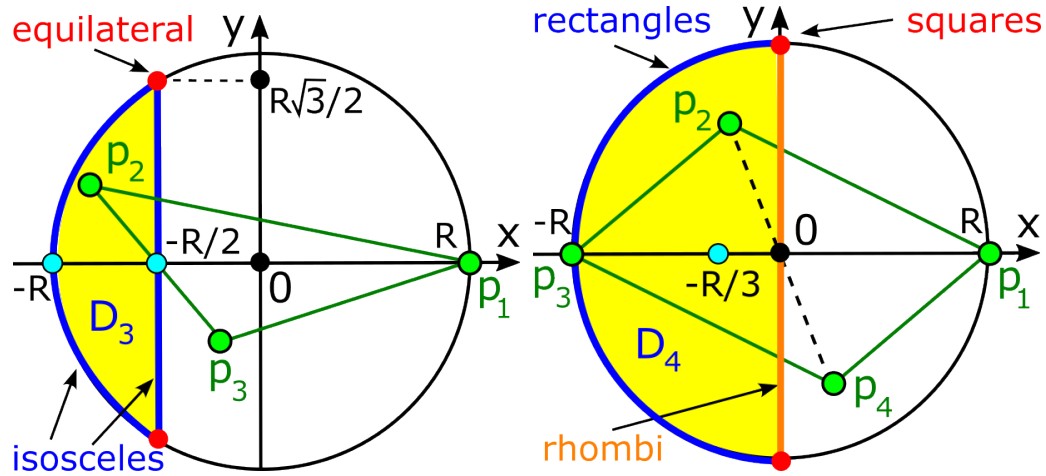

Figure 11: The spaces in yellow for triangles ($D_3$) and parallelograms ($D_4$) under rigid motion and uniform scaling in $\mathbb{R}^2$.

For $m = 4$, we can choose $s = p_2 + p_3 \in J$, then any $p_3$ in the disk with the radius $R$ and center $s$ so that $|p_2| = |p_3 - s| \leq R$. For any parallelogram in $\mathbb{R}^2$, its vertex cloud $A$ has a longest diagonal between (say) $p_1, p_3$ that should be at $(\pm R, 0)$. All possible $s = p_2 + (-R, 0) \in J$ mean that $p_2$ can be anywhere in $D$. Due to the symmetry $p_2 \leftrightarrow p_4$, the left half $D_4$ of $D$ in Fig. 11 (right) is the subspace of all parallelograms in $\mathrm{DCS}(\mathbb{R}^2; 4) = \mathrm{CRS}(\mathbb{R}^2; 4)/\text{scaling}$.

Similarly for $m > 4$, $n \geq 2$, we can sequentially sample points $p_2, \ldots, p_{m-1}$ from allowed disks (high-dimensional for $n > 2$) to get a unique representation of $A$ under rigid motion. The symmetry $f : (x, y) \mapsto (x, -y)$ on $D$ identifies mirror images of $A$. $\mathrm{CIS}(\mathbb{R}^n; m)$ is the quotient of $\mathrm{CRS}(\mathbb{R}^n; m)$ under $(x, y) \sim (x, -y)$, take the upper halves of $D_3, D_4$ for triangles and parallelograms, respectively.

We expand Fig. 11 above to illustrate severak important subspaces in the Isometry Cloud Space $\mathrm{CIS}(\mathbb{R}^2; m)$ and the Similarity Cloud Space $\mathrm{CSS}(\mathbb{R}^2; m)$ for $m = 3, 4$. For simplicity, we call all clouds of 3 and 4 unordered points triangles and quadrilaterals, respectively.

However, all these polygons are considered equivalent when we re-order their vertices. If all $m$ points are ordered, parametrizations of the resulting shape spaces were studied in geometry

Kapovich & Millson (1996) and shape theory Kendall et al. (2009). We focus on the much harder quotient spaces of $m$ unordered points.

Theorem C.7 explicitly describes all realizable Point-based Representations. Though the same point cloud $A \subset \mathbb{R}$ can have many $\mathrm{PR}(A; p\{n-1\})$ depending on a base sequence $p\{n-1\} \subset A$, we can easily sample any of them and always reconstruct $A$, while random sampling distance-based invariants doesn't guarantee the existence of $A$ because of extra relations between inter-point distances.

Though $\mathrm{PR}(A; p\{n-1\})$ consists of scalar products $q \cdot p_i$ with basis vectors $p_1, \ldots, p_n$, it is easier to visualize the isometry spaces by directly using some points $q \in A$ as parameters instead of their projections.

**Case** $m = 3$ of triangles is the same in all dimensions $n \geq 2$. We consider $\mathbb{R}^2$ for simplicity. Fig. 11 (left) showed the Dilation Cloud Space $\mathrm{DCS}(\mathbb{R}^2; 3)$ of triangles $A$ modulo rigid motion and uniform scaling in $\mathbb{R}^2$. We assume that the center of mass is at the origin: $C(A) = 0$ in $\mathbb{R}^2$. After the radius $R = 1$ of $A$ is fixed up to scaling, we also fix the first vertex at $p_1 = (R, 0)$. Then $\mathrm{DCS}(\mathbb{R}^2; 3)$ is parametrized by the second vertex $p_2 \in D_3$, because the vertex $p_3$ is uniquely determined by $p_1 + p_2 + p_3 = 0$.

The blue boundary of $\mathrm{DCS}(\mathbb{R}^2; 3)$ consists of points $p_2$ that define isosceles triangles. The vertical part of the blue boundary in Fig. 12 (left) represents all isosceles triangles with a unique angle (not equal to two equal ones) less than $60°$. The round part of the blue boundary in Fig. 12 (right) represents all isosceles triangles with a unique angle greater than $60°$. These boundary parts meet at the red points $(-\frac{R}{2}, \pm\frac{\sqrt{3}}{2}R)$ representing all equilateral triangles.

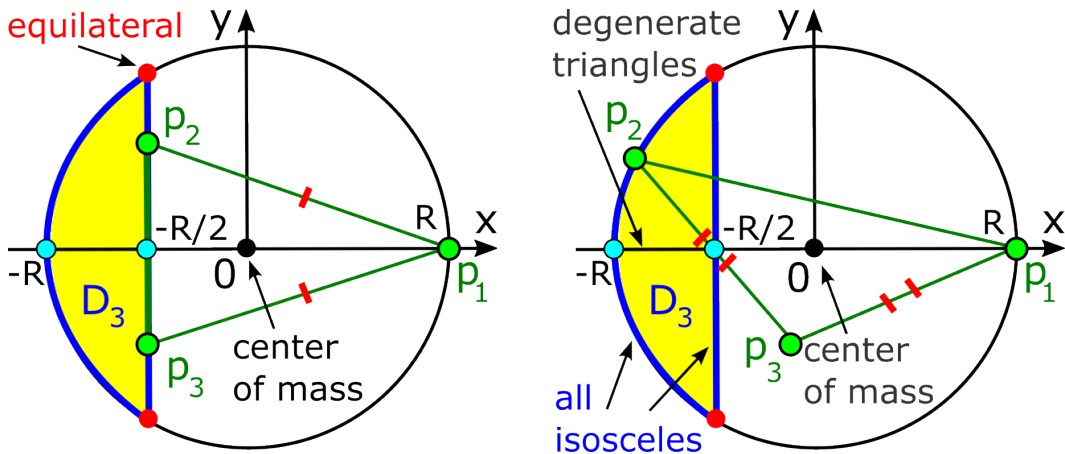

Figure 12: The (blue) subspace of all isosceles triangles in $\mathrm{CSS}(\mathbb{R}^2; 3)$. **Left**: isosceles triangles with $|p_1 - p_2| = |p_1 - p_3|$. **Right**: isosceles triangles with $|p_3 - p_1| = |p_3 - p_2|$.

If $p_2 = (x, 0)$ for $-R \leq x \leq -\frac{R}{2}$, then $p_3 = (-R - x, 0)$, so the triangle generates to three points in the line. In the yellow space $D_3 = \mathrm{CSS}^o(\mathbb{R}^2; 3)$, the mirror reflection $(x, y) \mapsto (x, -y)$ maps every isosceles triangle to itself, more exactly, to an equivalent triangle under rigid motion. Hence all points of the blue boundary of $D_3$ should be identified under $(x, y) \mapsto (x, -y)$. Then the space $D_3$ of all triangles (including degenerate ones) under rigid motion and uniform scaling can be visualized as a topological sphere $S^2$ whose the northern and southern hemispheres are obtained from the upper and lower halves of $D_3$.

**Case** $m = 4$ of quadrilaterals in $\mathbb{R}^2$. Fix the center of mass $O(A) = 0 \in \mathbb{R}^2$ at the origin, the radius $R(A) = R$, and a most distant (from 0) point $p_1$ at $(R, 0)$. The other vertices $p_2, p_3, p_4$ belong to the disk $D = \{x^2 + y^2 \leq R^2\}$ and have the shifted center of mass $\frac{p_2 + p_3 + p_4}{3} = (-\frac{R}{3}, 0)$. Hence, for a fixed radius $R$, the space $\mathrm{CSS}(\mathbb{R}^2; 4)$ is 4-dimensional.

The subspace of parallelograms in $\text{CSS}(\mathbb{R}^2; 4)$ is 2-dimensional. For any parallelogram $A$, its other most distant vertex is $p_3 = (-R, 0)$ opposite to $p_1$ with respect to 0. Then $p_2 + p_4 = 0$ and the symmetry $p_2 \leftrightarrow p_4$ allows us to consider only $p_2$ in the yellow half-disk $D_4$, which uniquely determines its symmetric image $p_4$ in Fig. 11 (left).

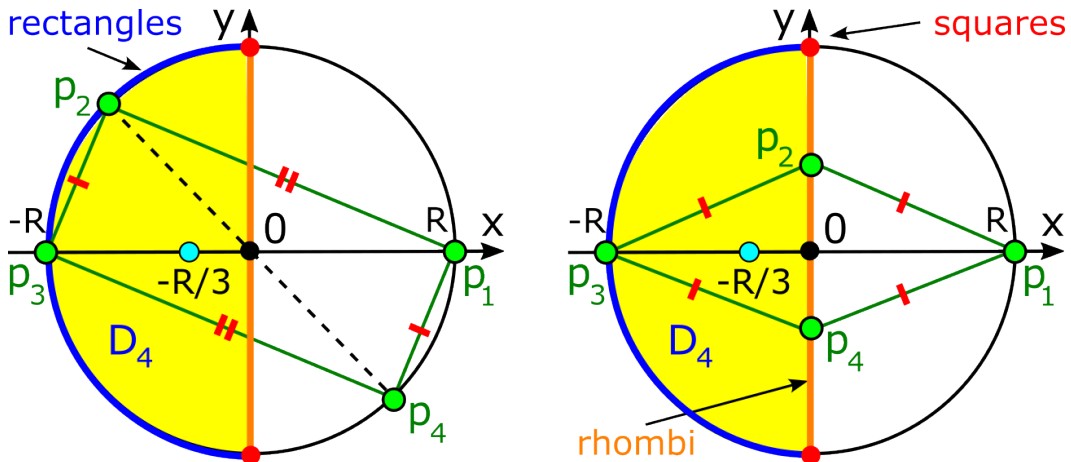

Figure 13: The (yellow) subspace $D_4$ of all parallelograms with $p_1 = (R, 0)$ and $p_3 = (-R, 0)$ in $\text{CSS}(\mathbb{R}^2; 4)$. **Left**: the (blue) subspace of rectangles. **Right**: the (orange) subspace of rhombi.

The round (blue) boundary of $D_4$ in Fig. 13 (left) represents all rectangles inscribed in the circle $x^2 + y^2 = R^2$. The vertical (orange) boundary of $D_4$ in Fig. 13 (right) represents all rhombi with equal sides. The reflection $(x, y) \mapsto (x, -y)$ maps any parallelogram to its mirror image and preserves the equivalence class (up to rigid motion) of any rectangle or rhombus, which are mirror-symmetric. Hence all points on the boundary of $D_4$ should be identified under $(x, y) \mapsto (x, -y)$. The resulting quotient is a topological sphere $S^2$ as $D_3$ for all triangles, unsurprisingly because a parallelogram can be considered as a double triangle.

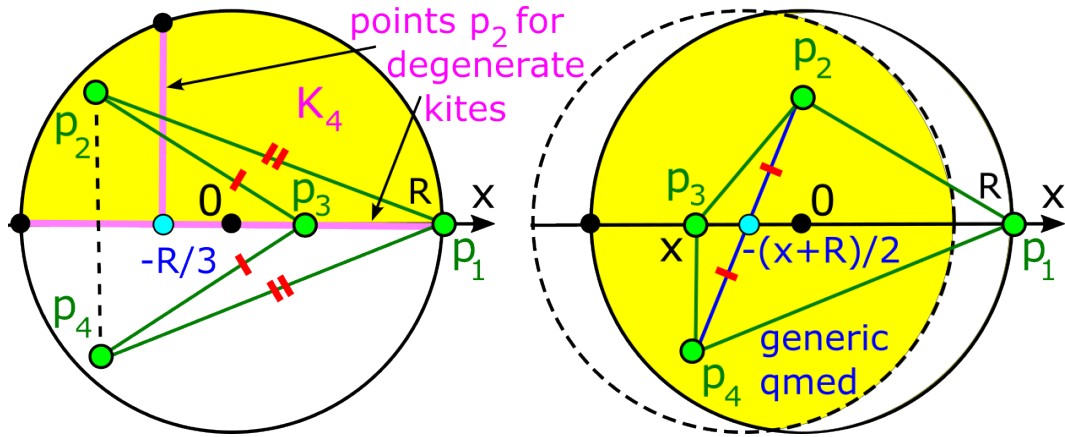

Figure 14: **Left**: the (yellow) subspace of kites in $\text{CSS}(\mathbb{R}^2; 4)$ parametrized by $p_2 \in K_4$. **Right**: the subspace of qmeds is parametrized by $x \in [-R, R]$ and $p_2$ in the yellow region.

Another interesting case is when one of the vertices $p_3 = (x, 0)$ belongs to the $x$-axis for $x \in [-R, R]$. Then the (horizontal line passing through) diagonal joining $p_1, p_3$ intersects another diagonal at its mid-point $\frac{p_2 + p_4}{2} = (x_{2,4}, 0)$ for $x_{2,4} = -\frac{x+R}{2} \in [-R, 0]$. The resulting cloud $A$ can be called a *quadrilateral with a median diagonal*, briefly *qmed*. If a qmed $A$ is also symmetric with

respect to its median diagonal, the $A$ has two pairs of equal sides and is often called a *kite*, see the kite $K$ in Fig. 2 (right).

Since any kite is mirror-symmetric, the points $p_2 = (x, y)$ and $p_4 = (x, -y)$ represents the same kite up to rigid motion. Hence the (yellow) subspace of all kites in $\text{CSS}(\mathbb{R}^2; 4)$ is the upper half $K_4$ of the disk $D$ in Fig. 14 (left). For points $p_2$ in the vertical line $x = -\frac{R}{3}$, we get a degenerate kites whose vertices $p_2, p_3, p_4$ are in the same straight line. If $p_2 = (x, 0)$, the kite degenerates even further to the case of identical vertices $p_2 = p_4$.

So the subspace $K_4$ of kites in $\text{CSS}(\mathbb{R}^2; 4)$ is 2-dimensional, while the larger subspace of qmeds is 3-dimensional, parametrized by $x \in [-R, R]$ and a point $p_2$ that can take any position in the intersection of the disk $D = \{x^2 + y^2 \leq R^2\}$ and its symmetric image with respect to the diagonal mid-point $(x_{2,4}, 0) = (-\frac{x+R}{2}, 0)$.

The full space $\text{CSS}(\mathbb{R}^2; 4)$ is parametrized by the sum $s = p_2 + p_3$ in the intersection $J = D \cap \{(R + x)^2 + y^2 \leq R^2\}$ and then taking $p_2$ in the disk with the radius $R$ and center $s$ to guarantee that $|p_3| = |p_2 - s| \leq R$.

**Case $m = 4$ of tetrahedra in $\mathbb{R}^3$.** In $\mathbb{R}^3$, we similarly fix the center of mass at the origin and the most distant points $p_1$ at $(R, 0, 0)$. The second most distant point $p_2$ (if not in the line through $0$ and $p_1$) forms a base sequence $p_1, p_2$ and can be fixed at $(x, y, 0)$ with $x^2 + y^2 \leq R^2$, which determines the mid-point $p_{3,4} \frac{p_3+p_4}{2} = (-\frac{x+R}{2}, -\frac{y}{2}, 0)$. Due to the symmetry $p_3 \leftrightarrow p_4$ around $p_{3,4}$, it remains to choose $p_3$ in the upper half ball with the center $p_{3,4}$ and radius $\sqrt{x^2 + y^2}$.

The clouds in Example B.1 are instances of $C^\pm$ from Example **??**: $K = C^+, T = C^-$ for $4a = b = c = 4d = 2\sqrt{2}$ and are easy enough to write their NDPs below.

**Example B.1** (4-point clouds $T, K$ in Fig. 2)**.** *Both clouds $T, K \subset \mathbb{R}^2$ in Fig. 2 have the center of mass at the origin.*

*(T) The cloud $T$ has the points $p_1 = (2, 1)$, $p_2 = (-2, 1)$, $p_3 = (-4, -1)$, $p_4 = (4, -1)$. For the basis point $p_1 = (2, 1)$ with $|p_1|^2 = 5$ and orthogonal vector $p_1^\perp = (-1, 2) \perp p_1$ from Lemma C.1, the point-based representation is* $\text{PR}(T; p_1) = \left[ 5, \begin{pmatrix} -3 & -9 & 7 \\ 4 & 2 & -6 \end{pmatrix} \right]$.

*For the second point $p_2 = (-2, 1)$ with $|p_2|^2 = 5$, $p_2^\perp = (-1, -2)$, we have $\text{PR}(T; p_2) = \left[ 5, \begin{pmatrix} -3 & 7 & -9 \\ -4 & 6 & -2 \end{pmatrix} \right]$, which differs from $\text{PR}(T; p_1)$ by the sign of the last row (up to a permutation of columns). The symmetries under $p_1 \leftrightarrow p_2$ (above) and $p_3 \leftrightarrow p_4$ (below) are explained by the reflection $(x, y) \mapsto (-x, y)$ mapping $T$ to itself.*

*For $p_3 = (-4, -1)$ with $|p_3|^2 = 17$, $p_3^\perp = (1, -4)$, we have $\text{PR}(T; p_3) = \left[ 17, \begin{pmatrix} -9 & 7 & -15 \\ -2 & -6 & 8 \end{pmatrix} \right]$.*

*For the fourth point $p_4 = (4, -1)$ with $|p_4|^2 = 17$, $p_4^\perp = (1, 4)$, we have $\text{PR}(T; p_4) = \left[ 17, \begin{pmatrix} 7 & -9 & -15 \\ 6 & 2 & -8 \end{pmatrix} \right]$.*

*So $\text{NDP}(T)$ is the unordered set of the four PRs above.*

*(K) The cloud $K$ has the points $p_1 = (5, 0)$, $p_2 = (-1, 2)$, $p_3 = (-3, 0)$, $p_4 = (-1, -2)$.*

*For the basis point $p_1 = (5, 0)$ with $|p_1|^2 = 25$ and $p_1^\perp = (0, 5) \perp p_1$, the point-based representation is* $\text{PR}(K; p_1) = \left[ 25, \begin{pmatrix} -5 & -15 & -5 \\ 10 & 0 & -10 \end{pmatrix} \right]$.

*For the second point $p_2 = (-1, 2)$ with $|p_2|^2 = 5$ and $p_2^\perp = (-2, -1)$, we have $\text{PR}(K; p_2) = \left[ 5, \begin{pmatrix} -5 & 3 & 1 \\ -10 & 6 & 4 \end{pmatrix} \right]$.*

*For the third point $p_3 = (-3, 0)$ with $|p_3|^2 = 9$ and $p_3^\perp = (0, -3)$, we have $\mathrm{PR}(K; p_3) =$*

$$\left[ 9, \begin{pmatrix} -15 & 3 & 3 \\ 0 & -6 & 6 \end{pmatrix} \right].$$

*For the point $p_4 = (-1, -2)$ with $|p_4|^2 = 5$ and $p_4^\perp = (2, -1)$, we have $\mathrm{PR}(K; p_4) =$*

$$\left[ 5, \begin{pmatrix} -5 & 1 & 3 \\ 10 & -4 & -6 \end{pmatrix} \right].$$

*So $\mathrm{NDP}(K)$ is the unordered set of the four PRs above.*

*$T \not\cong K$ are distinguished by (unordered) squared distances to their centers: $5, 5, 17, 17$ for $T$, and $25, 5, 9, 5$ for $K$.* ∎

Example B.2 finishes the computations of the Nested Distributed Projection (NDP) for the 4-point clouds $C^\pm \subset \mathbb{R}^2$ in Fig. 2, which we started in Example 4.5.

**Example B.2** (4-point clouds $C^\pm$ in Fig. 2). *In $\mathbb{R}^2$, consider the 4-point clouds $C^\pm = \{p_1, p_2, p_3, p_4^\pm\}$, where $p_1 = (4a, 0)$, $p_2 = (b, c)$, $p_3 = -p_2 = (-b, -c)$, $p_4^+ = (0, 4d)$, and $p_4^- = (0, -4d)$ for parameters $a, b, c, d \geq 0$.*

*After shifting the center $O(C^+) = (a, d)$ to the origin $(0, 0)$, the points of $C^+$ become $p_1^+ = (3a, -d)$, $p_2^+ = (b - a, c - d)$, $p_3^+ = (-a - b, -c - d)$, $\hat{p}_4^+ = (-a, 3d)$.*

*Each matrix $\mathrm{SD}(C^+; p)$ is one squared distance $|p|^2$.*

$$\mathrm{SD}(C^+; p_1^+) = 9a^2 + d^2,$$
$$\mathrm{SD}(C^+; p_2^+) = (a - b)^2 + (c - d)^2,$$
$$\mathrm{SD}(C^+; p_3^+) = (a + b)^2 + (c + d)^2,$$
$$\mathrm{SD}(C^+; \hat{p}_4^+) = a^2 + 9d^2.$$

*For the second cloud $C^-$, after shifting the center $O(C^-) = (a, -d)$ to the origin $(0, 0)$, the points become $p_1^- = (3a, d)$, $p_2^- = (b - a, d + c)$, $p_3^- = (-a - b, d - c)$, $\hat{p}_4^- = (-a, -3d)$.*

*Hence $C^-$ has the following squared distances to its center:*

$$\mathrm{SD}(C^-; p_1^-) = 9a^2 + d^2,$$
$$\mathrm{SD}(C^-; p_2^-) = (a - b)^2 + (c + d)^2,$$
$$\mathrm{SD}(C^-; p_3^-) = (a + b)^2 + (c - d)^2,$$
$$\mathrm{SD}(C^-; \hat{p}_4^-) = a^2 + 9d^2.$$

*The (unordered) collections of squared distances above differ unless at least one of $a, b, c, d$ is zero. Indeed, the squared distances $9a^2 + d^2$ and $a^2 + 9d^2$ are shared by $C^\pm$ but $\mathrm{SD}(C^+; p_2^+)$ is unique and cannot equal $\mathrm{SD}(C^-; p_2^-)$ or $\mathrm{SD}(C^-; p_3^-)$. Indeed, if all $a, b, c, d \neq 0$, then*

$$(a - b)^2 + (c - d)^2 \neq (a - b)^2 + (c + d)^2 \text{ or } cd \neq 0,$$
$$(a - b)^2 + (c - d)^2 \neq (a + b)^2 + (c - d)^2 \text{ or } ab \neq 0.$$

*If $d = 0$, then $p_4^\pm = (0, 0)$, so the clouds $C^\pm$ are identical.*

*If $a = 0$, then $p_1 = (0, 0)$ and $C^\pm$ are related by the $180°$ rotation around the origin: $(x, y) \mapsto (-x, -y)$.*

*If $b = 0$ or $c = 0$, then $C^\pm$ are related by the reflection $(x, y) \mapsto (x, -y)$, so distances cannot distinguish these mirror images. We compute $\mathrm{NDP}(C^\pm)$ below to distinguish all non-rigidly equivalent $C^+ \not\cong C^-$, see Fig. 4.*

*For the basis point $p_1^+$, the matrix $\mathrm{SD}(C^+; p_1^+) = 9a^2 + d^2$ is the single squared distance. Lemma C.1 gives the orthogonal vector $q_1^+ = (d, 3a) \perp p_1^+$. $M(C^+; p_1^+)$ consists of the 3 unordered columns*

$$\begin{pmatrix} p_2^+ \cdot p_1^+ \\ p_2^+ \cdot q_1^+ \end{pmatrix} = \begin{pmatrix} 3a(b-a) + d(d-c) \\ d(b-a) + 3a(c-d) \end{pmatrix},$$

$$\begin{pmatrix} p_3^+ \cdot p_1^+ \\ p_3^+ \cdot q_1^+ \end{pmatrix} = \begin{pmatrix} -3a(a+b) + d(c+d) \\ -d(a+b) - 3a(c+d) \end{pmatrix},$$

$$\begin{pmatrix} \hat{p}_4^+ \cdot p_1^+ \\ \hat{p}_4^+ \cdot q_1^+ \end{pmatrix} = \begin{pmatrix} -3(a^2 + d^2) \\ 8ad \end{pmatrix}.$$ *The second point* $p_2^+ = (b-a, c-d)$ *has the orthogonal vector* $q_2^+ = (d-c, b-a) \perp p_2^+$, $\mathrm{SD}(C^+; p_2^+) = (a-b)^2 + (c-d)^2$ *and* $M(C^+; p_2^+)$ *consisting of the 3 unordered columns*

$$\begin{pmatrix} p_1^+ \cdot p_2^+ \\ p_1^+ \cdot q_2^+ \end{pmatrix} = \begin{pmatrix} 3a(b-a) + d(d-c) \\ 3a(d-c) + d(a-b) \end{pmatrix},$$

$$\begin{pmatrix} p_3^+ \cdot p_2^+ \\ p_3^+ \cdot q_2^+ \end{pmatrix} = \begin{pmatrix} a^2 - b^2 - c^2 + d^2 \\ 2(ac - bd) \end{pmatrix},$$

$$\begin{pmatrix} \hat{p}_4^+ \cdot p_2^+ \\ \hat{p}_4^+ \cdot q_2^+ \end{pmatrix} = \begin{pmatrix} a(a-b) + 3d(c-d) \\ a(c-d) + 3d(b-a) \end{pmatrix}.$$ *The third point* $p_3^+ = (-a-b, -c-d)$ *has the vector* $q_3^+ = (c+d, -a-b) \perp p_3^+$, $\mathrm{SD}(C^+; p_3^+) = (a+b)^2 + (c+d)^2$ *and* $M(C^+; p_3^+)$ *consisting of the 3 unordered columns*

$$\begin{pmatrix} p_1^+ \cdot p_3^+ \\ p_1^+ \cdot q_3^+ \end{pmatrix} = \begin{pmatrix} -3a(a+b) + d(c+d) \\ 3a(c+d) + d(a+b) \end{pmatrix},$$

$$\begin{pmatrix} p_2^+ \cdot p_3^+ \\ p_2^+ \cdot q_3^+ \end{pmatrix} = \begin{pmatrix} a^2 - b^2 - c^2 + d^2 \\ 2(bd - ac) \end{pmatrix},$$

$$\begin{pmatrix} \hat{p}_4^+ \cdot p_3^+ \\ \hat{p}_4^+ \cdot q_3^+ \end{pmatrix} = \begin{pmatrix} a(a+b) - 3d(c+d) \\ -a(c+d) - 3d(a+b) \end{pmatrix}.$$ *The fourth point* $\hat{p}_4^+ = (-a, 3d)$ *has the vector* $q_4^+ = (-3d, -a) \perp p_4^+$, $\mathrm{SD}(C^+; \hat{p}_4^+) = a^2 + 9d^2$, $M(C^+; \hat{p}_4^+)$ *has the columns*

$$\begin{pmatrix} p_1^+ \cdot \hat{p}_4^+ \\ p_1^+ \cdot q_4^+ \end{pmatrix} = \begin{pmatrix} -3(a^2 + d^2) \\ -8ad \end{pmatrix},$$

$$\begin{pmatrix} p_2^+ \cdot \hat{p}_4^+ \\ p_2^+ \cdot q_4^+ \end{pmatrix} = \begin{pmatrix} a(a-b) + 3d(c-d) \\ 3d(a-b) + a(d-c) \end{pmatrix},$$

$$\begin{pmatrix} p_3^+ \cdot \hat{p}_4^+ \\ p_3^+ \cdot q_4^+ \end{pmatrix} = \begin{pmatrix} a(a+b) - 3d(c+d) \\ 3d(a+b) + a(c+d) \end{pmatrix}.$$ *The Nested Distributed Projection* $\mathrm{NDP}(C^+)$ *consists of the four pairs (of a squared distance and* $2 \times 3$ *matrix) above.*

*For* $C^-$, *after shifting the center* $O(C^-) = (a, -d)$ *to the origin* $(0, 0)$, *the points of* $C^-$ *become* $p_1^- = (3a, d)$, $p_2^- = (b-a, d+c)$, $p_3^- = (-a-b, d-c)$, $\hat{p}_4^- = (-a, -3d)$. *The first point* $p_1^-$ *has the vector* $q_1^- = (-d, 3a) \perp p_1^-$, $\mathrm{SD}(C^-; p_1^-) = 9a^2 + d^2$, $M(C^-; p_1^-)$ *has the columns*

$$\begin{pmatrix} p_2^- \cdot p_1^- \\ p_2^- \cdot q_1^- \end{pmatrix} = \begin{pmatrix} 3a(b-a) + d(d+c) \\ d(a-b) + 3a(d+c) \end{pmatrix},$$

$$\begin{pmatrix} p_3^- \cdot p_1^- \\ p_3^- \cdot q_1^- \end{pmatrix} = \begin{pmatrix} -3a(b+a) + d(d-c) \\ d(b+a) + 3a(d-c) \end{pmatrix},$$

$$\begin{pmatrix} \hat{p}_4^- \cdot p_1^- \\ \hat{p}_4^- \cdot q_1^- \end{pmatrix} = \begin{pmatrix} -3(a^2 + d^2) \\ -8ad \end{pmatrix}.$$ *The second point* $p_2^- = (b-a, d+c)$ *has the vector* $q_2^- = (-d-c, b-a) \perp p_2^-$, $\mathrm{SD}(C^-; p_2^-) = (a-b)^2 + (c+d)^2$, $M(C^-; p_2^-)$ *of*

$$\begin{pmatrix} p_1^- \cdot p_2^- \\ p_1^- \cdot q_2^- \end{pmatrix} = \begin{pmatrix} 3a(b-a) + d(d+c) \\ -3a(c+d) + d(b-a) \end{pmatrix},$$

$$\begin{pmatrix} p_3^- \cdot p_2^- \\ p_3^- \cdot q_2^- \end{pmatrix} = \begin{pmatrix} a^2 - b^2 - c^2 + d^2 \\ 2(ac + bd) \end{pmatrix},$$

$$\begin{pmatrix} \hat{p}_4^- \cdot p_2^- \\ \hat{p}_4^- \cdot q_2^- \end{pmatrix} = \begin{pmatrix} a(a-b) - 3d(c+d) \\ a(c+d) + 3d(a-b) \end{pmatrix}.$$ *The third point* $p_3^- = (-a-b, d-c)$ *has* $q_3^- = (c-d, -a-b) \perp p_3^-$, $\mathrm{SD}(C^-; p_3^-) = (a+b)^2 + (c-d)^2$, $M(C^-; p_3^-)$ *of*

$$\begin{pmatrix} p_1^- \cdot p_3^- \\ p_1^- \cdot q_3^- \end{pmatrix} = \begin{pmatrix} -3a(a+b) + d(d-c) \\ 3a(c-d) - d(a+b) \end{pmatrix},$$

$$\begin{pmatrix} p_2^- \cdot p_3^- \\ p_2^- \cdot q_3^- \end{pmatrix} = \begin{pmatrix} a^2 - b^2 - c^2 + d^2 \\ -2(ac + bd) \end{pmatrix},$$

$$\begin{pmatrix} \hat{p}_4^- \cdot p_3^- \\ \hat{p}_4^- \cdot q_3^- \end{pmatrix} = \begin{pmatrix} a(a+b) + 3d(c-d) \\ a(d-c) + 3d(a+b) \end{pmatrix}.$$ *The fourth point* $\hat{p}_4^- = (-a, -3d)$ *has* $q_4^- = (3d, -a) \perp \hat{p}_4^-$, $\mathrm{SD}(C^-; \hat{p}_4^-) = a^2 + 9d^2$, $M(C^-; \hat{p}_4^-)$ *consisting of*

$$\begin{pmatrix} p_1^- \cdot \hat{p}_4^- \\ p_1^- \cdot q_4^- \end{pmatrix} = \begin{pmatrix} -3(a^2 + d^2) \\ 8ad \end{pmatrix},$$

$$\begin{pmatrix} p_2^- \cdot \hat{p}_4^- \\ p_2^- \cdot q_4^- \end{pmatrix} = \begin{pmatrix} a(a-b) - 3d(d+c) \\ 3d(b-a) - a(d+c) \end{pmatrix},$$

$$\begin{pmatrix} p_3^- \cdot \hat{p}_4^- \\ p_3^- \cdot q_4^- \end{pmatrix} = \begin{pmatrix} a(a+b) + 3d(c-d) \\ -3d(a+b) + a(c-d) \end{pmatrix}.$$ *The Nested Distributed Projection* $\mathrm{NDP}(C^-)$ *consists of the four pairs (of a squared distance and $2 \times 3$ matrix) above.*

*Shorter Example 4.5 justified that $C^+ \not\cong C^-$ unless at least of the parameters $a, b, c, d$ is 0. If $a = 0$ or $d = 0$, then $C^+ \cong C^-$ are isometric. In the remaining cases $b = 0$ and $c = 0$, the clouds $C^\pm$ are mirror images, which can be distinguished by matrices $M$ above, not by any distances.*

***Case** $b = 0$. We write down the above matrices $M(C^+; p_i^+)$ with unordered columns after substituting $b = 0$.*

$$\begin{pmatrix} -3a^2 + d(d-c) & -3a^2 + d(d+c) & -3(a^2 + d^2) \\ a(3c - 4d) & -a(3c + 4d) & 8ad \end{pmatrix}$$

$$\begin{pmatrix} -3a^2 + d(d-c) & a^2 - c^2 + d^2 & a^2 + 3d(c-d) \\ a(4d - 3c) & 2ac & a(c - 4d) \end{pmatrix}$$

$$\begin{pmatrix} -3a^2 + d(d+c) & a^2 - c^2 + d^2 & a^2 - 3d(c+d) \\ a(3c + 4d) & -2ac & -a(c + 4d) \end{pmatrix}$$

$$\begin{pmatrix} -3(a^2 + d^2) & a^2 + 3d(c-d) & a^2 - 3d(c+d) \\ -8ad & a(4d - c) & a(c + 4d) \end{pmatrix}$$

*The mirror image $C^-$ has the following matrices:*

$$\begin{pmatrix} -3a^2 + d(d+c) & -3a^2 + d(d-c) & -3(a^2 + d^2) \\ a(3c + 4d) & a(4d - 3c) & -8ad \end{pmatrix}$$

$$\begin{pmatrix} -3a^2 + d(d+c) & a^2 - c^2 + d^2 & a^2 - 3d(c+d)) \\ -a(3c + 4d) & 2ac & a(c + 4d) \end{pmatrix}$$

$$\begin{pmatrix} -3a^2 + d(d-c) & a^2 - c^2 + d^2 & a^2 + 3d(c-d) \\ a(3c - 4d) & -2ac & a(4d - c) \end{pmatrix}$$

$$\begin{pmatrix} -3(a^2 + d^2) & a^2 - 3d(c+d) & a^2 + 3d(c-d) \\ 8ad & -a(c + 4d) & a(c - 4d) \end{pmatrix}$$

*By Lemma C.3(b), the reflection $C^+ \to C^-$ changes the sign of the last row in the matrix $M$ from any point-based representation PR. Indeed, changing the sign of the last row in each matrix $M$ from $\mathrm{NDP}(C^+)$ makes this matrix identical to one of the matrices from $\mathrm{NDP}(C^-)$, up to a permutation of columns as always. However, with all signs kept, the above unordered collections of four matrices are different unless all elements in the last row vanish, which happens only for a=0, when $C^+ = C_-$ are identical.*

***Case** $c = 0$ is symmetric to the case $c = 0$ under the reflection $(x, y) \mapsto (y, x)$, which swaps $b \leftrightarrow c$ and $a \leftrightarrow d$.*

*We have considered only non-negative values of $a, b, c, d$ because all other cases are obtained by symmetries. For example, the reflection $y \mapsto -y$ maps the cloud $C^+(a, b, c, d)$ to $C^-(a, -b, c, d) = C^-(a, b, -c, d)$.* ∎

Example B.2 importantly demonstrates that the invariant NDP is simple enough for manual computations.

A numerical experiment can only illustrate but not prove the conclusion of Example B.2 that all (infinitely many) non-rigidly equivalent clouds $C^{\pm}$ are distinguished by NDP.

## C  GENERALIZATION OF SECTION 3 AND ALL PROOFS IN DIMENSIONS $n \geq 2$

This appendix extends all concepts from section 3 to dimensions $n \geq 2$, extends Theorem 3.3 to Theorem C.7, which is proved with Theorem C.9 for any $n \geq 2$.

**Lemma C.1** (vector $p_n^{\perp}$ orthogonal to $p_1, \ldots, p_{n-1}$ in $\mathbb{R}^n$). *Let $e_1, \ldots, e_n$ be an orthonormal basis of $\mathbb{R}^n$, so $|e_i| = 1$ and $e_i \cdot e_j = 0$ for $i \neq j$. For any $n-1$ vectors $p_1, \ldots, p_{n-1} \in \mathbb{R}^n$, there is a vector $p_n^{\perp}$ that is orthogonal to all $p_1, \ldots, p_{n-1}$ and has coordinates that are degree $n-1$ polynomials in the coordinates of $p_1, \ldots, p_{n-1}$.*

***Proof of Lemma C.1.*** Below the 'unusual determinant' with the $n-1$ vector columns $p_1, \ldots, p_{n-1}$ and the last column of the $n$ vectors $e_1, \ldots, e_n$ is only a short notation for the following expansion

by the last column: $\begin{vmatrix} | & \cdots & | & e_1 \\ p_1 & \cdots & p_{n-1} & \vdots \\ | & \cdots & | & e_n \end{vmatrix} = \sum_{i=1}^{n} (-1)^{n+i} \det(i) e_i$, where $\det(i)$ is the usual

$(n-1) \times (n-1)$ determinant obtained from the $n-1$ vector columns $p_1, \ldots, p_{n-1}$ by removing the $i$-th row, so we set $p_n^{\perp} = \sum_{i=1}^{n} (-1)^{n+i} \det(i) e_i$.

For example, if $n = 2$ then $p_1 = (x_1, x_2)$ has the vector $p_2^{\perp} = \begin{vmatrix} x_1 & e_1 \\ x_2 & e_2 \end{vmatrix} = x_1 e_2 - x_2 e_1 =$

$(-x_2, x_1) \perp p_1$ If $n = 3$, $p_1 = (x_1, x_2, x_3)$ and $p_2 = (y_1, y_2, y_3)$, then $p_3^{\perp} = \begin{vmatrix} x_1 & y_1 & e_1 \\ x_2 & y_2 & e_2 \\ x_3 & y_3 & e_3 \end{vmatrix} =$

$\begin{vmatrix} x_2 & y_2 \\ x_3 & y_3 \end{vmatrix} e_1 - \begin{vmatrix} x_1 & y_1 \\ x_3 & y_3 \end{vmatrix} e_2 + \begin{vmatrix} x_1 & y_1 \\ x_2 & y_2 \end{vmatrix} e_3 = p_1 \times p_2$ is the *vector* product of $p_1, p_2$.

To show that $p_n^{\perp}$ is orthogonal to each $p_i$, we compute the scalar product $p_n^{\perp} \cdot p_i = \sum_{i=1}^{n} (-1)^{n+1} \det(i) e_i \cdot p_i$. Since $e_i \cdot p_i$ equals the $i$-th coordinate of the vector $p_i$, the last sum is the expansion of the $n \times n$ determinant obtained from the original $p_n^{\perp}$ above by replacing the last column with $p_i$. Since the resulting determinant contains two identical columns equal to $p_i$, we conclude that $p_n^{\perp} \cdot p_i = 0$. □

Lemma C.1 holds when given vectors $p_1, \ldots, p_{n-1} \in \mathbb{R}^n$ are linearly dependent, even if some $p_j = 0$. Then $p_n^{\perp} = 0$ is orthogonal to each $p_j$ so that $p_n^{\perp} \cdot p_j = 0$.

Definition C.2 extends a point-based representation from Definition 3.1 to dimensions $n \geq 2$. The key idea is to represent any $m$-point cloud $A \subset \mathbb{R}^n$ relative to (a simplex of) any base sequence of ordered points $p_1, \ldots, p_{n-1} \in A$. If the vectors $p_1, \ldots, p_{n-1}$ are linearly independent, they form with the vector $p_n^{\perp}$ from Lemma C.1 a (not necessarily orthogonal) basis in $\mathbb{R}^n$. Below we represent any point $p \in A$ by normalized scalar products, which are valid even if $p_1, \ldots, p_{n-1}$ are linearly dependent.

**Definition C.2** (point-based representation PR for $n \geq 2$). *For any cloud $A \subset \mathbb{R}^n$ of $m$ unordered points, the* center of mass *is $O(A) = \dfrac{1}{m} \sum_{p \in A} p$. Shift $A$ so that $O(A)$ is the origin $0 \in \mathbb{R}^n$. The* radius *of $A$ is $R(A) = \max_{p \in A} |p|$. For any* basis *sequence of points $p_1, \ldots, p_{n-1} \in A$, the* squared distance matrix $\mathrm{SD}(p_1, \ldots, p_{n-1})$ *consists of $|p_i - p_j|^2$ for $i, j = 0, \ldots, n-1$, where $p_0 = 0$. Let $p_n^{\perp}$ be the vector in Lemma C.1. For any point $q \in A - \{p_1, \ldots, p_{n-1}\}$, the $n \times (m - n + 1)$ matrix $M(A; p_1, \ldots, p_{n-1})$ has a column of scalar products $q \cdot p_1, \ldots, q \cdot p_n$. The* point-based representation $\mathrm{PR}(A; p_1, \ldots, p_{n-1})$ *is the pair*

$$\left[ \mathrm{SD}(p_1, \ldots, p_{n-1}), M(A; p_1, \ldots, p_{n-1}) \right].$$

The normalized *representation* $\mathrm{NPR}(A; p_1, \ldots, p_{n-1})$ *is obtained by dividing all components of* $\mathrm{PR}(A; p_1, \ldots, p_{n-1})$ *by* $R^2(A)$, *except the last row of* $M(A; p_1, \ldots, p_{n-1})$, *which is divided by* $R^n(A)$. ∎

**Lemma C.3** (PR under isometry). *Let a point cloud $A \subset \mathbb{R}^n$ have a base sequence $(p_1, \ldots, p_{n-1})$.*

*(a) Any rigid motion $f$ of $\mathbb{R}^n$ respects point-based representations from Definition C.2 so that*

$$\mathrm{PR}(A; p_1, \ldots, p_{n-1}) = \mathrm{PR}(f(A); f(p_1), \ldots, f(p_{n-1})).$$

*(b) For any orientation-reversing isometry $f$ of $\mathbb{R}^n$, the representation $\mathrm{PR}(f(A); f(p_1), \ldots, f(p_{n-1}))$ differs from $\mathrm{PR}(A; p_1, \ldots, p_{n-1})$ by reversing all signs in the last row of the matrix $M(A; p_1, \ldots, p_{n-1})$.*

*(c) The normalized point-based representation $\mathrm{NPR}(A; p_1, \ldots, p_{n-1})$ in Definition C.2 is preserved by any composition of rigid motion and uniform scaling.*

***Proof of Lemma C.3.*** **(a)** Since rigid motion preserves distances and scalar products, all components of the point-based representation $\mathrm{PR}(A; p_1, \ldots, p_{n-1})$ are invariant.

**(b)** Using a composition with a suitable orientation-preserving isometry (rigid motion), one can assume that $f$ is the mirror reflection in a linear hyperspace $H$ containing the origin $0$ and the base sequence $p_1, \ldots, p_{n-1}$ of $A$. Since $f$ preserves distances, $R(A)$ and $\mathrm{SD}(A; p_1, \ldots, p_{n-1})$ are invariant. Then $f$ fixes all points from $H$ including $p_1, \ldots, p_{n-1}$, hence the vector $p_n$ from Lemma C.1. Any point $q \in A - p_1, \ldots, p_{n-1}$ keeps its scalar product $q \cdot p_i$ for $i = 1, \ldots, n-1$ and changes the sign of $q \cdot p_n$, because $q$ and its mirror image $f(q)$ have opposite projections to $p_n$. The above arguments hold even if the base sequence $p_1, \ldots, p_{n-1}$ is degenerate, not generating an $(n-1)$-dimensional subspace in $\mathbb{R}^n$. Then there are infinitely many choices of $H$ above and $p_n = 0$, so the last row of $M(A; p_1, \ldots, p_{n-1})$ consists of zeros.

**(c)** Under uniform scaling by a factor $s$, all squared distances and scalar products $q \cdot p_i$, $i = 1, \ldots, n-1$, are multiplied by $s^2$. The vector $p_n^\perp$ from Lemma C.1 is multiplied by $s^{n-1}$, hence all scalar products $q \cdot p_n$ in the last row of $M(A; p_1, \ldots, p_{n-1})$ are divided by $R^n(A)$. □

The *affine dimension* $0 \le \mathrm{aff}(A) \le n$ of a cloud $A = \{p_1, \ldots, p_m\} \subset \mathbb{R}^n$ is the maximum dimension of the vector space generated by all inter-point vectors $p_i - p_j$, $i, j \in \{1, \ldots, m\}$. Then $\mathrm{aff}(A)$ is an isometry invariant and is independent of an order of points of $A$. Any cloud $A$ of 2 distinct points has $\mathrm{aff}(A) = 1$. Any cloud $A$ of 3 points that are not in the same straight line has $\mathrm{aff}(A) = 2$.

Lemma C.4 provides a simple criterion for a matrix to be realizable by squared distances of a point cloud in $\mathbb{R}^n$.

**Lemma C.4** (realization of distances). *(a) A symmetric $m \times m$ matrix of $s_{ij} \ge 0$ with $s_{ii} = 0$ is realizable as a matrix of squared distances between points $p_0 = 0, p_1, \ldots, p_{m-1} \in \mathbb{R}^n$ if and only if the $(m-1) \times (m-1)$ matrix $g_{ij} = \dfrac{s_{0i} + s_{0j} - s_{ij}}{2}$ has only non-negative eigenvalues.*

*(b) If the condition in (a) holds, $\mathrm{aff}(0, p_1, \ldots, p_{m-1})$ equals the number $k \le m-1 \le n$ of positive eigenvalues. Also in this case, $g_{ij} = p_i \cdot p_j$ define the* Gram matrix GM *of the vectors $p_1, \ldots, p_{m-1} \in \mathbb{R}^n$, which are uniquely determined in time $O(m^3)$ up to an orthogonal map in $\mathbb{R}^n$.*

***Proof of Lemma C.4.*** **(a)** We extend Theorem 1 from Dekster & Wilker (1987) to the case $m < n+1$ and also justify the reconstruction of $p_1, \ldots, p_{m-1}$ in time $O(m^3)$ uniquely in $\mathbb{R}^n$ up to an orthogonal map from the group $\mathrm{O}(n)$.

The part *only if* $\Rightarrow$. Let a symmetric matrix $S$ consist of squared distances between points $p_0 = 0, p_1, \ldots, p_{m-1} \in \mathbb{R}^n$. For $i, j = 1, \ldots, m-1$, the matrix with the elements

$$g_{ij} = \frac{s_{0i} + s_{0j} - s_{ij}}{2} = \frac{p_i^2 + p_j^2 - |p_i - p_j|^2}{2} = p_i \cdot p_j$$

is the Gram matrix, which can be written as $\mathrm{GM} = P^T P$, where the columns of the $n \times (m-1)$ matrix $P$ are the vectors $p_1, \ldots, p_{m-1}$. For any vector $v \in \mathbb{R}^{m-1}$, we have

$$0 \le |Pv|^2 = (Pv)^T(Pv) = v^T(P^T P)v = v^T \mathrm{GM} v.$$

Since the quadratic form $v^T \mathrm{GM} v \ge 0$ for any $v \in \mathbb{R}^{m-1}$, the matrix GM is positive semi-definite meaning that GM has only non-negative eigenvalues, see Theorem 7.2.7 in Horn & Johnson (2012).

The part *if* $\Leftarrow$. For any positive semi-definite matrix GM, there is an orthogonal matrix $Q$ such that $Q^T \mathrm{GM} Q = D$ is the diagonal matrix, whose $m-1$ diagonal elements are non-negative eigenvalues of GM. The diagonal matrix $\sqrt{D}$ consists of the square roots of eigenvalues of GM.

**(b)** The number of positive eigenvalues of GM equals the dimension $k = \mathrm{aff}(\{0, p_1, \ldots, p_{m-1}\})$ of the subspace in $\mathbb{R}^n$ linearly spanned by $p_1, \ldots, p_{m-1}$. We may assume that all $k \le n$ positive eigenvalues of GM correspond to the first $k$ coordinates of $\mathbb{R}^n$. Since $Q^T = Q^{-1}$, the given matrix $\mathrm{GM} = QDQ^T = (Q\sqrt{D})(Q\sqrt{D})^T$ becomes the Gram matrix of the columns of $Q\sqrt{D}$. These columns become the reconstructed vectors $p_1, \ldots, p_{m-1} \in \mathbb{R}^n$.

If there is another diagonalization $\tilde{Q}^T \mathrm{GM} \tilde{Q} = \tilde{D}$ for $\tilde{Q} \in \mathrm{O}(n)$, then $\tilde{D}$ differs from $D$ by a permutation of eigenvalues, which is realized by an orthogonal map, so we set $\tilde{D} = D$. Then $\mathrm{GM} = \tilde{Q} D \tilde{Q}^T = (\tilde{Q}\sqrt{D})(\tilde{Q}\sqrt{D})^T$ is the Gram matrix of the columns of $\tilde{Q}\sqrt{D}$.

The new columns differ from the previously reconstructed vectors $p_1, \ldots, p_{m-1} \in \mathbb{R}^n$ by the orthogonal map $Q\tilde{Q}^T$. Hence the reconstruction is unique up to $\mathrm{O}(n)$-transformations. Computing eigenvectors $p_1, \ldots, p_{m-1}$ needs a diagonalization of GM in time $O(m^3)$, see (Press et al., 2007, section 11.5). $\qquad\square$

Though Lemma C.4 gives a two-sided criterion for realizability of distances by points $p_1, \ldots, p_m \in \mathbb{R}^n$, the space of distance matrices is highly singular and cannot be easily sampled. Even $m = 4$ points in $\mathbb{R}^2$ have 6 distances that should satisfy a polynomial equation saying that the tetrahedron with these 6 edge lengths has volume 0.

So a randomly sampled matrix of potential distances for $m > n + 1$ is unlikely to be realizable by a cloud of $m$ ordered points in $\mathbb{R}^n$. Hence Lemma C.4 for $m \le n + 1$ is complemented by Theorem C.7 describing the much more practical realizabilty of a point-based representation.

Chapter 3 in Liberti & Lavor (2017) discusses realizations of a complete graph given by a distance matrix in $\mathbb{R}^n$.

Lemma C.5(a) and later results hold for all clouds including degenerate ones, e.g. for 3 points in a straight line.

Any points $p_1, \ldots, p_{n-1} \in A$ have $\mathrm{aff}(p_1, \ldots, p_{n-1}) \le n - 2$. For example, any two distinct points in $A \subset \mathbb{R}^3$ generate a straight line. Lemma C.5(c) proves that $\mathrm{PR}(A; p_1, \ldots, p_{n-1})$ suffices to reconstruct a cloud $A \subset \mathbb{R}^n$ for a suitable sequence $p_1, \ldots, p_{n-1}$. In $\mathbb{R}^2$, any point $p_1 \ne O(A)$ forms a suitable $\{p_1\}$. In $\mathbb{R}^3$, one can choose any distinct points $p_1, p_2 \in A$ so that the infinite straight line via $p_1, p_2$ avoids $O(A)$.

If there are no such $p_1, p_2$, then $A \subset \mathbb{R}^3$ is contained in a straight line $L$, so $\mathrm{aff}(A) = 1$. In this degenerate case, the stronger condition $\mathrm{aff}(O(A) \cup \{p_1, \ldots, p_{n-1}\}) = \mathrm{aff}(A)$ will help reconstruct $A \subset L$ by using any point $p_1 \ne O(A)$. The first step is to reconstruct any ordered sequence from its distance matrix in Lemma C.5(a).

Lemma C.5 improves Lemma E.5 in Widdowson & Kurlin (2023) by justifying a time for a point cloud reconstruction based on Lemma C.4.

**Lemma C.5** (reconstruction). *(a) Any sequence of ordered points $p_1, \ldots, p_m$ in $\mathbb{R}^n$ can be reconstructed (uniquely up to isometry) from the matrix of the Euclidean distances $|p_i - p_j|$ in time $O(m^3)$. If all distances are divided by $R = \max\limits_{i=1,\ldots,m} |p_i|$, the reconstruction of $p_1, \ldots, p_m$ is unique up to isometry and uniform scaling in $\mathbb{R}^n$.*

*(b) If $m \leq n$, the uniqueness of reconstructions in part (a) remains true if we replace isometry by rigid motion in $\mathbb{R}^n$.*

*(c) Any cloud $A \subset \mathbb{R}^n$ of $m$ unordered points can be reconstructed (uniquely up to rigid motion in $\mathbb{R}^n$) from a point-based representation $\mathrm{PR}(A; p_1, \ldots, p_{n-1})$ in time $O(m^3)$ for any $p_1, \ldots, p_{n-1} \in A$ with $\mathrm{aff}(O(A) \cup \{p_1, \ldots, p_{n-1}\}) = \mathrm{aff}(A)$. If $\mathrm{aff}(A) = n$, then $\mathrm{aff}(O(A) \cup \{p_1, \ldots, p_{n-1}\}) = n - 1$ suffices. Any cloud $A \subset \mathbb{R}^n$ has a suitable sequence $p_1, \ldots, p_{n-1}$ in all cases.*

***Proof of Lemma C.5.*** **(a)** By translation, we can put $p_1$ at the origin $0 \in \mathbb{R}^n$. Let $G$ be the $(m - 1) \times (m - 1)$ matrix $G_{ij} = \dfrac{p_i^2 + p_j^2 - |p_i - p_j|^2}{2} = p_i \cdot p_j$ constructed from squared distances between $p_1 = 0, \ldots, p_m$ for $i, j = 2, \ldots, m$. By Lemma C.4 if $G$ has $k \leq n$ positive eigenvalues, then $p_1 = 0, \ldots, p_m$ can be uniquely determined up to isometry in $\mathbb{R}^k \subset \mathbb{R}^n$ in time $O(m^3)$. If all distances are divided by the same radius $R(p\{m\})$, the above construction guarantees uniqueness up to isometry and uniform scaling.

**(b)** If $m \leq n$, any mirror images of $p\{m\} \subset \mathbb{R}^n$ after a suitable rigid motion in $\mathbb{R}^n$ can be assumed to belong to an $(n - 1)$-dimensional hyperspace $H \subset \mathbb{R}^n$, where they are matched by a mirror reflection $H \to H$ with respect to an $(n - 2)$-dimensional subspace $S \subset H$, which is realized by the $180°$ orientation-preserving rotation of $\mathbb{R}^n$ around $S$.

**(c)** We will reconstruct a cloud $A \subset \mathbb{R}^n$ so that the center of mass $O(A)$ is the origin $0 \in \mathbb{R}^n$. If $\mathrm{aff}(A) = k < n$, the cloud $A \subset \mathbb{R}^n$ is contained in an affine $k$-dimensional subspace, which can be rigidly moved to the linear subspace $\mathbb{R}^k \subset \mathbb{R}^n$ for the first $k$ of $n$ coordinates in $\mathbb{R}^n$.

It suffices to reconstruct $A \subset \mathbb{R}^k$ up to rigid motion in $\mathbb{R}^k$. Since $\mathrm{aff}(0, p_1, \ldots, p_{n-1}) = k$, some $k$ vectors (say) $p_1, \ldots, p_k$ from $p_1, \ldots, p_{n-1}$ form a linear basis of $\mathbb{R}^k$. The $k$ points $p_1, \ldots, p_k$ are uniquely reconstructed up to rigid motion in $\mathbb{R}^k$ by part (b). Any other point $q \in A - \{p_1, \ldots, p_k\}$ is uniquely determined by its projections $(q \cdot p_i)/|p_i|$, which can be found from the first $k < n$ rows of the matrix $M(A; p_1, \ldots, p_{n-1})$ for the point $q$, see Definition C.2.

In the generic case $\mathrm{aff}(A) = n$, the condition $\mathrm{aff}(0, p_1, \ldots, p_{n-1}) = n - 1$ means that $p_1, \ldots, p_{n-1}$ are linearly independent and hence form a linear basis of $\mathbb{R}^n$ with the extra vector $p_n^\perp$ from Lemma C.1. The sequence $(0, p_1, \ldots, p_{n-1})$ of $n$ points can be uniquely reconstructed up to rigid motion in $\mathbb{R}^n$ by part (b). Any other point $q \in A - \{p_1, \ldots, p_{n-1}\}$ is uniquely determined by its projections $\dfrac{q \cdot p_i}{|p_i|}$ to the $n$ basis vectors $p_1, \ldots, p_{n-1}, p_n^\perp$, which can be found from the column of $M(A; p_1, \ldots, p_{n-1})$ for $q$. $\qquad\square$

Lemma C.5(b) for $m = n = 3$ implies that any triangle is determined by its sides up to rigid motion in $\mathbb{R}^3$. For example, the sides $3, 4, 5$ define a right-angled triangle whose mirror images are not related by rigid motion inside a plane $H \subset \mathbb{R}^3$, but are matched by composing a suitable rigid motion in $H$ and a $180°$ rotation of $\mathbb{R}^3$ around a line in $H$.

**Lemma C.6** (smoothness of PR). *For any cloud $A \subset \mathbb{R}^n$ and a base sequence $p_1, \ldots, p_{n-1} \in A$, all components of $\mathrm{PR}(A; p_1, \ldots, p_{n-1})$ have continuous partial derivatives (of any order) with respect to all (coordinates of) points of $A$ as long as $R(A) > 0$, so some points of $A$ remain distinct.*

***Proof of Lemma C.6.*** The point-based representation $\mathrm{PR}(A; p\{n - 1\})$ consists of squared distances in the matrix $\mathrm{SD}(p\{n - 1\})$ and scalar products in the matrix $M(A; p\{n - 1\})$ of all points $q \in A - p\{n - 1\}$ with the vectors $p_1, \ldots, p_{n-1}$ from the base sequence $p\{n - 1\}$ and the vector $p_n \perp p_1, \ldots, p_{n-1}$ from Lemma C.1. All these components are polynomials in the coordinates of the points of $A$, so have all continuous partial derivatives. $\qquad\square$

Theorem C.7 extends Theorem 3.3 to dimensions $n \geq 2$.

**Theorem C.7** (realizability of abstract PR). *Let $S$ be a symmetric $n \times n$ matrix of $s_{ij} \geq 0$ with $s_{ii} = 0$. Let $M$ be any $n \times (m - n + 1)$ matrix for $m \geq n$. The pair $[S, M]$ is realizable as a point-based representation $\mathrm{PR}(A; p_1, \ldots, p_{n-1})$ for a cloud $A \subset \mathbb{R}^n$ of $m$ points with $O(A) = 0$*

*and a base sequence $p_1, \ldots, p_{n-1}$ if and only if (1) the $(n-1) \times (n-1)$ matrix $G_{ij} = \frac{1}{2}(s_{1i} + s_{1j} - s_{ij})$ has only positive eigenvalues, which uniquely determines $p_1, \ldots, p_{n-1}$ up to isometry, and (2) $\sum_{j=1}^{n-1} (p_i \cdot p_j) + \sum_{j=1}^{m-n+1} M_{ij} = 0$ for $i = 1, \ldots, n$, where $p_n = p_n^{\perp}$ is the orthogonal vector from Lemma C.1.*

**Proof of Theorem** *C.7.* The realizability of $S$ as a matrix of squared distances between $n$ points $0, p_1, \ldots, p_{n-1}$ from the base sequence $p_1, \ldots, p_{n-1}$ follows from Lemma C.4. The orthogonal vector $p_n^{\perp}$ (also denoted by $p_n$ here for uniformity) from Lemma C.1 complements $p_1, \ldots, p_{n-1}$ to a linear basis of $\mathbb{R}^n$. By Definition C.2, every element $M_{ij}$ of the matrix $M = M(A; p_1, \ldots, p_{n-1})$ equals $p_i \cdot q$ for some $q \in A - \{p_1, \ldots, p_{n-1}\}$, where $i = 1, \ldots, n$.

Hence $\sum_{j=1}^{n-1} (p_i \cdot p_j) + \sum_{j=1}^{m-n+1} M_{ij} = 0$ can be rewritten as $p_i \cdot (\sum_{p \in A} p) = 0$ for $i = 1, \ldots, n$. These $n$ equations mean that $O(A) = \frac{1}{m} \sum_{p \in A} p$ is at the origin $0 \in \mathbb{R}^n$.

Conversely, for any $M$ satisfying condition (2), we interpret every column $(M_{1j}, \ldots, M_{nj})^T$ as a vector of scalar products $(q \cdot p_1, \ldots, q \cdot p_n)$, which determine a position of a point $q \in A - \{p_1, \ldots, p_{n-1}\}$ in the basis $p_1, \ldots, p_n$. □

In Theorem C.7, condition (2) is equivalent to $O(A) = 0 \in \mathbb{R}^n$ and implies that $m - n$ columns of $M$ consist of free parameters, which determine the remaining column.

For $n = 2$, condition (1) means only that $s_{12} > 0$, so the distance between the points $p_0 = 0$ and $p_1$ is positive.

For $n = 3$, condition (1) about positive eigenvalues of the $2 \times 2$ matrix $G$ means that 3 distances $a \leq b \leq c$ between points $0, p_1, p_2$ in $\mathbb{R}^3$ satisfy $a > 0$ and $a + b > c$, so the triangle on $0, p_1, p_2$ is non-degenerate. By the cosine theorem $p_1 \cdot p_2 = \frac{1}{2}(a^2 + b^2 - c^2)$, so the matrix $G = \begin{pmatrix} a^2 & \frac{1}{2}(a^2 + b^2 - c^2) \\ \frac{1}{2}(a^2 + b^2 - c^2) & b^2 \end{pmatrix}$ has $a^2 > 0$ and a positive determinant:

$$4 \det G = 4a^2 b^2 - (a^2 + b^2 - c^2)^2 =$$
$$(c^2 - (a^2 - 2ab + b^2))((a^2 + 2ab + b^2) - c^2) =$$
$$(c^2 - (a - b)^2)((a + b)^2 - c^2) > 0.$$

Assuming that $0 < a \leq b \leq c$, the last inequality is equivalent to one triangle inequality $a + b > c$.

Now we extend a point-based representation from Definition C.2 to a complete invariant of a point cloud $A$ under rigid motion in $\mathbb{R}^n$. In applications, $A$ can have distinguished points, for example, heavy atoms in atomic clouds, which can be used to minimize choices for $p_1, \ldots, p_{n-1}$.

Definition C.8 will extend Definition 3.4 to $n > 2$ by combining all $\mathrm{PR}(A; p_1, \ldots, p_{n-1})$ in a nested invariant by dropping points $p_1, \ldots, p_{n-1} \in A$ one at a time. This invariant is needed only for comparisons (metric computations), while any cloud $A$ can be stored in computer memory as a single $\mathrm{PR}(A; p_1, \ldots, p_{n-1})$ due to Theorem C.7.

**Definition C.8** (NDP : Nested Distributed Projection). *Let $A \subset \mathbb{R}^n$ be any cloud of $m$ unordered points. For any ordered points $p_1, \ldots, p_{n-2} \in A$, let $\mathrm{NDP}(A; p_1, \ldots, p_{n-2})$ be the unordered collection of $\mathrm{PR}(A; p_1, \ldots, p_{n-1})$ for all points $p_{n-1} \in A - \{p_1, \ldots, p_{n-2}\}$. Similarly, for any $1 \leq k \leq n - 2$, let $\mathrm{NDP}(A; p_1, \ldots, p_{k-1})$ be the unordered collection of $\mathrm{NDP}(A; p_1, \ldots, p_k)$ for all points $p_k \in A - \{p_1, \ldots, p_{k-1}\}$. For $k = 1$, the full Nested Distributed Projection $\mathrm{NDP}(A)$ depends only on $A$.* ∎

For $n = 2$ and any cloud $A \subset \mathbb{R}^2$, the Nested Distributed Projection $\mathrm{NDP}(A)$ in Definition C.8 is the same as in Definition 3.4, i.e. $\mathrm{NDP}(A)$ is the unordered collection of point-based representations $\mathrm{PR}(A; p_1)$ for all $p_1 \in A$.

For $n = 3$ and any $A \subset \mathbb{R}^3$, the Nested Distributed Projection $\mathrm{NDP}(A)$ is the unordered collection of $\mathrm{NDP}(A; p_1)$ for all $p_1 \in A$. Each $\mathrm{NDP}(A; p_1)$ is the unordered collection of $\mathrm{PR}(A; p_1, p_2)$ for all $p_2 \in A - \{p_1\}$.

Similarly to Definition 3.4, if a cloud $A$ has internal symmetries as in Example 3.2, one can collapse identical objects to a single one with a weight to speed up computations. We avoid collapsing only to simplify arguments for $n > 2$.

Lemma C.5(c) implies that any cloud $A \subset \mathbb{R}^n$ of $m$ unordered points can be reconstructed from $\mathrm{NDP}(A)$ uniquely up to rigid motion. Indeed, $\mathrm{NDP}(A)$ contains (nested) PRs depending on all possible $n - 1$ points $p_1, \ldots, p_{n-1} \in A$. At least one $\mathrm{PR}(A; p_1, \ldots, p_{n-1})$ satisfies Lemma C.5(c) and suffices to reconstruct $A$ uniquely up to rigid motion.

In Theorem C.9 for $n > 2$, the equality $\mathrm{NDP}(A) = \mathrm{NDP}(B)$ means a bijection $\beta : \mathrm{NDP}(A) \to \mathrm{NDP}(B)$ respecting the nested structure of all PRs in Definition C.8.

In detail, for any $1 \leq k \leq n-1$ and points $p_1, \ldots, p_k$, the bijection $\beta$ matches $\mathrm{NDP}(A; p_1, \ldots, p_k)$ with a unique $\mathrm{NDP}(B; q_1, \ldots, q_k)$ for some $q_1, \ldots, q_k \in B$.

If $n = 3$, then $\beta$ matches every $\mathrm{NDP}(A; p_1)$ with a unique $\mathrm{NDP}(B; q_1)$ in the sense that this bijection $\mathrm{NDP}(A; p_1) \to \mathrm{NDP}(B; q_1)$ matches $\mathrm{PR}(A; p_1, p_2)$ for every $p_2 \in A - \{p_1\}$ with $\mathrm{PR}(B; q_1, q_2)$ for a unique $q_2 \in B - \{q_1\}$.

**Theorem C.9** (completeness of NDP). *The Nested Distributed Projection is complete in the sense that any clouds $A, B \subset \mathbb{R}^n$ of $m$ unordered points are related by rigid motion in $\mathbb{R}^n$ if and only if $\mathrm{NDP}(A) = \mathrm{NDP}(B)$ so that there is a bijection $\mathrm{NDP}(A) \to \mathrm{NDP}(B)$ matching all PRs.* ∎

***Proof of Theorem C.9.*** The part *only if* : we will prove that any rigid motion $f$ moving the cloud $A$ to $B = f(A)$ implies that $\mathrm{NDP}(A) = \mathrm{NDP}(B)$. By Lemma C.3(a) the rigid motion $f$ matches every $\mathrm{PR}(A; p_1, \ldots, p_{n-1})$ from $\mathrm{NDP}(A)$ with $\mathrm{PR}(B; f(p_1), \ldots, f(p_{n-1}))$. Then, for any $1 \leq k \leq n - 2$ and $p_1, \ldots, p_k \in A$, we get a bijection $\mathrm{NDP}(A; p_1, \ldots, p_k) \to \mathrm{NDP}(B; f(p_1), \ldots, f(p_k))$ Hence $f$ induces a bijecton $\mathrm{NCP}(A) \to \mathrm{NCP}(B)$ between all PRs respecting the nested structure in Definition C.8.

The part *if* : $\mathrm{NDP}(A) = \mathrm{NDP}(B)$ will guarantee a rigid motion $f$ moving the cloud $A$ to $B = f(A)$. Choose any base sequence $p_1, \ldots, p_{n-1} \in A$ that suffices for a unique reconstruction of $A \subset \mathbb{R}^n$ up to rigid motion in Lemma C.5(c). The given bijection $\mathrm{NDP}(A) \to \mathrm{NDP}(B)$ matches $\mathrm{PR}(A; p_1, \ldots, p_{n-1})$ with an equal $\mathrm{PR}(B; q_1, \ldots, q_{n-1})$ for some $q_1, \ldots, q_{n-1} \in B$.

Lemma C.5(c) implies that a reconstruction of $A, B$ from $\mathrm{PR}(A; \sigma(p_1, \ldots, p_{n-1})) = \mathrm{PR}(B; q_1, \ldots, q_{n-1})$ is unique up to rigid motion in $\mathbb{R}^n$ so that $A, B$ are matched by a rigid motion $f$ as required. If $\mathrm{aff}(A) = \mathrm{aff}(B) < n$, this motion $f$ may not be unique. For example, any clouds $A, B \subset \mathbb{R}^3$ that are contained in a straight line $L \subset \mathbb{R}^3$ are pointwise fixed by any rotation around the line $L$. □

# D  GENERALIZATION OF SECTION 4 AND ALL PROOFS IN DIMENSIONS $n \geq 2$

This appendix extends the metrics to dimensions $n \geq 2$ and proves all metric results from section 4 in full generality.

The point-based representation in Definition C.2 included the matrix $\mathrm{SD}(p_1, \ldots, p_{n-1})$ of squared distances, which can be rewritten as a vector row-by-row.

Below we can take any norm on matrices and choose the simplest max norm below for consistency with the bottleneck distance and for Lipschitz constant 2 in Theorem E.5.

**Definition D.1** (max norm and metric on matrices). *The* max norm *$||D||_\infty = \max\limits_{i,j} |D_{ij}|$ of a matrix is the maximum absolute value of its elements $D_{ij}$. The* max metric *between matrices $M, M'$ of the same size is $d_\infty = ||M - M'||_\infty$.*

Definition D.2 will extend Definition 4.2 to dimensions $n \geq 2$. Below the notation $\mathrm{SD}/R$ means that all elements of a matrix SD are divided by $R$. The radius of a base sequence $p\{n-1\} = (p_1, \ldots, p_{n-1}) \subset A$ is defined as $R(p\{n-1\}) = \max_{i=1,\ldots,n-1} |p_i|$ in the same way as $R(A)$ of a full cloud $A$. The notation $M/R$ means that all elements in the first $n-1$ rows of a matrix $M$ are divided by $R$, and by $R^{n-1}$ in the $n$-th row, because $p_n^{\perp}$ in Lemma C.1 is a polynomial of degree $n-1$. Then PRM and further metrics have units of original points. One more division by $R$ makes all metrics invariant under scaling.

**Definition D.2** (Point-Based Representation Metric). *Let clouds $A, B \subset \mathbb{R}^n$ of $m$ unordered points have base sequences $p\{n-1\} = (p_1, \ldots, p_{n-1})$, $q\{n-1\} = (q_1, \ldots, q_{n-1})$ of ordered points, from Definition C.2. The* Point-Based Representation Metric *between the PRs above is*

$$\mathrm{PRM} = \max\{ |R(p\{n-1\}) - R(q\{n-1\})|, w_D, |R(A) - R(B)|, w_M \}, \text{ where}$$

$$w_D = d_\infty \left( \frac{\mathrm{SD}(p\{n-1\})}{R(p\{n-1\})}, \frac{\mathrm{SD}(q\{n-1\})}{R(q\{n-1\})} \right), \text{ and } w_M = \mathrm{BD} \left( \frac{M(A; p\{n-1\})}{R(A)}, \frac{M(B; q\{n-1\})}{R(B)} \right).$$

**Lemma D.3** (axioms for PRM). *PRM in Definition D.2 satisfies all metric axioms from Problem (1.1b) on any point-based representations from Definition C.8.*

***Proof of Lemma D.3****. The first axiom means that $\mathrm{PRM}(\mathrm{PR}(A; p\{n-1\}), \mathrm{PR}(B; q\{n-1\})) = 0$ if and only if these PRs are identical. The part *if*: by Lemma C.5(c), equal PRs guarantee that the clouds $A, B$ are rigidly equivalent, so $R(p\{n-1\}) = R(q\{n-1\})$, $R(A) = R(B)$, $\mathrm{SD}(p\{n-1\}) = \mathrm{SD}(q\{n-1\})$, and $M(A; p\{n-1\}) = M(B; q\{n-1\})$, so $\mathrm{PRM} = 0$.

The part *only if*: by Definition D.2 the equality $\mathrm{PRM} = 0$ means that $R(A) = R(B)$ and $w_D = 0 = w_M$. The coincidence axioms for the max metric and bottleneck distance together with $R(p\{n-1\}) = R(q\{n-1\})$ and $R(A) = R(B)$ imply that $\mathrm{SD}(p\{n-1\}) = \mathrm{SD}(q\{n-1\})$ and $M(A; p\{n-1\}) = M(B; q\{n-1\})$. Then the point-based representations become identical: $\mathrm{PR}(A; p\{n-1\}) = \mathrm{PR}(B; q\{n-1\})$.

The symmetry axiom for PRM follows from the symmetry axiom for the bottleneck distance and max metric $d_\infty$. Since each of the distances $|R(A) - R(B)|, w_D, w_M$ satisfies the triangle inequality, then so does their maximum, see metric transforms in section 4.1 of Deza & Deza (2009). $\square$

Definition D.4 extends Definition 4.4 to all dimensions $n > 2$.

**Definition D.4** (NBM : Nested Bottleneck Metric). *Let $A, B \subset \mathbb{R}^n$ be any clouds of $m$ unordered points. For any ordered points $p_1 \ldots, p_{n-2} \in A$ and $q_1 \ldots, q_{n-2} \in B$, the complete bipartite graph $\Gamma(A; p_1, \ldots, p_{n-2}; B; q_1, \ldots, q_{n-2})$ has $m - n + 2$ white vertices and $m - n + 2$ black vertices representing $\mathrm{PR}(A; p_1, \ldots, p_{n-1})$ and $\mathrm{PR}(B; q_1, \ldots, q_{n-1})$ for all $m - n + 1$ variable points $p_{n-1} \in A - \{p_1, \ldots, p_{n-2}\}$ and $q_{n-1} \in B - \{q_1, \ldots, q_{n-2}\}$, respectively.*

*Set the* weight $w(e)$ *of an edge $e$ joining the vertices represented by $\mathrm{PR}(A; p_1, \ldots, p_{n-1})$ and $\mathrm{PR}(B; q_1, \ldots, q_{n-1})$ as PRM between these PRs, see Definition D.2. Then Definition 4.3 gives us the bottleneck matching distance $\mathrm{BMD}(\Gamma(A; p_1, \ldots, p_{n-2}; B; q_1, \ldots, q_{n-2}))$. We continue dropping points iteratively. For any $1 \leq k \leq n - 2$ and ordered points $p_1 \ldots, p_{k-1} \in A$ and $q_1 \ldots, q_{k-1} \in B$, the complete bipartite graph $\Gamma(A; p_1, \ldots, p_{k-1}; B; q_1, \ldots, q_{k-1})$ has $m - k + 1$ white vertices and $m - k + 1$ black vertices representing $\mathrm{NDP}(A; p_1, \ldots, p_k)$ and $\mathrm{NDP}(B; q_1, \ldots, q_k)$ for all $m - k + 1$ variable points $p_k \in A - \{p_1, \ldots, p_{k-1}\}$ and $q_k \in B - \{q_1, \ldots, q_{k-1}\}$, respectively.*

*Set the* weight $w(e)$ *of an edge $e$ joining the vertices represented by $\mathrm{NDP}(A; p_1, \ldots, p_k)$ and $\mathrm{NDP}(B; q_1, \ldots, q_k)$ as $\mathrm{BMD}(\Gamma(A; p_1, \ldots, p_k; B; q_1, \ldots, q_k))$ obtained above. Then Definition 4.3 gives us the bottleneck matching distance $\mathrm{BMD}(\Gamma(A; p_1, \ldots, p_{k-1}; B; q_1, \ldots, q_{k-1}))$. Finally, for $k = 1$, we get the* Nested Bottleneck Metric $\mathrm{NBM}(A, B) = \mathrm{BMD}(\Gamma(A, B))$. $\blacksquare$

**Lemma D.5** (metric axioms for the bottleneck matching distance BMD). *Let $S, Q$ be any unordered distributions of the same number of objects with a base metric $d$. Define the complete bipartite graph $\Gamma(S, Q)$ whose every edge $e$ joining objects $R_S \in S$ and $R_Q \in Q$ has the weight $w(e) = d(R_S, R_Q)$. Then the bottleneck matching distance $\mathrm{BMD}(\Gamma(S, Q))$ from Definition 4.3 satisfies all metric axioms on such unordered distributions.*

***Proof of Lemma*** *D.5.* The coincidence axiom means that $\mathrm{NBM}(S,Q) = 0$ if and only if the weighted distributions $S, Q$ are equal in the sense that there is a bijection $g : S \to Q$ so that $d(g(R), R) = 0$ for any $R \in S$.

Indeed, if the weighted distributions $S, Q$ can be matched by a bijection, we get a vertex matching $E$ of $\Gamma(S, Q)$ whose all edges have weights $w(e) = 0$. Definition 4.3 implies that $\mathrm{BMD}(\Gamma(S, Q)) = 0$ as required.

Conversely, if $\mathrm{BMD}(\Gamma(S, Q)) = 0$, there is a vertex matching $E$ in $\Gamma(S, Q)$ with all $w(e) = 0$. This matching $E$ defines a required bijection $S \to Q$. The symmetry $\mathrm{BMD}(\Gamma(S, Q)) = \mathrm{BMD}(\Gamma(Q, S))$ follows from Definition 4.3 and the symmetry of the base metric $d$.

To prove the triangle inequality

$$\mathrm{BMD}(\Gamma(S, Q)) + \mathrm{BMD}(\Gamma(Q, T)) \geq \mathrm{BMD}(\Gamma(S, T)),$$

let $E_{SQ}, E_{QT}$ be optimal vertex matchings in the graphs $\Gamma(S, Q), \Gamma(Q, T)$, respectively, such that

$$\mathrm{BMD}(\Gamma(S, Q)) = W(E_{SQ}), \mathrm{BMD}(\Gamma(Q, T)) = W(E_{QT}),$$

see Definition 4.3. The composition $E_{SQ} \circ E_{QT}$ is a vertex matching in $\Gamma(S, T)$, so $W(E_{SQ} \circ E_{QT}) \geq \mathrm{BMD}(\Gamma(S, T))$. It suffices to prove that

$$W(E_{SQ}) + W(E_{QT}) \geq W(E_{SQ} \circ E_{QT}).$$

Let $e_{ST}$ be an edge with a largest weight from $E_{SQ} \circ E_{QT}$, so $W(E_{SQ} \circ E_{QT}) = w(e_{ST})$. The edge $e_{ST}$ can be considered the union of edges $e_{SQ} \in E_{SQ}, e_{QT} \in E_{QT}$.

By the triangle inequality for the base metric $d$,

$$w(e_{SQ}) + w(e_{QT}) \geq w(e_{ST}) = W(E_{SQ} \circ E_{QT})$$

implies that

$$W(E_{SQ}) + W(E_{QT}) \geq W(E_{SQ} \circ E_{QT})$$

because both terms on the left-hand side are maximized for all edges (not only $e_{SQ}, e_{QT}$) from $E_{SQ}, E_{QT}$. $\qquad\square$

**Lemma D.6** (metric axioms for NBM between NDPs). *The Nested Bottleneck Metric* NBM *from Definition D.4 satisfies all metric axioms on Nested Distributed Projections.*

***Proof of Lemma*** *D.6.* Induction on $k = n - 2, \ldots, 1$. The inductive base $k = n - 2$ follows from the metric axioms in Lemma D.3 for PRM in Definition D.2. The inductive step from $1 < k < n - 2$ to $k - 1$ follows from Lemma D.5 and the metric axioms in the inductive hypothesis for $k$. $\qquad\square$

# E GENERALIZATION OF SECTION 5 AND ALL PROOFS

This appendix proves Theorems E.5, E.8, and E.9 extending Lemmas 5.1, 5.2, and 5.3, respectively to dimensions $n \geq 2$ by using auxiliary Lemmas E.1, E.2, E.4, and Proposition E.3.

**Lemma E.1** (orthogonal vector length). *For any sequence $p_1, \ldots, p_{n-1} \in \mathbb{R}^n$, set $R = \max\limits_{i=1,\ldots,n-1} |p_i|$. Then the orthogonal vector $p_n^{\perp} \perp p_1, \ldots, p_{n-1}$ from Lemma C.1 has a length satisfying $|p_2^{\perp}| = R$, $|p_3^{\perp}| \leq R^2$, and $|p_n^{\perp}| \leq \sqrt{n} R^{n-1}$ for any $n > 3$.*

***Proof of Lemma*** *E.1.* For $n = 2$, the explicit formula $p_2^{\perp} = (-y, x)$ for $p_1 = (x, y)$ gives the exact equality $|p_2^{\perp}| = |p_1| = R$. For $n = 3$, $p_3^{\perp}$ equals the vector product $p_1 \times p_2$ whose length is $|p_3^{\perp}| \leq$

$|p_1| \cdot |p_2| \leq R^2$. For $> 3$, the expansion of the $n \times n$ determinant $p_n^{\perp} = \begin{vmatrix} | & \cdots & | & e_1 \\ p_1 & \cdots & p_{n-1} & \vdots \\ | & \cdots & | & e_n \end{vmatrix}$

along the last column gives $p_n^{\perp} = \sum\limits_{i=1}^{n} (-1)^{n+i} \det(i) e_i$, where $\det(i)$ is the $(n-1) \times (n-1)$ determinant obtained from the $n - 1$ vector columns $p_1, \ldots, p_{n-1}$ by removing the row of all $i$-th

coordinates. Any determinant on vectors $v_1, \ldots, v_{n-1} \in \mathbb{R}^{n-1}$ equals the signed volume of the parallelepiped on $v_1, \ldots, v_{n-1}$, which has the upper bound $|v_1| \cdots |v_{n-1}|$.

Since each vector $v_i$ is obtained from $p_i$ by removing one coordinate, we get $|v_i| \leq |p_i|$. So each coordinate of $p_n^\perp$ in the orthonormal basis $e_1, \ldots, e_n$ has the upper bound $|p_1| \cdots |p_{n-1}| \leq R^{n-1}$. Then the Euclidean length has the upper bound $|p_n^\perp| \leq \sqrt{n(R^{n-1})^2} = \sqrt{n} R^{n-1}$. $\qquad\square$

**Lemma E.2** (vector perturbations). *Let points $q_1, \ldots, q_{n-1}$ be $\varepsilon$-perturbations of $p_1, \ldots, p_{n-1} \in \mathbb{R}^n$ so that $|p_i - q_i| \leq \varepsilon$ for any $i = 1, \ldots, n-1$. Set $R = \max\limits_{i=1,\ldots,n-1}\{|p_i|, |q_i|\}$. The orthogonal vectors $p_n^\perp \perp p_1, \ldots, p_{n-1}$ and $q_n^\perp \perp q_1, \ldots, q_{n-1}$ from Lemma C.1 satisfy $|p_2^\perp - q_2^\perp| \leq \varepsilon$ for $n = 2$, $|p_3^\perp - q_3^\perp| \leq \varepsilon 2\sqrt{6}R$ for $n = 3$, and $|p_n^\perp - q_n^\perp| \leq \varepsilon n(n-1)R^{n-2}$ for any $n > 3$.*

***Proof of Lemma E.2.*** If $n = 2$, then $p_2^\perp = (-y, x)$ for $p_1 = (x, y)$, so $|p_2^\perp - q_2^\perp| = |p_1 - q_1| \leq \varepsilon$.

Let $x_i(v_j)$ be the $i$-th coordinate of a variable vector $v_j \in \mathbb{R}^n$ moving from $p_j$ to its $\varepsilon$-perturbation $q_j$ for $i, j = 1, \ldots, n$ in the given orthonormal basis $e_1, \ldots, e_n$, where we set $p_n = p_n^\perp$ and $q_n = q_n^\perp$ for brevity. For each $k = 1, \ldots, n$, the coordinate $x_k(v_n)$ is the scalar function $f_k(v_1, \ldots, v_{n-1})$ of the $(n-1)^2$ variables $x_i(v_j)$ for $i, j = 1, \ldots, n-1$.

The upper bound for $|p_n - q_n|$ will follow from the Mean Value Theorem 5.10 from Rudin et al. (1976) for the functions $f_1, \ldots, f_n$ because the coordinates of the vector $q_n^\perp$ are $f_k(q_1, \ldots, q_{n-1})$ evaluated at close (coordinates of the) vectors $q_1, \ldots, q_{n-1}$ so that $|p_j - q_j| \leq \varepsilon$ for $i, j = 1, \ldots, n-1$.

First we estimate the gradient $\nabla f_k$ of $f_k$ at any intermediate point in the line segment between $(p_1, \ldots, p_{n-1})$ and $(q_1, \ldots, q_{n-1})$ with respect to the $(n-1)^2$ variables $x_i(v_j)$ for $i, j = 1, \ldots, n-1$. For $k = i$, the $k$-th coordinate of $v_n = \begin{vmatrix} | & \cdots & | & e_1 \\ v_1 & \cdots & v_{n-1} & \vdots \\ | & \cdots & | & e_n \end{vmatrix}$ is $(-1)^{n+k}\det(k)$, where $\det(k)$ is the $(n-1) \times (n-1)$ determinant obtained from the $n-1$ vector columns $v_1, \ldots, v_{n-1}$ by removing the row of all $k$-th coordinates. Then $\dfrac{\partial f_k}{\partial x_i(v_j)} = (-1)^{n+k}\dfrac{\partial \det(k)}{\partial x_i(v_j)}$, which equals 0 for $k = i$ because $f_k$ is independent of the coordinate $x_k(v_j)$ for $j = 1, \ldots, n-1$.

After expanding the determinant $\det(k)$ along the $i$-th row, the only terms containing the factor $x_i(v_j)$ form the smaller $(n-2) \times (n-2)$ determinant $\det(k, i)$ obtained from the $n-2$ vector columns $v_1, \ldots, v_{j-1}, v_{j+1}, \ldots, v_{n-1}$ after removing the rows of all $k$-th and $i$-th coordinates.

Then $|v_j| \leq R = \max\limits_{i=1,\ldots,n-1}\{|p_i|, |q_i|\}$ for any points $(v_1, \ldots, v_{n-1})$ in the line segment between $(p_1, \ldots, p_{n-1})$ and $(q_1, \ldots, q_{n-1})$. The $(n-2) \times (n-2)$ determinant $\det(k, i)$ equals the signed volume on $n-2$ vectors of maximum length $R$ and hence has the upper bound $R^{n-2}$, so $\left|\dfrac{\partial f_k}{\partial x_i(v_j)}\right| = |\det(k, i)| \leq R^{n-2}$. The gradient $\nabla f_k$ is the vector of $(n-1)^2$ partial derivatives and can be considered a vector $(\nabla_1 f_k, \ldots, \nabla_{n-1} f_k)$, where $\nabla_j f_k = \left(\dfrac{\partial f_k}{x_1(v_j)}, \ldots, \dfrac{\partial f_k}{x_{n-1}(v_j)}\right)$ has

$$|\nabla_j f_k| \leq \sqrt{n-1} \max_{i=1,\ldots,n-1}\left|\dfrac{\partial f_k}{\partial x_i(v_j)}\right| \leq \sqrt{n-1} R^{n-2}.$$

We consider the $k$-th coordinate $f_k$ of $v_n$ as a function depending on one parameter $t \in [0, 1]$ when the point $(v_1, \ldots, v_{n-1})$ moves along the line segment from $(p_1, \ldots, p_{n-1})$ to $(q_1, \ldots, q_{n-1})$. Then Theorem 5.10 from Rudin et al. (1976) implies for some intermediate point $(v_1, \ldots, v_{n-1})$ that

$$|f_k(p_1, \ldots, p_{n-1}) - f_k(q_1, \ldots, q_{n-1})| = |\nabla f_k(v_1, \ldots, v_{n-1}) \cdot (p_1 - q_1, \ldots, p_{n-1} - q_{n-1})| =$$

$$= \left|\sum_{i,j=1}^{n-1} \dfrac{\partial f_k}{\partial x_i(v_j)} \cdot (x_i(p_j) - x_i(q_j))\right| = \left|\sum_{j=1}^{n-1} \nabla_j f_k \cdot (p_j - q_j)\right| \leq \sum_{j=1}^{n-1} |\nabla_j f_k| \cdot |p_j - q_j| \leq$$

$$\leq \varepsilon(n-1) \max_{j=1,\ldots,n-1} |\nabla_j f_k| \leq \varepsilon(n-1)\sqrt{n-1}R^{n-2}.$$

Since $e_1, \ldots, e_n$ form an orthonormal basis, we get

$$|p_n^\perp - q_n^\perp| = \sqrt{\sum_{k=1}^{n} |f_k(p_1,\ldots,p_{n-1}) - f_k(q_1,\ldots,q_{n-1})|^2}$$

$$\leq \sqrt{n} \max_{k=1,\ldots,n} |f_k(p_1,\ldots,p_{n-1}) - f_k(q_1,\ldots,q_{n-1})| \leq \sqrt{n}\varepsilon(n-1)\sqrt{n-1}R^{n-2} \leq \varepsilon n(n-1)R^{n-2}$$

for any $n \geq 3$. If $n = 3$, the final upper bound can be improved to $\varepsilon 2\sqrt{6}R$. $\qquad\square$

**Proposition E.3** (Lipschitz continuity of PR under perturbations of a cloud). *Let $B \subset \mathbb{R}^n$ and a base sequence $q\{n-1\} \subset B$ be obtained from a cloud $A \subset \mathbb{R}^n$ and a base sequence $p\{n-1\} \subset A$, respectively, by perturbing every point in its Euclidean $\varepsilon$-neighborhood. Then*

*(a) $|O(A) - O(B)| \leq \varepsilon$, $|R(p\{n-1\} - R(q\{n-1\})| \leq 2\varepsilon$, and $|R(A) - R(B)| \leq 2\varepsilon$;*

*(b) $\mathrm{PRM}\big(\mathrm{PR}(A; p\{n-1\}), \mathrm{PR}(B; q\{n-1\})\big) \leq \lambda_n \varepsilon$ for $\lambda_2 = 6$, $\lambda_3 = 16$, $\lambda_n = 3n^2$, $n > 3$.*

***Proof of Proposition** E.3*. **(a)** Let $p_1 \ldots, p_m$ be all points of $A$ so that the first $n-1$ points $p_1, \ldots, p_{n-1}$ form the base sequence $p\{n-1\}$. Let $q_i \in B$ be an $\varepsilon$-perturbation of $p_i$, so $q_1 \ldots, q_m$ are all points of $B$ and the first $n-1$ points $q_1, \ldots, q_{n-1}$ form the base sequence $q\{n-1\}$. The radius of $A$ is $R(A) = \max_{p \in A} |p - O(A)|$, where $O(A) = \dfrac{1}{m} \sum_{p \in A} p$ is the center of mass. Then

$$|O(A) - O(B)| = \frac{1}{m} \left| \sum_{i=1}^{m} p_i - \sum_{i=1}^{m} q_i \right| \leq \frac{1}{m} \sum_{i=1}^{m} |p_i - q_i| \leq \varepsilon.$$

If the radius $R(A)$ is attained at a point $p_i \in A$, then $R(A) = |p_i - O(A)| \leq$

$$\leq |p_i - q_i| + |q_i - O(B)| + |O(B) - O(A)| \leq \varepsilon + \max_{i=1,\ldots,m} |q_i - O(B)| + \varepsilon = 2\varepsilon + R(B).$$

Swapping the clouds $A, B$ gives the opposite inequality $R(B) \leq 2\varepsilon + R(A)$, so $|R(A) - R(B)| \leq 2\varepsilon$. The radii of the base sequences also differ by at most $2\varepsilon$, i.e. $|R(p\{n-1\}) - R(q\{n-1\})| \leq 2\varepsilon$.

**(b)** All corresponding points of the given clouds $A, B$ are $\varepsilon$-close so that $|p_i - q_i| \leq \varepsilon$ for all $i = 1, \ldots, m$. Any distance $|p_i - p_j|$ changes by at most $2\varepsilon$ under perturbation, because

$$|p_i - p_j| \leq |p_i - q_i| + |q_i - q_j| + |q_j - p_j| \leq |q_i - q_j| + 2\varepsilon,$$
$$|q_i - q_j| \leq |q_i - p_i| + |p_i - p_j| + |p_j - q_j| \leq |p_i - p_j| + 2\varepsilon.$$

Hence $\big| |p_i - p_j| - |q_i - q_j| \big| \leq 2\varepsilon$ for all $i, j = 1, \ldots, m$.

To estimate the max metric $d_\infty$ in (D.2), we rewrite the difference between the corresponding elements in the matrices $\mathrm{SD}/R$ of squared distances normalized by the radii in the notations $r(A) = R(p\{n-1\})$ and $r(B) = R(q\{n-1\})$. Without loss of generality, assume that $r(A) \geq r(B)$.

$$\text{Then } \left| \frac{|p_i - p_j|^2}{r(A)} - \frac{|q_i - q_j|^2}{r(B)} \right| \leq \frac{\big| |p_i - p_j|^2 - |q_i - q_j|^2 \big|}{r(A)} + |q_i - q_j|^2 \frac{|r(B) - r(A)|}{r(A)r(B)}$$

for $i, j = 0, \ldots, n-1$, where $p_0 = O(A)$ and $q_0 = O(B)$ are centers of mass. In the first term above, we estimate the difference of squares by factorizing:

$$\big| |p_i - p_j|^2 - |q_i - q_j|^2 \big| = \big| |p_i - p_j| - |q_i - q_j| \big| \cdot (|p_i - p_j| + |q_i - q_j|) \leq 2\varepsilon(2r(A) + 2r(B)).$$

Using $r(A) \geq r(B)$, the bounds $\dfrac{\big| |p_i - p_j|^2 - |q_i - q_j|^2 \big|}{r(A)} \leq 4\varepsilon \dfrac{r(A) + r(B)}{r(A)} \leq 8\varepsilon$, $|q_i - q_j|^2 \dfrac{|r(B) - r(A)|}{r(A)r(B)} \leq \dfrac{(2r(B))^2 \cdot 2\varepsilon}{r(A)r(B)} \leq 8\varepsilon$ give $d_\infty \left( \dfrac{\mathrm{SD}(p\{n-1\})}{r(A)}, \dfrac{\mathrm{SD}(q\{n-1\})}{r(B)} \right) \leq 16\varepsilon$.

To estimate the bottleneck distance BD between the matrices $M/R$ in (D.2), which involve scalar products, we shift both clouds $A, B$ so that their centers $O(A)$ and $O(B)$ coincide with the origin $0 \in \mathbb{R}^n$. We keep the same notation $p_i, q_i$ for all points for simplicity. Since $|O(A) - O(B)| \leq \varepsilon$ by part (a), the relative shift by a vector of a maximum length $\varepsilon$ guarantees all corresponding points are now $2\varepsilon$-close, i.e. $|p_i - q_i| \leq 2\varepsilon$. Below we estimate the difference between scalsr products involving any $2\varepsilon$-close points $p \in A - p\{n-1\}$ and $q \in B - q\{n-1\}$ for $i = 1, \ldots, n-1$ (indexing points from the base sequences) and $i = n$ for the orthogonal vectors $p_n = p_n^\perp$, $q_n = q_n^\perp$.

**Case $i = 1, \ldots, n-1$.** The bottleneck distance BD has the upper bound obtained from estimating the differences below in the $M/R$ matrices for any point $p \in A - p\{n-1\}$ matched with its $2\varepsilon$-perturbation $q \in B - q\{n-1\}$. Without loss of generality, assume that $R(A) \geq R(B)$. Then

$$\left| \frac{p \cdot p_i}{R(A)} - \frac{q \cdot q_i}{R(B)} \right| \leq \frac{|p \cdot p_i - q \cdot q_i|}{R(A)} + |q \cdot q_i| \frac{|R(B) - R(A)|}{R(A)R(B)}.$$

Due to $|q \cdot q_i| \leq |q| \cdot |q_i| \leq R^2(B)$, the second term above has the upper bound $\dfrac{R^2(B) \cdot 2\varepsilon}{R(A)R(B)} \leq 2\varepsilon$.

Estimate the difference of products in the first term above:

$$|p \cdot p_i - q \cdot q_i| \leq |(p - q) \cdot p_i + q \cdot (p_i - q_i)| \leq |p - q| \cdot |p_i| + |q| \cdot |p_i - q_i| \leq 2\varepsilon(R(A) + R(B)).$$

Then $\dfrac{|p \cdot p_i - q \cdot q_i|}{R(A)} \leq 2\varepsilon \dfrac{R(A) + R(B)}{R(A)} = 4\varepsilon$. For every $i = 1, \ldots, n-1$, we get $\left| \dfrac{p \cdot p_i}{R(A)} - \dfrac{q \cdot q_i}{R(B)} \right| \leq 6\varepsilon$ for every point $p \in A - p\{n-1\}$ and its $2\varepsilon$-perturbation $q \in B - q\{n-1\}$.

**Case $i = n$** is for the $n$-th row of the matrices $M/R$ in (D.2), where the scalar products with the orthogonal vectors $p_n^\perp, q_n^\perp$ from Lemma C.1 are divided by $R^{n-1}$ instead of $R$.

**Subcase $i = n = 2$** coincides with the case $i < n$ above because $R^{n-1} = R$. Combining the upper bounds above, we get BD $\left( \dfrac{M(A; p\{n-1\})}{R(A)}, \dfrac{M(B; q\{n-1\})}{R(B)} \right) \leq 6\varepsilon$ By Definition 4.2, the Point-based Representation Metric PRM equals the maximum of the bounds $d_\infty = |R(p_1) - R(q_1)| = ||p_1| - |q_1|| \leq 2\varepsilon$, $|R(A) - R(B)| \leq 2\varepsilon$, and BD above, so $\mathrm{PRM}(\mathrm{PR}(A; p_1), \mathrm{PR}(B; q_1)) \leq 6\varepsilon$, which finishes the proof of part (b) for $n = 2$.

**Subcase $i = n = 3$.** Without loss of generality, we can assume that $R(A) \geq R(B)$. The upper bounds of Lemmas E.1 and E.2 imply that

$$|p_3^\perp| \leq R^2(A), \quad |q_3^\perp| \leq R^2(B), \quad |p_3^\perp - q_3^\perp| \leq 2\varepsilon \cdot 2\sqrt{6}R(A).$$

We start estimating similarly to the case $i < n$ above:

$$|p \cdot p_3^\perp - q \cdot q_3^\perp| \leq |(p - q) \cdot p_3^\perp + q \cdot (p_3^\perp - q_3^\perp)| \leq |p - q| \cdot |p_3^\perp| + |q| \cdot |p_3^\perp - q_3^\perp| \leq$$
$$2\varepsilon R^2(A) + R(B) \cdot 2\varepsilon \cdot 2\sqrt{6}R(A) = 2\varepsilon R(A)(R(A) + 4\sqrt{6}R(B)).$$

$$\text{Then } \left| \frac{p \cdot p_3^\perp}{R^2(A)} - \frac{q \cdot q_3^\perp}{R^2(B)} \right| \leq \frac{|p \cdot p_3^\perp - q \cdot q_3^\perp|}{R^2(A)} + |q \cdot q_3^\perp| \frac{|R^2(B) - R^2(A)|}{R^2(A)R^2(B)} \leq$$

$$\leq 2\varepsilon \frac{R(A) + 2\sqrt{6}R(B)}{R(A)} + |q| \cdot |q_3^\perp| \frac{R^2(A) - R^2(B)}{R^2(A)R^2(B)} \leq 2\varepsilon(1 + 2\sqrt{6}) + R^3(B) \left( \frac{1}{R^2(B)} - \frac{1}{R^2(A)} \right).$$

We use $R(A) \leq R(B) + 2\varepsilon$ to bound last term:

$$R(B) \left( 1 - \frac{R^2(B)}{R^2(A)} \right) \leq R(B) \left( 1 - \frac{R^2(B)}{(R(B) + 2\varepsilon)^2} \right) \leq \frac{R(B)}{(R(B) + 2\varepsilon)^2} 4\varepsilon(R(B) + \varepsilon) \leq 4\varepsilon.$$

Then $\left| \dfrac{p \cdot p_3^\perp}{R^2(A)} - \dfrac{q \cdot q_3^\perp}{R^2(B)} \right| \leq 2\varepsilon(1 + 2\sqrt{6}) + 4\varepsilon < 16\varepsilon$. By Definition D.2, the Point-based Representation Metric PRM equals the maximum of

$$d_\infty = |R(p\{2\}) - R(q\{2\})| \leq 2\varepsilon, \quad |R(A) - R(B)| \leq 2\varepsilon, \quad d_\infty \leq 16\varepsilon, \quad \mathrm{BD} < 16\varepsilon,$$

so $\mathrm{PRM}\big(\mathrm{PR}(A; p\{2\}), \mathrm{PR}(B; q\{2\})\big) \leq 16\varepsilon$ which finishes the proof of part (b) for $n = 3$.

**Final subcase** $i = n > 3$. Assuming again that $R(A) \geq R(B)$, Lemmas E.1 and E.2 give

$$|p_n^\perp| \leq \sqrt{n}R^{n-1}(A), \quad |q_n^\perp| \leq \sqrt{n}R^{n-1}(B), \quad |p_n^\perp - q_n^\perp| \leq 2\varepsilon n(n-1)R^{n-2}(A) \text{ for any } n > 3.$$

We start estimating similarly to the case $i < n$.

$$|p \cdot p_n^\perp - q \cdot q_n^\perp| \leq |(p - q) \cdot p_n^\perp + q \cdot (p_n^\perp - q_n^\perp)| \leq |p - q| \cdot |p_n^\perp| + |q| \cdot |p_n^\perp - q_n^\perp| \leq$$
$$2\varepsilon \cdot \sqrt{n}R^{n-1}(A) + R(B) \cdot 2\varepsilon n(n-1)R^{n-2}(A).$$

Then $\left| \dfrac{p \cdot p_n^\perp}{R^{n-1}(A)} - \dfrac{q \cdot q_n^\perp}{R^{n-1}(B)} \right| \leq \dfrac{|p \cdot p_n^\perp - q \cdot q_n^\perp|}{R^{n-1}(A)} + |q \cdot q_n^\perp| \cdot \left| \dfrac{R^{n-1}(B) - R^{n-1}(A)}{R^{n-1}(A)R^{n-1}(B)} \right| \leq$

$$\leq \dfrac{2\varepsilon\sqrt{n}R^{n-1}(A) + 2\varepsilon n(n-1)R^{n-2}(A)R(B)}{R^{n-1}(A)} + |q| \cdot |q_n^\perp| \cdot \left| \dfrac{1}{R^{n-1}(A)} - \dfrac{1}{R^{n-1}(B)} \right| \leq$$

$$\leq 2\sqrt{n}\varepsilon + 2\varepsilon n(n-1) + \sqrt{n}R^n(B)\left( \dfrac{1}{R^{n-1}(B)} - \dfrac{1}{R^{n-1}(A)} \right).$$

We use $R(A) \leq R(B) + 2\varepsilon$ and the simpler notation $R = R(B)$ to bound last term after factorizing the difference of the $(n-1)$-st powers as follows:

$$R(B)\left( 1 - \dfrac{R^{n-1}(B)}{R^{n-1}(A)} \right) \leq R\left( 1 - \dfrac{R^{n-1}}{(R + 2\varepsilon)^{n-1}} \right) = R\dfrac{(R + 2\varepsilon)^{n-1} - R^{n-1}}{(R + 2\varepsilon)^{n-1}} =$$

$$= \dfrac{R(R + 2\varepsilon - R)}{(R + 2\varepsilon)^{n-1}} \sum_{j=0}^{n-2}(R + 2\varepsilon)^j R^{n-2-j} \leq \dfrac{2\varepsilon R}{(R + 2\varepsilon)^{n-1}} \sum_{j=0}^{n-2}(R + 2\varepsilon)^{n-2} \leq 2\varepsilon(n-1).$$

Then $\mathrm{BD}\left( \dfrac{M(A; p\{n-1\})}{R(A)}, \dfrac{M(B; q\{n-1\})}{R(B)} \right) \leq \left| \dfrac{p \cdot p_n}{R^{n-1}(A)} - \dfrac{q \cdot q_n}{R^{n-1}(B)} \right| \leq$

$2\varepsilon(\sqrt{n} + n(n-1) + \sqrt{n}(n-1)) = 2\varepsilon\sqrt{n}(1 + \sqrt{n}(n-1) + n - 1) \leq 2\varepsilon\sqrt{n}(\sqrt{n}(n-1) + n) =$

$2\varepsilon n(n + \sqrt{n} - 1) \leq 3\varepsilon n^2$ because $\sqrt{n} - 1 \leq \dfrac{n}{2}$. For $n = 4$, the upper bound above is $3\varepsilon(4)^2 >$

$6\varepsilon \geq d_\infty$. Hence the final upper bound is $\mathrm{PRM}\big(\mathrm{PR}(A; p\{n-1\}), \mathrm{PR}(B; q\{n-1\})\big) \leq 3\varepsilon n^2$. $\qquad\square$

**Lemma E.4** (Lipschitz continuity of BMD). *Let $\Gamma$ be a complete bipartite graph with a vertex matching $E$ such that any $e \in E$ has a weight $w(e) \leq \varepsilon$. Then $\mathrm{BMD}(\Gamma) \leq \varepsilon$.*

***Proof of Lemma E.4.*** By Definition 4.3, the vertex matching $E$ has the weight $W(E) = \max_{e \in E} w(e) \leq \varepsilon$. Since $\mathrm{BMD}(\Gamma) = \min_E W(E)$ is minimized for all matchings, $\mathrm{BMD}(\Gamma) \leq \varepsilon$. $\qquad\square$

The Lipschitz continuity of NDP in Theorem E.5 extends Theorem 5.1 to any $n \geq 2$ by using Proposition E.3 and Lemma E.4.

**Theorem E.5** (Lipschitz continuity of NBM). *Let a cloud $B \subset \mathbb{R}^n$ be obtained from a cloud $A \subset \mathbb{R}^n$ by perturbing every point of $A$ within its Euclidean $\varepsilon$-neighborhood. Then $\mathrm{NBM}(A, B) \leq \lambda_n\varepsilon$, where the Lipschitz constants are $\lambda_2 = 6$, $\lambda_3 = 16$, $\lambda_n = 3n^2$ for $n > 3$ as in Proposition E.3.*

***Proof of Theorem E.5.*** Order all vertices of the given clouds $A, B$ so that every point $p_i \in A$ has the same index as its $\varepsilon$-perturbation $q_i \in B$.

In Definition D.4, for any ordered points $p_1, \ldots, p_{n-1} \in A$, there are points $q_1, \ldots, q_{n-1} \in B$, which are $\varepsilon$-perturbations of $p_1, \ldots, p_{n-1}$, respectively, such that $\mathrm{PRM}(\mathrm{PR}(A; p_1, \ldots, p_{n-1}), \mathrm{PR}(B; q_1, \ldots, q_{n-1})) \leq \lambda_n\varepsilon$ by Proposition E.3. These PRMs are weights of edges in the index-preserving vertex matching $E$ of the complete bipartite graph $\Gamma(A; p_1, \ldots, p_{n-1}; B; q_1, \ldots, q_{n-1})$ for any $p_1, \ldots, p_{n-1}$ and their $\varepsilon$-perturbations $q_1, \ldots, q_{n-1}$. Then $\mathrm{BMD}(\Gamma(A; p_1, \ldots, p_{n-1}; B; q_1, \ldots, q_{n-1})) \leq \lambda_n\varepsilon$ by Lemma E.4. Since this conclusion holds for all (choices of) $p_1, \ldots, p_{n-1} \in C$, we iteratively apply this argument for the bipartite graphs $\Gamma(A; p_1, \ldots, p_k; B; q_1, \ldots, q_k)$ for $1 \leq k \leq n - 2$ and finally conclude that $\mathrm{NBM}(A, B) \leq \lambda_n\varepsilon$. $\qquad\square$

The upper bounds are higher than the real ratios NBM/BD in practical examples, see Fig. 5.

**Lemma E.6** (time of PR). *For any cloud $A \subset \mathbb{R}^n$ of $m$ unordered points, any point-based representation $\mathrm{PR}(A; p\{n-1\})$ in Definition C.2 needs $O(n^3 + mn)$ time.*

**Proof of Lemma E.6.** We find the center $O(A)$ and translate the cloud $A$ of $m$ points so that $O(A)$ becomes the origin $0 \in \mathbb{R}^n$ in time $O(m)$. We compute the $n \times n$ matrix $\mathrm{SD}(p_1, \ldots, p_{n-1})$ of squared distances between $p_0 = 0, p_1, \ldots, p_{n-1}$ in time $O(n^2)$. The vector $p_n^\perp$ from Lemma C.1 needs the $n \times n$ determinant computable in time $O(n^3)$. For any point $q \in A - \{p_1, \ldots, p_{n-1}\}$, the column of scalar products $q \cdot p_1, \ldots, q \cdot p_n$ needs $O(n)$ time. The $n \times (m-n+1)$ matrix $M(A; p\{n-1\})$ needs $O(mn)$ time. The point-based representation $\mathrm{PR}(A; p_1, \ldots, p_{n-1})$ in Definition C.2 needs $O(n^3 + mn)$ time. $\qquad\square$

**Lemma E.7** (time of PRM). *For any clouds $A, B \subset \mathbb{R}^n$ of $m$ unordered points with base sequences $p\{n-1\}$ and $q\{n-1\}$, respectively, the point-based representation Metric on the equivalences classes of $\mathrm{PR}(A; p\{n-1\})$ and $\mathrm{PR}(B; q\{n-1\})$ is found in time $O(n^2 + m^{1.5} \log^n m)$.*

**Proof of Lemma E.7.** The centers of masses $O(A), O(B)$ and radii $R(A), R(B)$ are computed in time $O(m)$.

The max metric $w_D$ between the $n \times n$ matrices in (D.2) needs $O(n^2)$ time. For the bottleneck distance $w_M(\sigma)$, the $n \times (m - n + 1)$ matrices of unordered columns are interpreted as fixed (not under isometry) clouds of $(m - n + 1)$ points in $\mathbb{R}^n$. Then $w_M$ can be computed in time $O(m^{1.5} \log^n m)$ by Theorem 6.5 in Efrat et al. (2001). $\qquad\square$

Assuming that $n^2 \leq O(m^{1.5} \log^n m)$, the time of PRM in Lemma E.7 becomes $O(m^{1.5} \log^n m)$.

Theorems E.8, E.9 extend Theorems 5.2, 5.3 for $n \geq 2$.

**Theorem E.8** (time of NDP). *For any cloud $A \subset \mathbb{R}^n$ of $m$ unordered points, the Nested Distributed Projection $\mathrm{NDP}(A)$ in Definition C.8 is computable in time $O(n^2 m^n)$.*

**Proof of Theorem E.8.** The given cloud $A$ has $\emptyset(m^{n-1})$ base sequences of $n - 1$ ordered points $p_1, \ldots, p_{n-1} \in A$. Lemma E.6 computes each $\mathrm{PR}(A; p_1, \ldots, p_{n-1})$ in time $O(n^3 + mn)$. By Definition C.8, the invariant $\mathrm{NDP}(A)$ consisting of $O(m^{n-1})$ point-based representations can be computed in time $O(n^2 m^n)$ because $n \leq m$. $\qquad\square$

**Theorem E.9** (time of NBM). *For any clouds $A, B \subset \mathbb{R}^n$ of $m$ unordered points, the Nested Bottleneck Metric $\mathrm{NBM}(A, B)$ in Definition D.4 can be computed in time $O(m^{2n-0.5} \log^n m)$. If $n = 2$, the time is $O(m^{3.5} \log m)$.*

**Proof of Theorem E.9.** In Definition D.4, for any fixed $1 \leq k \leq n - 1$ and ordered points $p_1 \ldots, p_{k-1} \in A$ and $q_1 \ldots, q_{k-1} \in B$, the bipartite graph $\Gamma(A; p_1, \ldots, p_{k-1}; B; q_1, \ldots, q_{k-1})$ has $V = 2(m - k + 1) = O(m)$ vertices and $E = (m - k + 1)^2 = O(m^2)$ edges.

For $k = n - 1$, the weight $w(e)$ of each edge $e$ equals PRM, which needs time $O(m^{1.5} \log^n m)$ by Lemma E.7. For all $O(m^2)$ edges of $\Gamma(A; p_1, \ldots, p_{n-2}; B; q_1, \ldots, q_{n-2})$, the time is $O(m^{3.5} \log^n m)$. The bottleneck matching distance BMD for such a graph is computed by Hopcroft & Karp (1973) in time $O(E\sqrt{V}) = O(m^{2.5})$, which is dominated by the time $O(m^{3.5} \log^n m)$ preparing the weights.

For all $O(m^{n-2})$ choices of ordered points $p_1, \ldots, p_{n-2} \in A$ and all $O(m^{n-2})$ choices of $q_1, \ldots, q_{n-2} \in B$, the Bottleneck Matching Distances for all graphs $\Gamma(A; p_1, \ldots, p_{n-2}; B; q_1, \ldots, q_{n-2})$ are computed in time $O(m^{2(n-2)} m^{3.5} \log^n m) = O(m^{2n-0.5} \log^n m)$.

For any next iteration $k = n - 2, \ldots, 1$ in Definition D.4, the parameter $k$ goes down by 1 and the exponent of $m$ drops by 2 each time. The sum over $k = n - 1, \ldots, 1$ is dominated by the time $O(m^{2n-0.5} \log^n m)$ of the first iteration.

For $n = 2$, the bottleneck distance between fixed $m$-point clouds in $\mathbb{R}^2$ can be computed in time $O(m^{1.5} \log m)$ without an extra logarithm by Theorem 6.5 from Efrat et al. (2001), which simplifies the time to $O(m^{3.5} \log m)$. $\qquad\square$

Theorem E.9 improves the time $O(m^{3(n-1)} \log m)$ of another metric on rigid classes of unordered point clouds from Theorem 4.7(b) in Widdowson & Kurlin (2023).

***Proof of Theorem* 5.4.** As usual, we shift both centers of mass $O(A), O(B)$ to the origin $0 \in \mathbb{R}^2$. By Definition 4.4, the distance $d = \mathrm{NBM}(A, B)$ is the Bottleneck Matching Distance $\mathrm{BMD}(\Gamma(A, B))$ computed in time $O(m^{3.5} \log m)$ by Theorem 5.3. Here $\Gamma(A, B)$ is the complete bipartite graph on $m + m$ vertices represented by $\mathrm{PR}(A; p)$ and $\mathrm{PR}(B; q)$ for all points $p \in A$ and $q \in B$.

By Definition 4.3, $\mathrm{BMD}(\Gamma(A, B))$ equals the maximum weight $w(e)$ of an edge $e$ in a vertex matching $E$ of $\Gamma(A, B)$, which can be considered a bijection between the $m$-point clouds $A \to B$. For any pair $e = (p, p')$ of matched points, the weight $w(e)$ is $\mathrm{PRM}(\mathrm{PR}(A; p), \mathrm{PR}(B; p'))$.

The distance $\mathrm{NBM}(A, B) = \delta \geq w(e)$ is an upper bound for $|R(A) - R(B)|$, where $R(A) = \max_{p \in A} |p|$ and $R(B) = \max_{p' \in B} |p'|$. Choose a point $p \in A$ with $|p| = R(A)$ and the positive $x$-axis in $\mathbb{R}^2$ through $p' \in B$ matched with $p$ via $E$. Let $f$ be the rotation of $\mathbb{R}^2$ around $0$ such that $f(p)$ is also in the positive $x$-axis. By Definition 4.2, $f(p), p'$ in the $x$-axis have lengths satisfying $|p| = |f(p)|$, $||p| - |p'|| \leq d$ and hence are $d$-close: $|f(p) - p'| \leq d$.

It suffices to show that the image $f(q)$ of any other point $q \in A - \{p\}$ is $3\sqrt{2}d$-close to a unique point $q' \in B$ that we will find below. Since all distances and scalar products are preserved under $f$, we use the matrix $M(f(A); f(p))$ instead of $M(A; p)$ in computing PRM. Each column of $\dfrac{M(f(A); f(p))}{R(A)}$ consists of $\dfrac{f(q) \cdot f(p)}{|R(A)|}, \dfrac{f(q) \cdot f(p^\perp)}{|R(A)|}$, where $f(p) = (|p|, 0)$, $f(p^\perp) = (0, |p|)$, $R(A) = |p|$.

The distance $\mathrm{BD}\left( \dfrac{M(f(A); f(p))}{R(A)}, \dfrac{M(B; q)}{R(B)} \right) \leq d$ guarantees that the above column is $d$-close to the column of $\dfrac{q' \cdot p'}{|R(B)|}, \dfrac{q' \cdot p'^\perp}{|R(B)|}$ for a point $q' \in B$ determined by computing the bottleneck distance BD above. For the first scalar products involving $p, p'$, we have $\left| \dfrac{f(q) \cdot f(p)}{R(A)} - \dfrac{q' \cdot p'}{R(B)} \right| \leq \delta$, where the first fraction is the $x$-coordinate of $f(q)$.

To get the $x$-coordinate $\dfrac{q' \cdot p'}{|p'|}$ of the point $q' \in B$, where $|p'|$ is $\delta$-close to $R(A) = |p|$, use the triangle inequality:

$$\left| \frac{f(q) \cdot f(p)}{R(A)} - \frac{q' \cdot p'}{|p'|} \right| \leq \left| \frac{f(q) \cdot f(p)}{R(A)} - \frac{q' \cdot p'}{R(B)} \right| +$$

$$+ \left| \frac{q' \cdot p'}{R(B)} - \frac{q' \cdot p'}{|p'|} \right| \leq d + \frac{|q' \cdot p'|}{R(B)|p'|} |R(B) - |p'|| \leq$$

$$d + \frac{|q'| \cdot |p'|}{R(B)|p'|} |R(B) - |p'|| = d + \frac{|q'|}{R(B)} |R(B) - |p'|| \leq$$

$$d + |R(B) - |p'|| \leq d + |R(B) - |p|| + ||p| - |p'|| \leq$$

$$2d + |R(B) - |p|| = 2d + |R(B) - R(A)| \leq 3d.$$

Then the $x$-coordinates of $f(q) \in f(A)$ and $q' \in B$ differ by at most $3d$. Applying the same arguments to the scalar products involving the orthogonal vectors $p^\perp, p'^\perp$, which have the same lengths as $p, p'$, respectively, conclude that the $y$-coordinates of $f(q), q'$ also differ by at most $3d$. So $|f(q) - q'| \leq \sqrt{(3d)^2 + (3d)^2} = 3\sqrt{2}d$, set $\beta(q) = q'$. $\qquad\square$

**Corollary E.10** (continuous morphing). *Any clouds $A, B \subset \mathbb{R}^n$ of $m$ unordered points can be 'morphed' into each other in time $O(m^{2n-0.5} \log^n m)$ by inverting a continuous path between the complete invariants $\mathrm{NDP}(A), \mathrm{NDP}(B)$ in the space $\mathrm{NDP}(\mathrm{CRS}(\mathbb{R}^n; m))$ of realizable invariants.* ∎

***Proof of Corollary* E.10.** The 'morphing' is realized for rigid classes, so we will first rigidly move $A, B$ to convenient positions and only after that deform one cloud into another along a straight-line path inverted from the moduli space $\mathrm{NDP}(\mathrm{CRS}(\mathbb{R}^n; m))$. As usual, we translate $A, B$ so that their centers of mass are at the origin $0 \in \mathbb{R}^n$.

Theorem E.9 in time $O(m^{2n-0.5} \log^n m)$ computes the Nested Bottleneck Metric $\mathrm{NBM}(A, B)$ giving a bijection between all point-based representations $\mathrm{PR}(A; p_1, \ldots, p_{n-1}) \leftrightarrow \mathrm{PR}(B; p'_1, \ldots, p'_{n-1})$.

Choose any ordered points $p_1, \ldots, p_{n-1}$, which define their matched points $p'_1, \ldots, p'_{n-1} \in B$. For example, we could choose $p_1 \in A$ as the most distant point from the origin $0$, then $p_2$ as the most distant point to the line through $0, p_1$, and so on. Now we rotate $A$ so that $p_1$ lies in the positive 1st coordinate axis of $\mathbb{R}^n$, then rotate $A$ again so that $p_2$ lies in the positive half-plane of the first two coordinates axis of $\mathbb{R}^n$, and so on until $p_1, \ldots, p_{n-1}$ are fixed.

We similarly rotate $B$ to fix the positions of $p'_1, \ldots, p'_{n-1}$, which intuitively should become close to the already fixed positions of $p_1, \ldots, p_{n-1}$. Theorem 5.4 proves an explicit bound of the closeness for $n = 2$.

The computation of the bottleneck distance $\mathrm{BD}\left(\dfrac{M(A; p_1, \ldots, p_{n-1})}{R(A)}, \dfrac{M(B; p'_1, \ldots, p'_{n-1})}{R(B)}\right)$ within the same time of $\mathrm{NBM}(A, B)$ provides a bijection between the remaining points: $A - \{p_1, \ldots, p_{n-1}\} \leftrightarrow B - \{p'_1, \ldots, p'_{n-1}\}$.

According to this bijection, we index the corresponding points and columns of $M(A; p_1, \ldots, p_{n-1})$ and $M(B; p'_1, \ldots, p'_{n-1})$ by $n, \ldots, m$. We connect all matched points $p_i \leftrightarrow q_i$, $i = 1, \ldots, m$, by the straight-line segment $p_i(t) = (1 - t)p_i + tq_i$ in $\mathbb{R}^n$, where $t \in [0, 1]$ is a time parameter. So $A$ 'morphs' into $B$ via the continuous family of intermediate clouds $A(t) = \{p_1(t), \ldots, p_m(t)\}$, $t \in [0, 1]$.

By Theorem E.5, the images $\mathrm{NDP}(A(t))$ for $t \in [0, 1]$ form a continuous path whose every point (invariant value) is reconstructable back to the cloud $A(t)$. □

Thank you for reading all the proofs!

