# OpenReview forum: "Bi-continuous and complete SE(2)-invariants parametrize all clouds of unordered points"
_ICLR.cc/2025/Conference — Submitted to ICLR 2025_

### Official Review · Reviewer_mWsb · 2024-11-01

**Soundness:** 3
**Presentation:** 2
**Contribution:** 2
**Rating:** 3
**Confidence:** 3

**Summary:**

The paper addresses the problem of the exact matching of full (not partially observed, as in e.g. CV/LIDAR data) point clouds. It proposes a descriptor (NDP/NCP) that can uniquely identifies a specific point cloud and from which a unique point cloud can be reconstructed, as well as a distance metric (NBM) between point clouds that permits exact identification and is proven to be Lipschitz-smooth. The descriptor requires set matching for point cloud matching though - it cannot be directly compared by comparing descriptor vectors. The main paper discusses the 2D case; general dimensions (as well as all of the proofs) are presented in the appendix.

The writing of the paper is quite technical, so I might be missing something important somewhere. To the best of my understanding, the whole method hinges on the well-known fact that one needs only $d$ point-to-point correspondences (with linearly independent difference vectors) to fix a rigid mapping in $d$ dimensions (because a rigid coordinate frame can be derived from these), which makes searching for correspondences under rigid mappings much easier than general correspondence problems.

Naively, one could fix one reference point (e.g., the center of mass) and try correspondences between one (in 2D, two in 3D, etc.) of the source and every single target points (in 2D, all possible pairs in 3D, etc.) to find the mapping and efficiently check whether the clouds match. The complexity is polynomial for each fixed $d$ and exponential in $d$.

The paper exploits this mathematical fact (not sure if implicitly or explicitly) and builds systematically enumerates all coordinate frames in a single point cloud by using the center of mass and every point in the cloud as correspondences. The center of mass must match, thus only all other points have to be considered. It projects into the coordinate system induced by the difference to the center (scaled by the length of this vector to maintain smoothness) and then enumerates all of the points, giving an $O(n^2)$ representation of $n$ points (there are $O(n)$ coordinate frames with $O(n)$ entries each). The descriptor is then the set of the set of points. By comparing these via set-matching, exact matches can be found in polynomial time.

To compute distances, a cascade of two perfect bi-partite graph matching steps is used to first reconstruct a good ordering of the rotated and scaled copies of the point cloud and then match their points as much as possible. The paper states a theorem and provides a proof for Lipschitz-continuity of this process (which involves discrete decisions).

Finally, the paper shows some examples of comparing point clouds using the QM-9 data set as an example.

**Strengths:**

The paper seems to provide a solution for the rigid point cloud comparison problem that addresses all of the formal requirements that one would probably desire in theory and practice (with the exception, as far as I understand, of encoding the descriptor as a vector in a linear vector space with simple $L_p$ distance and literal equality computations, which would be useful for efficient querying in geometric data bases).
It makes great effort in formalizing requirements and formally proving the results, and gives a comprehensive account of a broad set of earlier work. It also provides examples to illustrate the challenges of distance-based rigidity.
If my understanding of the method outlined above is correct, I would also see the method as in its core rather straightforward and easy to understand, which increases confidence in correctness and might make it easier to implement the ideas in other contexts.

**Weaknesses:**

My understanding of potential weaknesses hinges upon my understanding of the method as a whole; as the paper is technical and extensive, I might be misunderstanding something.

A major restriction of the submission is its focus on exact matching of full point clouds, excluding partially acquired data and significant outlier issues, both of which is very common in practical handling of, for example, 3D point clouds acquired from real-world scenes (for cases like molecular data bases, such as QM-9, this restriction probably does not matter). The exact matching is relaxed by providing a continuous metric, but partial matching still remains impossible. The method is also restricted to rigid motions, and due the exponential costs in $d$, it can only be realistically applied to low-dimensional data.

I think that the exact matching algorithm, which is described first, is not very useful: First, it is not new - guessing a small number of point-wise correspondences, e.g. via RANSAC or randomized forward-search, and then verifying the match is an established technique, see e.g. [Huang et al., Reassembling Fractured Objects by Geometric Matching, Siggraph 2006], which is already robust to deteriorated and partial data. Second, it requires a large amount of memory (squared the number of input points) and accordingly significant runtimes, which can be avoided by not storing the intermediate results and using heuristics filters in matching candidates (such as Huang et al.'s and many similar papers), as well as by implicitly excluding impossible matching candidates (see e.g. 4PCS-Matching [Aiger et al., Siggraph 2008]).

The continuous metric, on the other hand, is a bit more interesting. While the basic procedure (using bi-partite graph matching to find the closest matching points) seems again to be a canonical path for comparing full point clouds, the design of two nested matching procedures does not appear very straightforward, and the smoothness guarantees obtain in that way might be appealing in applications where such a tool is needed. Unfortunately, I have not been able to verify the formal proof, but the intuitively, the I would assume that this should work (as nearest points change discretely, but only at points where distances coincide, thus plausibly maintaining smoothness).

The writing of the paper could be improved. While I would see the strive for a formal treatment positively, the paper in its current form, to a reader like me (different people have of course different inclinations to intuition vs. abstraction) seems to obscure the main ideas behind formal details to much. Then, the appendix, where a lot of the most interesting contributions reside (such as proofs of the theorems), is even harder to parse. For example, I was looking the proof of Theorem 3.5, where I would see a minor technical difficulty but could only find a treatment of a much more general statement in arbitrary dimensions, which made it very difficult to verify the statements made in the paper. In my perception, one could reorganize and rephrase the text to make everything much more accessible. Overall, I have to say that I was not able to read and understand much of the appendix in its current form (maybe fellow reviewers who were able to can give a better assessment of the overall work).

Finally, I see a small technical issue in Theorem 3.5: The representation computed for the NDP/NCP descriptors consists of a set of matrices (in a tuple with the base-point distance), which are compared by equality. The matrices are descriped as unordered in terms of the ordering of the 2D points (after projection, which is of course ordered by the two coordinate axes). Comparing sets then implicitly matches the unordered matrices, restoring correspondences between rearranged points. However, within the matrices, a "standard" equality / comparison would perform a column-wise comparison, which matches points in the specified order, which can differ between point clouds A and B to be matched. I think that here, again, a set comparison rather than a column-wise matrix comparision would be required. The continuous metric uses two nested linear assignment / bi-partite graph matching steps, the inner one corresponding to such a (soft) set matching here.

Overall, I think that the paper is (most likely, limited by my ability to follow the appendix and some remaining doubts wrt. potential misunderstandings) correct in essence and solves the problem that it states. However, the conceptual step over known work seems a bit narrow to me (given the limited setup and that we do not get vector-valued invariants but only an encoding of point matching / finding of closest points in mathematical language), potentially with the exception of proving continuity. However, that part was hard to understand. The set of practical problems that can be solved with the new proposal but not with already known methods also seems to be limited (in particular given the large costs). The problem statement also seems to be to some degree tangential to the focus of ICLR.

For these reasons, I am somewhat skeptical of the suitability of the submission for ICLR.

**Questions:**

First of all, it might be possible that I fundamentally misunderstand how all of this work - then it would be important to rectify this. If not, I would be curious about the matrix-matching problem discussed above.

---

> ### Author Response · Authors · 2024-11-15
> **Thank you for the detailed review**
>
> >To the best of my understanding, the whole method hinges on the well-known fact that one needs only d point-to-point correspondences (with linearly independent difference vectors) to fix a rigid mapping in d dimensions (because a rigid coordinate frame can be derived from these), which makes searching for correspondences under rigid mappings much easier than general correspondence problems.
>
> You are right that d point correspondences certainly help if they are given. However, these d points should span a (d-1) dimensional subspace in R^d, which may not be guaranteed for an arbitrary cloud. Otherwise a rigid frame is not defined for the full d-dimensional space.
>
> >one could fix one reference point (e.g., the center of mass) and try correspondences between one (in 2D, two in 3D, etc.) of the source and every single target points (in 2D, all possible pairs in 3D, etc.) to find the mapping and efficiently check whether the clouds match. The complexity is polynomial in d for each fixed and exponential in d. The paper exploits this mathematical fact (not sure if implicitly or explicitly) and builds systematically enumerates all coordinate frames in a single point cloud by using the center of mass and every point in the cloud as correspondences.
>
> Thank you for correctly summarizing the construction of the invariant NDP in section 3, which explicitly uses the center of mass one base point p in Definitions 3.1 and 3.4.
>
> >The paper seems to provide a solution for the rigid point cloud comparison problem that addresses all of the formal requirements that one would probably desire in theory and practice (with the exception, as far as I understand, of encoding the descriptor as a vector in a linear vector space with simple distance and literal equality computations, which would be useful for efficient querying in geometric data bases).
>
> The hierarchical nature of the invariants SRV, SDV, PDD, NDP and the polynomial-time metric of the complete invariants NDP already suffices to query geometric databases due to many nearest neighbor algorithms that work in (roughly) near-linear time. Because the invariant spaces are obtained from compact regions of a Euclidean space by boundary identifications, see details and examples in Appendix B, these spaces can be embedded into a higher-dimensional Euclidean spaces, where other L_p metrics can be computed faster. We postpone this embeddings for future work because the current paper is long enough.
>
> >It makes great effort in formalizing requirements and formally proving the results, and gives a comprehensive account of a broad set of earlier work. It also provides examples to illustrate the challenges of distance-based rigidity.
>
> Thank you for appreciating the effort in formalizing all results and providing illustrative examples.
>
> >If my understanding of the method outlined above is correct, I would also see the method as in its core rather straightforward and easy to understand, which increases confidence in correctness and might make it easier to implement the ideas in other contexts.
>
> Your understanding is generally correct. We would add that the Lipschitz continuity.
>
> >A major restriction of the submission is its focus on exact matching of full point clouds, excluding partially acquired data and significant outlier issues, both of which is very common in practical handling of, for example, 3D point clouds acquired from real-world scenes (for cases like molecular data bases, such as QM-9, this restriction probably does not matter).
>
> You are right that molecules have no atomic outliers because all atoms vibrate close to their average positions. Problem 1.2 asking for a bi-continuous complete invariant of point clouds was already hard without efficient solutions even for m=4 points in the plane, illustrated by Example 4.5 and Figure 4.
>
> For clouds of different sizes, lines 533-537 mentioned that "In other applications, for clouds with different numbers of points, we can replace the bottleneck distance BD in Definition 4.2 with any metric between fixed clouds of different sizes, e.g. the Hausdorff distance, to get a metric on PRs. Then we can compare NDPs of any clouds as weighted distributions by EMD."
>
> If we need to detect a common rigid subset in two larger clouds, this problem is also important and includes Problem 1.1 as a partial case, which motivates solving Problem 1.1 first.
>
> > The method is also restricted to rigid motions
>
> NDP is adjustable for isometry and compositions with uniform scaling. Lines 284-285: "The completeness of NDP(A) under rigid motion in Theorem 3.5 implies the completeness of the pair NDP(A), {overline NDP}(A) under isometry including reflections." Line 308: "One more division by R(A) makes metrics invariant under uniform scaling." Lines 527-529 conclude that "Problem 1.1 was stated for unordered clouds under rigid motion but was also solved for isometry and compositions of these equivalences with uniform scaling in R^2". We continue below.

---

> > ### Author Response · Authors · 2024-11-15
> > **Continuation of the response above**
> >
> > >due the exponential costs in d, it can only be realistically applied to low-dimensional data.
> >
> > The hierarchy of the faster invariants SRV, SDV, PDD, whose computations are near-linear or quadratic in the cloud size for any dimension, allows us to use the complete invariants only for rare cases when the past incomplete invariants cannot distinguish point clouds. This hierarchical approach is a practical way to resolve the curse of dimensionality.
> >
> > >I think that the exact matching algorithm, which is described first, is not very useful: First, it is not new - guessing a small number of point-wise correspondences, e.g. via RANSAC or randomized forward-search, and then verifying the match is an established technique, see e.g. [Huang et al., Reassembling Fractured Objects by Geometric Matching, Siggraph 2006]
> >
> > We are happy to add the described strengths of the recommended references to the review of point registration methods in lines 113-122. Problem 1.1 goes beyond exact matching, so the words "point matching" were used in the sense of approximate matching in condition (1.1e) and Theorem 5.4: if the complete invariants of given clouds are close, these clouds can be matched by rigid motion up to a guaranteed perturbation.
> >
> > >it requires a large amount of memory (squared the number of input points)
> >
> > The faster invariant SRV (Sorted Radial Vector) has a linear size.
> >
> > >by not storing the intermediate results and using heuristics filters in matching candidates (such as Huang et al.'s and many similar papers), as well as by implicitly excluding impossible matching candidates (see e.g. 4PCS-Matching [Aiger et al., Siggraph 2008]).
> >
> > Thank you for recommending these memory-saving tips and references, which we will certainly include in the bibliography.
> >
> > >the design of two nested matching procedures does not appear very straightforward, and the smoothness guarantees obtain in that way might be appealing in applications where such a tool is needed.
> >
> > Thank you for highlighting this non-trivial contribution, which was needed for the inverse Lipschitz continuity.
> >
> > >I have not been able to verify the formal proof, but the intuitively, the I would assume that this should work (as nearest points change discretely, but only at points where distances coincide, thus plausibly maintaining smoothness).
> >
> > Yes, you are right that nearest neighbors change discretely only in cases of neighbors at exactly the same distance.
> >
> > >While I would see the strive for a formal treatment positively
> >
> > Yes, we consider the rigorous definitions and rigorous proofs as the main strength because going from the already proved theorems to informal explanations in simpler language seems easier than from vague ideas to well-justified results.
> >
> > > I was looking the proof of Theorem 3.5, where I would see a minor technical difficulty but could only find a treatment of a much more general statement in arbitrary dimensions, which made it very difficult to verify the statements made in the paper.
> >
> > Theorem 3.5 has a long proof to cover all exceptional cases, e.g. when a point cloud belongs to a low-dimensional subspace of R^n. The exceptions appear even in R^2 for a cloud within a straight line. Hence the 2D version would not be much shorter.
> >
> > >In my perception, one could reorganize and rephrase the text to make everything much more accessible. Overall, I have to say that I was not able to read and understand much of the appendix in its current form (maybe fellow reviewers who were able to can give a better assessment of the overall work).
> >
> > We are happy to answer any questions.
> >
> > >I see a small technical issue in Theorem 3.5: The representation computed for the NDP/NCP descriptors consists of a set of matrices (in a tuple with the base-point distance), which are compared by equality. The matrices are descriped as unordered in terms of the ordering of the 2D points (after projection, which is of course ordered by the two coordinate axes). Comparing sets then implicitly matches the unordered matrices, restoring correspondences between rearranged points. However, within the matrices, a "standard" equality / comparison would perform a column-wise comparison, which matches points in the specified order, which can differ between point clouds A and B to be matched. I think that here, again, a set comparison rather than a column-wise matrix comparision would be required.
> >
> > Yes, we use a set comparison for columns, not a column-wise comparison, which is mentioned in lines 303-304 of the main paper: "The fast algorithm for BD above can compare any 2×(m−1) matrices M(A; p) and M(B; q) as fixed clouds of unordered columns (points in R^2)." Here the keyword *unordered* means that the matrix is treated as a cloud of unordered points representing unordered columns.
> >
> > >The continuous metric uses two nested linear assignment / bi-partite graph matching steps, the inner one corresponding to such a (soft) set matching here.
> >
> > Yes, you are absolutely right.
> >
> > We continue below

---

> ### Author Response · Authors · 2024-11-15
> **Continuation of the response above**
>
> >Overall, I think that the paper is (most likely, limited by my ability to follow the appendix and some remaining doubts wrt. potential misunderstandings) correct in essence and solves the problem that it states.
>
> Thank you for your helpful assessment.
>
> >the conceptual step over known work seems a bit narrow to me (given the limited setup and that we do not get vector-valued invariants but only an encoding of point matching / finding of closest points in mathematical language), potentially with the exception of proving continuity.
>
> We can get vector-valued invariants by embedding the invariant spaces (described by explicit inequalities on point coordinates in appendix B) but the invariants should be described before these embeddings, so Problem 1.1 cannot be skipped.
>
> Please notice that Problem 1.1 was essentially open even for m=4 points in the plane but now solved for any number m of points in all dimensions n.
>
> >The set of practical problems that can be solved with the new proposal but not with already known methods also seems to be limited (in particular given the large costs).
>
> The proposed invariants finalize the hierarchy of the faster incomplete invariants, which can solve large-scale problems in a hierarchical way as described in the experiments on QM9 in section 6.
>
> >The problem statement also seems to be to some degree tangential to the focus of ICLR.
>
> The invariant-based representations of point clouds nicely fit the ICLR conference title "International Conference on Learning Representations". This paper is relevant to at least two topics at https://www.iclr.cc/Conferences/2025/CallForPapers
>
> unsupervised, self-supervised, semi-supervised, and supervised representation learning
>
> applications to physical sciences (physics, chemistry, biology, etc.)
>
> If there are any more questions, we would be happy to answer.

---

> > ### Comment · Reviewer_mWsb · 2024-11-29
> >
> > Dear authors - thanks for the detailed feedback. I am happy to see my overall understanding of the technical approach being confirmed and I believe that there are no missunderstandings left to be addressed.

---

> > > ### Author Response · Authors · 2024-11-29
> > > **Thank you for the reply**
> > >
> > > Dear Reviewer mWsb,
> > >
> > > > thanks for the detailed feedback. I am happy to see my overall understanding of the technical approach being confirmed and I believe that there are no missunderstandings left to be addressed.
> > >
> > > Thank you for your helpful confirmation that all misunderstandings have been addressed. Can we expect an update of the overall assessment? Thank you again.

---

> > > > ### Comment · Reviewer_mWsb · 2024-11-29
> > > >
> > > > Dear authors, I do not see any major misunderstandings; we basically agree on all of the major aspects. So I do not see a reason to change my rating. Even though some other reviewers have been more positive, I do not believe that there is any misunderstanding of the facts here, but rather a matter of how to put weight on drawbacks and benefits of a submission. Different researchers have different perceptions here (which is obviously a good thing), but I nonetheless have not seen any facts that would alter my personal assessment.

---

> > > > > ### Author Response · Authors · 2024-11-29
> > > > > **different perceptions**
> > > > >
> > > > > Dear Reviewer mWsb, would it be possible please to clarify what aspects of the paper are still considered drawbacks? We are asking only about Problem 1.1, which was open even in the plane starting from 4 points and is now solved in any dimension with important applications to molecules, not about further questions such as partial matchings beyond the scope of Problem 1.1? Thank you.

---

### Official Review · Reviewer_CoFa · 2024-11-02

**Soundness:** 4
**Presentation:** 3
**Contribution:** 3
**Rating:** 6
**Confidence:** 3

**Summary:**

The paper studies unordered points-clouds in n-dimensional Euclidean space, under the equivalence of rigid motion (translation and rotation).
The main idea, which extends previous limited attempts, is in defining a *complete invariant*, which is a function that maps an m-point point-cloud to a point in a higher dimemsional space, over which a metric is defined such that point-coulds are mapped to a single point if and only if they are equivalent. In addition, the invariant is proven to have several other desired properties, such as bi-directional Lipschitz continuity and efficient ways to compute both directions of the mapping as well as and an approximate rigid motion for point-clouds that are close to being equivalent.
The requirements of the invariant are stated in a main Problem, followed by a detailed construction, with full proofs for the claimed properties (in the appendix).
An experimental section focuses on analyzing a large-scale dataset of 3D molucule structures, with the focus being on differentiating the proposed NDP from previous invariants (which are generally much faster to compute, but are not complete - i.e. they might map non-equivalent clouds to a single point).

**Strengths:**

1] The paper is well written - I like the way that the limitations in previous work are clearly presented, including by simple examples, and the decision to present the ideas in 2D and defer the proofs to the appendix. Writing is very accurate, but at the same time accessible to non-experts.
2] I believe that the new characterization (in Problem 1.1) of the conditions required from the parameterization captures different desired properties of such invariants in a good way, including the addition of properties like 'point-matching' and 'realizability' that were lacking in previous definitions. This definition, includes some properties, which were not demonstrated to be useful, but might be so in different applications. Especially, the bi-directional continuity, which gives a type of smoothness that might be useful for settings where one is interested with approximate forms of equivalence.
3] The discussions and examples for the 2D case of m=3 (triangles) and m=4 (quads) is very helpful for understanding both the limitations of previous work as well as the specifics of the construction, including very good visualizations.
4] The experiment is very original and interesting. It is done in a clever way, by hierarchically classifying pairs as being different, by gradually increasing the quality of the invariant. It is nicely shown on data that is real, even though not including measurement noise, with true new conclusions (that I presume were unknown) that could not be made using the reference techniques.

**Weaknesses:**

1] While the suggested parameterization is an elegant mathematical result, with clear progress made, it's applicability is still questionable. Computationally, quadratic complexity is clearly prohibitive, for most cases of 'point-clouds', or general n-dimensional sets (e.g. of features). Perhaps the focus should have been more on the computational sides of the problem. I would also argue that the *completeness* of the invariance is rarely relevant when dealing with real data, unless it is the case of differentiating between exact\inexact copies. On the other hand, the Lipschitz bi-continuity could be a key property for dealing with the inexact versions of the problem, which are much more general and common. Think of point-clouds that have noise-measurement, or that have resulted from different ways of sampling. Furthermore, such distaces could be used to measure the difference between discretely sampled disributions, under optimal alignment.
2] In the same lines, the experiment focuses on demonstrating the advantage of being a complete invariant, but does not analyze the advantages of having the other properties. I understand that the theory wasn't fully developed in such directions (e.g. assuming equal number of points), but it would have been useful to empirically demonstrate the merits of the invariant in other ways, even under controlled synthetic experiments.
3] There is a lack in addressing how the proposed idea could be beneficial for the more common tasks that are performed on unordered point-clouds, such as: alignment (perhaps with noise or partial overlap), classification, detection, segmentation and even generative modeling.

Some typos/mistakes:
* Wrong referencing, e.g. in lines 81 and 86
* bipartute (line 321)
* 'one four pairs' (line 336)
* sayss (line 411)

**Questions:**

1] I'm note sure whether there is a clear charcterization regarding what are the cases in which one would decide to use the suggested NDP, over the simpler and cheaper alternatives. Is there a principled way the you would suggest? The 'hierarchical' way used seems to only be suitable for the particular case of classifying data by exact equivalence (where the cheaper invariants provide fast 'rejection').
2] It is claimed in the experiment that the NDP provides best seperation. Is there any evidence that the metric on it is well aligned with what would be considered a good measure for the distance from being equivalent (under rigid transformation)? Have you tried testing how it looks when starting from perfect equivalence and degrading in different ways (such as noise in the points, or non-perfect rigidity)?
3] Computation wise - Are there relaxations with major gains that could make the approach more practical? Such as computing smoother (non permutation-like 1-1 mappings) matchings?

---

> ### Author Response · Authors · 2024-11-15
> **Thank you for your detailed and thorough review**
>
> >The paper is well written - I like the way that the limitations in previous work are clearly presented, including by simple examples, and the decision to present the ideas in 2D and defer the proofs to the appendix. Writing is very accurate, but at the same time accessible to non-experts.
>
> Thank you for appreciating the effort to make the paper accessible to non-experts.
>
> >I believe that the new characterization (in Problem 1.1) of the conditions required from the parameterization captures different desired properties of such invariants in a good way, including the addition of properties like 'point-matching' and 'realizability' that were lacking in previous definitions. Especially, the bi-directional continuity, which gives a type of smoothness that might be useful for settings where one is interested with approximate forms of equivalence.
>
> Yes, these new conditions in Problem 1.1(d,e) were important to guarantee for applications in molecular design because they allow anyone to sample the proposed invariant space with 100% probability and reconstruct a point cloud (uniquely under rigid motion) from a new value of the invariant. In the past, latent spaces based on incomplete or discontinuous invariants did not allow this guarantee continuous and well-defined inversion from invariants back to real point clouds.
>
> >The discussions and examples for the 2D case of m=3 (triangles) and m=4 (quads) is very helpful for understanding both the limitations of previous work as well as the specifics of the construction, including very good visualizations.
>
> Yes, we were surprised ourselves that the case of m=4 points in the plane was essentially open since ancient times when the side-side-side theorem parametrized all rigid triangles by three ordered inter-point distances satisfying one triangle inequality.
>
> >The experiment is very original and interesting. It is done in a clever way, by hierarchically classifying pairs as being different, by gradually increasing the quality of the invariant. It is nicely shown on data that is real, even though not including measurement noise, with true new conclusions (that I presume were unknown) that could not be made using the reference techniques.
>
> Yes, we have not seen any past reports of all-vs-all comparisons of molecules in large databases such as QM9. Even if we split all molecules by the simplest invariant (number of atoms), QM9 has 873,527,974 pairs of clouds with the same number of atoms.
>
> >While the suggested parameterization is an elegant mathematical result, with clear progress made, it's applicability is still questionable. Computationally, quadratic complexity is clearly prohibitive,
>
> The first invariant SRV (Sorted Radial Vector) has a near-linear time, also a linear-time for a metrics on these vectors. The SRV invariant can be further restricted to a smaller number of (say) first few components for even faster comparisons.
>
> > I would also argue that the completeness of the invariance is rarely relevant when dealing with real data, unless it is the case of differentiating between exact\inexact copies
>
> The past invariants SRV, SDV, PDD are based on distances and cannot distinguish any mirror images. However, chiral molecules (not-mirror-symmetric) can have different functional properties and hence are important to distinguish for drug design. Several Nobel prizes were given for such chiral molecules: https://www.nobelprize.org/prizes/chemistry/2021/summary/
> https://www.nobelprize.org/prizes/chemistry/2001/popular-information/
>
> >the Lipschitz bi-continuity could be a key property for dealing with the inexact versions of the problem, which are much more general and common. Think of point-clouds that have noise-measurement, or that have resulted from different ways of sampling. Furthermore, such distaces could be used to measure the difference between discretely sampled disributions, under optimal alignment
>
> Yes, thank you for proposing these extra applications.
>
> >the experiment focuses on demonstrating the advantage of being a complete invariant, but does not analyze the advantages of having the other properties. I understand that the theory wasn't fully developed in such directions (e.g. assuming equal number of points), but it would have been useful to empirically demonstrate the merits of the invariant in other ways, even under controlled synthetic experiments.
>
> Yes, this is the first paper introducing the new realizable invariants, so more developments are planned. The property of Lipschitz bi-continuity was empirically demonstrated by finding close pairs of molecules in QM9. The distances in Table 3 have the following physical meaning. If we perturb every atom up to epsilon (Euclidean distance measured in Angstroms), the invariant changes up to 6 epsilon by Theorem 5.1 (experimental constant less than 3 for QM9). By Theorem 5.4, any distance d between invariants guarantees a rigid motion that matches all atoms up to 3root(2)d, also in Angstroms. We continue below.

---

> > ### Author Response · Authors · 2024-11-15
> > **Continuation of the response above**
> >
> > >There is a lack in addressing how the proposed idea could be beneficial for the more common tasks that are performed on unordered point-clouds, such as: alignment (perhaps with noise or partial overlap), classification, detection, segmentation and even generative modeling.
> >
> > Yes, you are right that we can include the potential benefits for other tasks, thank you for your helpful suggestions.
> >
> > >I'm note sure whether there is a clear charcterization regarding what are the cases in which one would decide to use the suggested NDP, over the simpler and cheaper alternatives. Is there a principled way the you would suggest?
> >
> > Yes, section 6 described a hierarchical approach starting from the fastest incomplete invariants and then gradually progressing to the stronger and finally complete invariants. For the QM9 database, selecting 1% of closest pairs at every stage sufficed to finish all comparisons within a few hours. Larger datasets might require smaller thresholds or subdividing the first linear-time invariant SRV into smaller (fixed-size) vectors.
> >
> > >The 'hierarchical' way used seems to only be suitable for the particular case of classifying data by exact equivalence (where the cheaper invariants provide fast 'rejection').
> >
> > The Lipschitz continuity of all invariants in the hierarchy SRV, SDV, PDD, NDP allows us to start by mapping any database with the faster invariants as in Figure 7. Any subset or hot spot (dense cluster) on such a map can be zoomed in and expanded with stronger invariants. For example, the QM9 subset with SRV_1=SRV_2 on the horizontal axis in Figure 7 (left) was re-drawn with different coordinates in Figure 7 (right).
> >
> > >It is claimed in the experiment that the NDP provides best seperation.
> >
> > This best separation refers to the largest distance (between NDP invariants) in Table 3.
> >
> > >Is there any evidence that the metric on it is well aligned with what would be considered a good measure for the distance from being equivalent (under rigid transformation)?
> >
> > Yes, Theorems 5.1 and 5.4 justify the Lipschitz bi-continuity. If points are slightly perturbed, NDP changes only slightly. Conversely, if invariants are close in the metric NBM, the given clouds can be matched by a explicitly constructed rigid motion also up to a small perturbation.
> >
> > >Have you tried testing how it looks when starting from perfect equivalence and degrading in different ways (such as noise in the points, or non-perfect rigidity)?
> >
> > Yes, Figure 5 (left) shows how several distances change when the noise bound goes up, illustrating Theorem 5.1.
> >
> > >Are there relaxations with major gains that could make the approach more practical? Such as computing smoother (non permutation-like 1-1 mappings) matchings?
> >
> > Because we used only smooth functions (squared distances and dot products of vectors, divided by the larges radius R, which is away from 0), all components of the NDP invariant are smooth everywhere apart from singular configurations on identified boundaries. Appendix B describes maps of invariant spaces for 3 and 4 points with such boundary identifications, e.g. for the subspace of isosceles triangles. All these identifications can be made smooth everywhere  by embedding the invariant space into a slightly higher dimensional Euclidean space. For example, our flat geographic maps have two coordinates (with boundary identifications) but the surface of a sphere is smoothly embedded in R^3.
> >
> > If there are any more questions, we would be happy to answer.

---

### Official Review · Reviewer_gcAh · 2024-11-03

**Soundness:** 2
**Presentation:** 3
**Contribution:** 3
**Rating:** 6
**Confidence:** 4

**Summary:**

This paper introduces a novel approach to parametrize and analyze clouds of unordered points in the Euclidean plane, with a focus on their rigid shape under translations and rotations. The authors give a rigorous theoretical foundation for the proposed invariant and metric, proving their properties and demonstrating their effectiveness in higher dimensions. Experiments on QM9 database also reveal the efficacy of the designed invariant and metric.

**Strengths:**

1. The authors solve a central problem, i.e., Problem 1.1 defined in the paper, which aims finding an invariant that satisfies completeness, metric axioms, Lipschitz continuity, realizability, point matching, and computability in polynomial time.

2. The methods and results are not limited to two dimensions.

**Weaknesses:**

1. Although the proposed invariants are bi-Lipschitz continuous, meaning they mainly maintain a continuous relationship regarding small perturbations in the point cloud. This maybe difficult for large perturbations.

2. The method can be extended to higher dimensions, but the computational complexity in determining the invariants and metrics could become more demanding. Similarly, the method's scalability, especially for very large point clouds, might also be a concern.

3. My another concern is regarding the experiments. The current experiments are primarily conducted on the QM9 database of molecules. While this provides a strong proof of concept, the method's performance in other domains, such as different types of geometric data or in higher dimensions, requires further investigation.

**Questions:**

Please see the weakness part.

---

> ### Author Response · Authors · 2024-11-15
> **Thank you for the great summary and thoughtful questions**
>
> >The authors solve a central problem, i.e., Problem 1.1 defined in the paper, which aims finding an invariant that satisfies completeness, metric axioms, Lipschitz continuity, realizability, point matching, and computability in polynomial time. The methods and results are not limited to two dimensions.
>
> Thank you for thoroughly summarizing all the conditions satisfied by the new invariants, also in higher dimensions.
>
> >Although the proposed invariants are bi-Lipschitz continuous, meaning they mainly maintain a continuous relationship regarding small perturbations in the point cloud. This maybe difficult for large perturbations.
>
> The continuity results (Theorems 5.1 and 5.4) have no restrictions on the size of noise (epsilon), hence all larger perturbations are also covered.
>
> >The method can be extended to higher dimensions, but the computational complexity in determining the invariants and metrics could become more demanding. Similarly, the method's scalability, especially for very large point clouds, might also be a concern.
>
> Yes, the computational complexity grows with the dimension n. However, the most practical dimensions are n=2, 3. More importantly, the proposed complete invariants need practical computations only in cases when the past incomplete invariants cannot distinguish point clouds. The simpler invariants (SRV, SDV, PDD) are computed in a linear or quadratic time in the number of points for any dimension. Section 6 describes the hierarchical computations going from 873,527,974 pairs of 3D atomic clouds to 10K pairs by the new complete invariants, all within few hours on a modest desktop computer.
>
> >My another concern is regarding the experiments. The current experiments are primarily conducted on the QM9 database of molecules. While this provides a strong proof of concept, the method's performance in other domains, such as different types of geometric data or in higher dimensions, requires further investigation.
>
> QM9 database seems the largest public dataset of molecules with unordered atoms, which were thoroughly optimized by the Density Functional Theory tools. The Protein Data Bank has chains with ordered atoms, so simpler invariants will suffice. For any other dataset of point clouds, the same hierarchical approach can be applied: first comparing by the linear-time invariant SRV (Sorted Radial Vector), then a smaller fraction by the quadratic-time invariants SDV (Sorted Distance Vector) and PDD (Pointwise Distance Distribution), so the complete invariant NDP (Nested Distributed Projection) will be needed only for a final tiny subset of very close pairs. Depending on a given size or dimension of data, we can adjust closeness thresholds (top 1% closest pairs were selected at each stage of QM9) or even further subdivide the faster invariants, e.g. by comparing only the first few components of SRV instead of full vectors at the initial stage.
>
> If there are any more questions, we would be happy to answer.

---

> > ### Comment · Reviewer_gcAh · 2024-11-24
> > **Thanks for the reply**
> >
> > Thanks for the response.
> >
> > It would be better to conduct experiments that demonstrate the continuity results under larger noisy perturbations, which could further validate the method.

---

> > > ### Author Response · Authors · 2024-11-25
> > > **Thank you for your reply**
> > >
> > > >It would be better to conduct experiments that demonstrate the continuity results under larger noisy perturbations, which could further validate the method.
> > >
> > > Dear Reviewer gcAh, thank you for the nice proposal. The original submission validated the continuity (proved in Theorem 5.1 with Lipschitz constant 6) by Figure 5 (left), where the noise bound was up to 0.1 (about 10% of the cloud size).
> > >
> > > Following your helpful advice, we have extended this experiment to the noise bound 1 (now 100% of the cloud size), see the updated version of Figure 5 (left). The experimental Lipschitz constant lambda is still below 2 as in the previous smaller scale experiment, further validating Theorem 5.1.
> > >
> > > The revision has all conclusions under section 7 title "Discussion: conclusions, limitations, and significance". We have also corrected all minor typos. Thank you again for the supportive suggestion.

---

### Official Review · Reviewer_4XrN · 2024-11-03

**Soundness:** 4
**Presentation:** 2
**Contribution:** 4
**Rating:** 8
**Confidence:** 3

**Summary:**

This paper introduces an invariant and metric for unordered point clouds in the Euclidean plane, designed to meet specific desirable properties. The invariant is complete, ensuring that two point clouds related by a rigid transform can be uniquely matched. It is also realizable, enabling the reconstruction of the point cloud from the invariant up to an unknown rigid transform. The metric is Lipschitz continuous, meaning that bounded perturbations in the point cloud result in bounded changes in the distances between the invariants. Additionally, it guarantees that two close invariants correspond to point clouds that can be matched through a rigid motion, with a small Euclidean distance between matching points. Both the invariant and the metric are computable in polynomial time.

The paper includes experiments in chemistry using a publicly available dataset, demonstrating the method’s ability to identify closely related molecules based on the positions of their atom centers.

**Strengths:**

The paper formally defines the problem of finding complete invariants and their corresponding metric for clouds of unordered points, providing formal proofs for all proposed solutions.  According to the authors, this is the first complete invariant and associated metric fulfilling the conditions proposed in the paper. In this regard, the paper makes a significant theoretical contribution to the field.

The method's practical applicability is demonstrated through experiments in molecule matching within the field of Chemistry. The completeness of the invariant has a real impact on the results, separating molecules that are at a close distance from other invariants and metrics from the state-of-the-art.

Additionally, the paper includes extensive appendices that extend the invariant to any dimension d>2

**Weaknesses:**

The main issue with the paper is its readability, which is affected by several factors:

- Notation is introduced gradually throughout the paper, making it difficult to refer to where certain concepts are defined. This requires extensive back-and-forth searching, disrupting the reading flow.
- The number of acronyms is excessive and often very similar (e.g., NDP, NCP, BMD, NBM), making them hard to distinguish. Using different names or aliases could help reduce this confusion.
- Some sections of the paper lack clarity, such as the computation of the Bottleneck Distance in Definition 4.1.
- The paper does not follow a traditional structure and lacks a conclusion section.
The length of the paper and appendices is overwhelming. The extension for d>2 dimensions should be in a separate paper to better focus the content.

The invariant and metric are more computationally complex than other existing (incomplete) invariants and metrics.

**Questions:**

Q1: More insight is needed on the NBM computation in Definition 4.1. How is it derived using Efrat et al. (2001)? Additionally, why does the Point-Based Representation Metric require normalization for rigid motions without scale difference?

Q2: Corollary 5.5 on continuous morphing requires further clarification. How is the continuous path defined between NDP(A) and NDP(B)?

Q3: What would be the impact of the proposed invariant and its metric when applied to learning-based approaches?

---

> ### Author Response · Authors · 2024-11-15
> **Thank you for the great summary and thoughtful questions**
>
> Sorry about the slow reply because of other commitments in the last few days.
>
> >The paper formally defines the problem of finding complete invariants and their corresponding metric for clouds of unordered points, providing formal proofs for all proposed solutions. According to the authors, this is the first complete invariant and associated metric fulfilling the conditions proposed in the paper. In this regard, the paper makes a significant theoretical contribution to the field. The method's practical applicability is demonstrated through experiments in molecule matching within the field of Chemistry. The completeness of the invariant has a real impact on the results, separating molecules that are at a close distance from other invariants and metrics from the state-of-the-art.
>
> Thank you for correctly summarizing the strengths.
>
> >Notation is introduced gradually throughout the paper, making it difficult to refer to where certain concepts are defined.
>
> The key acronyms with references to all definitions re collected in Table 1 shorty after new notations appear in section 3.
>
> >The number of acronyms is excessive and often very similar (e.g., NDP, NCP, BMD, NBM), making them hard to distinguish. Using different names or aliases could help reduce this confusion.
>
> In this list only NDP and NCP are similar because they appear in the same Definition 3.4: NCP (Nested Compressed Projection) is a smaller (weighted) version of NDP (Nested Distributed Projection). Since we mainly use NDP, the acronym NCP can be dropped.
>
> >Some sections of the paper lack clarity, such as the computation of the Bottleneck Distance in Definition 4.1.
>
> Definition 4.1 is a standard definition of the bottleneck distance, see https://en.wikipedia.org/wiki/Topological_data_analysis#Concepts, whose computationally efficient algorithm is reference right after this definition in lines "Though the bottleneck distance is introduced as a minimum for m! bijections A → B between fixed m-point clouds, Theorem 6.5 in Efrat et al. (2001) computes BD(A,B) in time O(m^1.5 log^2 m)."
>
> >The paper does not follow a traditional structure and lacks a conclusion section.
>
> Section 6 (Experiments on 130K+ molecules, limitations, and significance) finishes with conclusions in lines 527-539. Could we ask what is meant by a traditional structure?
>
> >The extension for d>2 dimensions should be in a separate paper to better focus the content.
>
> Yes, the higher-dimensional case will be in another extended paper.
>
> >The invariant and metric are more computationally complex than other existing (incomplete) invariants and metrics.
>
> Yes, the completeness of the new invariant requires more complexity. Such a complete invariant finishes the hierarchy of the past incomplete invariants and is needed in case when the past invariant cannot distinguish point clouds.
>
> >Q1: More insight is needed on the NBM computation in Definition 4.1. How is it derived using Efrat et al. (2001)?
>
> The NBM distance is introduced in Definition 4.4. The actual computation extends the well-known algorithm of the Bottleneck Matching Distance (Definition 4.3) with edge weights obtained from the new Point-based Representation Metric (Definition 4.2).
>
> >Additionally, why does the Point-Based Representation Metric require normalization for rigid motions without scale difference?
>
> The Point-Based Representation Metric is based on Point-Based Representation consisting of squared distances and dot products of vectors (to make all components smooth by avoiding square roots), which behave "roughly quadratically" under perturbations and hence are not Lipschitz continuous. The extra division by the radius R makes the resulting quantities Lipschitz continuous, also measured in original units of coordinates (Angstroms, not squared Angstroms).
>
> >Q2: Corollary 5.5 on continuous morphing requires further clarification. How is the continuous path defined between NDP(A) and NDP(B)?
>
> This path is the straight line in the space NDP(CRS(R^2;m)) of realizable invariants. Appendix~B discusses how the invariant space is defined by inequalities, which make this space convex, also mapping this space in partial cases.
>
> >Q3: What would be the impact of the proposed invariant and its metric when applied to learning-based approaches?
>
> The learning can now be done on invariant spaces instead of latent spaces (based on incomplete or discontinuous descriptors), because any point in the space of realizable invariants can be efficiently inverted to a unique rigid cloud.
>
> If there are any more questions, we would be happy to answer.

---

> > ### Comment · Reviewer_4XrN · 2024-11-26
> > **Thanks for the reply**
> >
> > I thank the authors for answering my questions. It is clearer now. This is a complicated topic, and it's difficult to produce a paper that is easy to read for all kinds of audiences.

---

> > > ### Author Response · Authors · 2024-11-26
> > > **Thank you for your understanding!**
> > >
> > > >I thank the authors for answering my questions. It is clearer now. This is a complicated topic, and it's difficult to produce a paper that is easy to read for all kinds of audiences.
> > >
> > > Dear reviewer 4XrN, thank you for your helpful appreciation of the effort to make the paper readable.

---

### Author Response · Authors · 2024-11-23
**any acknowledgement?**

Dear reviewers and area chair, may we ask if we should expect any responses to our rebuttals submitted more than 8 days ago? Thank you!

---

### Author Response · Authors · 2024-12-04
**Thank you for the feedback**

Dear reviewers, thank you for the detailed comments and thoughtful questions, which will certainly help to make the paper easier to read. Kind regards, the authors.

---

### Meta-Review · Area_Chair_gCh9 · 2024-12-16

**Metareview:**

The paper presents bi-continuous and complete SE(2)-invariants from unordered points, which can be used for 1-1 matching between points in clouds and are computable in a quadratic time of the number of points.

The merit of the paper is that the authors establish a theoretical foundation for the proposed invariant and metric. Also experiments in QM9 database were performed verifying the method.

However, the practical advantages of the paper are not sufficiently demonstrated. Readability of the paper is not easy for people in the community of ICLR. To what extents the method solves the current difficulties in point clouds requires further studies and more experiments on more datasets are needed.

**Additional Comments On Reviewer Discussion:**

Reviewer 4XrN raised the writing drawback. The authors made explanations. Reviewer 4XrN thought it is clear.

Reviewer gcAh raised the robustness and scalability drawbacks and insufficient experiments.  About the  robustness, the authors added experiments showing  the Nested Bottleneck Metric with increasing noise, but didn't show the 1-1 matching.  About the scalability, the author agreed that the computational complexity grows with the dimension n and they thought the most practical dimensions are n=2, 3.  About the experiments,  the author said they can use the same hierarchical approach but didn't provide any results on other datasets.

Reviewer CoFa raised the applicability drawback. The authors explained they can be faster but no proof or experiments is provided.

Reviewer mWsb raised the usefulness, insufficient proof, and writing drawbacks. The authors explained the drawbacks but the result is unconvincing.

In the final discussions, the reviewers and AC thought the testing is limited, genuine contribution is not clear and lacks more proofs, the readability needs revisions.

---

### Decision · Program_Chairs · 2025-01-22

Reject